# Robust GNN Watermarking via Implicit Perception of Topological Invariants

## Abstract

Graph Neural Networks (GNNs) are valuable intellectual property, yet most watermarks use backdoor triggers that break under common model edits and create ownership ambiguity. To tackle this challenge, we present **InvGNN-WM**, which ties ownership to a model's implicit perception of a graph invariant, enabling trigger-free, black-box verification with negligible task impact. A lightweight head predicts normalized algebraic connectivity in an owner-private carrier set; a sign-sensitive decoder outputs bits, and a calibrated threshold $\tau(\alpha)$ controls the false-positive rate. Across diverse node and graph classification datasets and backbones, **InvGNN-WM** matches clean accuracy while yielding higher watermark accuracy than trigger- and explanation-based baselines. It remains strong under unstructured pruning, fine-tuning, and post-training quantization; plain knowledge distillation (KD) weakens the mark, while KD with a watermark loss (KD+WM) restores it. We provide guarantees for imperceptibility and robustness, and prove that exact removal is NP-complete.

## 1 Introduction

Graph Neural Networks (GNNs) are widely used in drug discovery, social networks, and recommendation (Wu et al., 2021; Gilmer et al., 2017; Hu et al., 2020), where training relies on proprietary data and the released models are valuable intellectual property (Adi et al., 2018). Watermarking aims to certify ownership after deployment, but many GNN schemes rely on *trigger keys* (Zhang et al., 2021): the model is trained to respond to graphs outside the task distribution $\mathcal{D}_{\text{task}}$ (OOD). During fine-tuning, pruning, and distillation, optimization only sees samples from $\mathcal{D}_{\text{task}}$, so trigger-specific parameters for these OOD graphs receive no preserving signal and drift or are pruned, which weakens the mark (Li et al., 2021). OOD triggers also make black-box verification unstable: the owner must choose a decision threshold without knowing the impostor distribution on private triggers, which complicates false-positive control (Saha et al., 2022), as observed in vision when the watermark is detached from normal inference (Uchida et al., 2017).

We instead ask whether ownership can be tied to the same computation that solves the learning task on graph $G$, so that adding the watermark leaves utility essentially unchanged. We introduce **InvGNN-WM**, which binds ownership to a model's implicit perception of a graph invariant. Concretely, the GNN learns to predict an invariant $I(G)$ on owner-private carrier graphs $G$, a lightweight head maps graph-level embeddings of $G$ to an estimate of $I(G)$, a separable sign-sensitive decoder turns these estimates into bits, and a calibrated threshold $\tau(\alpha)$ sets the false-positive rate for black-box verification. Our theory and algorithms are stated for a *generic* permutation-invariant functional $I(G)$ that admits a Lipschitz predictor and a separable sign-sensitive decoder. In experiments we instantiate $I(G)$ with the normalized algebraic connectivity $\tilde{\lambda}_2(G)$ of graph $G$ (Fiedler, 1973; Chung, 1997). Because expressive message-passing GNNs encode global structure of graph $G$ (Gilmer et al., 2017; Xu et al., 2023), coupling ownership to invariant perception ties the mark to the model's core logic rather than to exogenous patterns.

*On the theory side*, we formalize a quantitative coupling between watermark removal and task degradation. We define a robustness margin on the carrier set of graphs $G$ that measures how far watermark scores lie from the decision boundary, and summarize common edits (fine-tuning, unstructured pruning, distillation) into a composite drift budget. Under a local Polyak–Lojasiewicz condition on the task loss and a Lipschitz bound on the perception head on graph $G$, any edit that

reliably flips watermark bits must exceed the margin and therefore incur a nontrivial increase in task loss. In complement, choosing a small watermark weight and controlling the head's sensitivity via spectral normalization (Miyato et al., 2018) keeps this loss increase within a controlled range, so the watermark can be embedded with negligible task impact. The ownership statistic is compared to a threshold calibrated to a target false-positive level, and verification errors decay exponentially in the key length via large-deviation bounds (Hoeffding, 1963). Finally, exact removal is NP-complete under our separable, sign-sensitive decoder, which explains why practical attacks resort to heuristic edits already covered by the margin analysis.

*Empirically*, InvGNN-WM matches clean task accuracy across diverse node- and graph-classification datasets and backbones while delivering high watermark accuracy on graphs $G$ from the carrier set. The mark remains stable under unstructured pruning, fine-tuning, and post-training quantization; plain KD weakens the mark, while a simple KD with a watermark loss (KD+WM) restores it. Targeted "killshot" edits that collapse trigger-based designs have a limited effect on our invariant-coupled scheme, and our experiments also examine robustness–utility trade-offs, carrier secrecy, and ownership ambiguity.

**Contributions. (1) Method.** *InvGNN-WM* ties ownership to a GNN's implicit perception of a graph invariant through an invariant–carrier protocol on graph $G$, enabling trigger-free, black-box verification with minimal task impact. **(2) Theory.** We give guarantees for imperceptibility and robustness via a margin–budget analysis, show key uniqueness across independent carrier sets with calibrated thresholds, and prove that exact removal under our decoder is NP-complete. **(3) Evaluation.** Across datasets, backbones, and edit types, InvGNN-WM matches clean accuracy, achieves higher watermark fidelity than prior GNN watermarks, and remains reliable under pruning, fine-tuning, quantization, and KD+WM, while additional diagnostics study carrier secrecy and ownership ambiguity.

## 2 Preliminaries

This section defines the notation for Graph Neural Networks (GNNs), introduces the graph Laplacian spectrum used for watermarking, and formalizes the watermarking framework, including the threat model and the assumptions behind our theoretical guarantees. Throughout, $\mathcal{D}_{\text{task}}$ denotes the data distribution over simple, undirected graphs, $\mathcal{G}$ is the space of such graphs, and $[m] := \{1, \ldots, m\}$.

### 2.1 Graph Neural Networks (GNNs)

A simple undirected graph $G = (V, E) \in \mathcal{G}$ consists of $n := |V|$ nodes with feature vectors $x_v \in \mathbb{R}^{d_f}$ and a set of edges $E \subseteq \binom{V}{2}$. A message-passing GNN, parameterized by $\theta \in \mathbb{R}^d$, computes node representations $h_v^{(\ell)}$ across $L$ layers following the standard formulation in Wu et al. (2021). Initial representations $h_v^{(0)} := x_v, \forall \ell = 1 \ldots L$ are updated as:

$$m_v^{(\ell)} := \text{Aggregate}^{(\ell)}\big(\{h_u^{(\ell-1)} : u \in \mathcal{N}(v)\}\big), \ h_v^{(\ell)} := \text{Update}^{(\ell)}\big(h_v^{(\ell-1)}, m_v^{(\ell)}\big), \quad (1)$$

where $\mathcal{N}(v)$ is the set of neighbors of node $v$. A final permutation-invariant $\text{Readout}$ function produces a graph-level embedding. The GNN is trained by minimizing a supervised loss $\mathcal{L}_{\text{task}}(\theta)$.

### 2.2 A Watermark from the Laplacian Spectrum

We embed the watermark through a global graph property that the GNN already uses for reasoning on graph $G$. The Laplacian spectrum captures global structure and connectivity of $G$ (Chung, 1997), so it links the watermark to the model's computation. While our theory applies to any permutation-invariant graph functional $I(G)$, we instantiate $I(G)$ with the normalized algebraic connectivity $\tilde{\lambda}_2(G)$ for its stability and interpretability.

Let $\mathbf{A} \in \{0, 1\}^{n \times n}$ be the adjacency matrix of a graph $G$ and $\mathbf{D} := \text{diag}(\mathbf{A1})$ its degree matrix. The combinatorial Laplacian is $\mathbf{L} := \mathbf{D} - \mathbf{A}$ with eigenvalues $0 = \lambda_1 \leq \cdots \leq \lambda_n$, the Laplacian spectrum of $G$. We focus on $\lambda_2(G)$, the algebraic connectivity. In practice we add a small diagonal perturbation $\varepsilon\mathbf{I}$ (with $\varepsilon = 10^{-6}$) for numerical stability; the analysis only needs continuity of $\lambda_2(G)$, not distinct eigenvalues.

To make algebraic connectivity comparable across datasets, we normalize $\lambda_2(G)$ to $[0, 1]$. Let $\lambda_{\min}$ and $\lambda_{\mathrm{scale}}$ be the 5th and 95th empirical percentiles of $\lambda_2(G)$ over graphs drawn from $\mathcal{D}_{\mathrm{task}}$. We define

$$\tilde{\lambda}_2(G) \; := \; \mathrm{clip}_{[0,1]}\left( \frac{\lambda_2(G) - \lambda_{\min}}{\lambda_{\mathrm{scale}} - \lambda_{\min}} \right), \tag{2}$$

where $\mathrm{clip}_{[0,1]}(z) := \min\{1, \max\{0, z\}\}$. Empirically, this concentrates almost all observed $\lambda_2(G)$ values into $[0, 1]$ while safely clipping a few outliers.

We introduce a scalar perception head $s_\theta : \mathcal{G} \to [0, 1]$ that estimates the normalized invariant from a graph embedding. For a private carrier set $\{G_W^{(k)}\}_{k=1}^m \subset \mathcal{G}$, the robustness margin of a model with parameters $\theta$ is

$$\kappa_{\mathrm{marg}}(\theta) \; := \; \min_{k \in [m]} \big| s_\theta(G_W^{(k)}) - \tfrac{1}{2} \big|.$$

This margin $\kappa_{\mathrm{marg}}(\theta)$ for graphs $G_W^{(k)}$ is the minimum change in $s_\theta$ required to flip any embedded bit and serves as a measure of watermark resilience. For ease of reference, we use $I(G)$ for a scalar graph invariant (instantiated as $\tilde{\lambda}_2(G)$ in experiments), $s_\theta$ for the scalar perception head on graph $G$, and $\mathcal{G}_W = \{G_W^{(k)}\}_{k=1}^m \subset \mathcal{G}$ for the private carrier set. The quantity $\kappa_{\mathrm{marg}}(\theta)$ is the robustness margin defined above, $\tau$ will denote a verification threshold in Section 4, $\delta > 0$ the radius of parameter perturbations around a task-only solution in Section 5, and $\beta_{\mathrm{wm}} > 0$ the weight of the watermark loss in the joint training objective in Section 4.

## 2.3 Watermarking Framework and Assumptions

A secure and practical watermarking scheme should satisfy four key properties (Zhao et al., 2021; Xu et al., 2023; Downer et al., 2025):
- **Imperceptibility:** Embedding the watermark should not noticeably harm primary task performance.
- **Robustness:** The watermark must remain detectable after common model modifications, like fine-tuning, pruning, or knowledge distillation.
- **Uniqueness:** Secret keys must yield statistically distinct watermarks to prevent ownership disputes.
- **Unremovability (Hardness):** Watermarks should be hard to remove without the secret key.

**Threat Model.** We consider a gray-box attacker that knows (i) the backbone GNN architecture, (ii) the training objective and optimization procedure, and (iii) the watermarking algorithm, including the scalar perception head tied to a graph invariant and the carrier-sampling protocol. However, the attacker does *not* know the owner's secret key, that is, the concrete carrier set $\mathcal{G}_W = \{G_W^{(1)}, \ldots, G_W^{(m)}\} \subset \mathcal{G}$ and its embedded bit pattern. The attacker can obtain any released watermarked model and apply structured pruning, fine-tuning on auxiliary data, post-training quantization, plain KD to a student architecture, or black-box model extraction; we interpret these operations as watermark-removal attacks that require no access to the secret carrier graphs. By contrast, the KD+WM procedure that we evaluate later is an *owner-side* recovery or transfer mechanism: it assumes access to $\mathcal{G}_W$ and the key and is therefore not available to the attacker.

**Assumptions.** Our theoretical guarantees rely on the following assumptions; the concrete protocols for enforcing them are in Appendix B.

**Assumption 2.1** (Graph-level Separation). *The carrier graph set is disjoint from the task data support, i.e., $\mathcal{G}_W \cap \mathrm{supp}(\mathcal{D}_{task}) = \varnothing$. We enforce this by sampling carriers from graphs drawn from $\mathcal{D}_{task}$ and applying degree-preserving double-edge swaps. Candidates that are isomorphic to task graphs are removed using a Weisfeiler–Lehman (WL) hash, and we reject candidates whose statistics (degree distribution, clustering coefficient, spectral radius) are not sufficiently separated from those of $\mathcal{D}_{task}$ under Kolmogorov–Smirnov tests. The resulting carrier set has empirical support that does not overlap with $\mathrm{supp}(\mathcal{D}_{task})$; see Appendix B.1 for pseudocode.*

**Assumption 2.2** (Empirical $\rho$-mixing). *The carrier graphs are weakly correlated. Formally, there exists a constant $\rho_0 \leq 10^{-3}$ such that for all $i \neq j$ and any measurable $f : \mathcal{G} \to [0, 1]$,*

$$\big| \mathrm{Corr}\big(f(G_W^{(i)}), f(G_W^{(j)})\big) \big| \leq \rho_0.$$

*In practice, we estimate an upper bound $\hat{\rho}_0$ by computing pairwise Pearson correlations $\mathrm{Corr}\big(s_{\theta^\star}(G_W^{(i)}), s_{\theta^\star}(G_W^{(j)})\big)$ across all distinct carrier pairs. Across all datasets we observe*

$\hat{\rho}_0 < 7.6 \times 10^{-4}$ *and conservatively set* $\rho_0 := 10^{-3}$ *in our analysis. This empirical bound enters the Hoeffding-type deviation inequality used to calibrate the verification threshold* $\tau$ *in Section 4.*

**Assumption 2.3** (Perception Lipschitzness). *The perception head* $s_\theta$ *is* $L_s$-*Lipschitz with respect to its parameters* $\theta$ *in a neighborhood of the trained solution, that is,* $\left| s_{\theta + \Delta\theta}(G) - s_\theta(G) \right| \leq L_s \left\| \Delta\theta \right\|$ *for small perturbations* $\Delta\theta$. *We encourage a finite and moderate* $L_s$ *by combining weight clipping on the perception head with an explicit penalty on* $\|\nabla_\theta s_\theta\|$ *during training; spectral normalization on the head further controls input-Lipschitzness and keeps gradients bounded. Empirically these regularizers keep* $L_s$ *bounded by a small constant across all models, and in our theorems we treat* $L_s$ *as such a fixed constant.*

**Local Polyak–Lojasiewicz (PL) condition.**   We also assume a standard local PL condition for the task loss $\mathcal{L}_{\text{task}}$: there exist $\mu > 0$, a reference parameter vector $\theta^\star$, and a radius $\delta > 0$ such that

$$\frac{1}{2} \left\| \nabla_\theta \mathcal{L}_{\text{task}}(\theta) \right\|_2^2 \; \geq \; \mu \big( \mathcal{L}_{\text{task}}(\theta) - \mathcal{L}_{\text{task}}(\theta^\star) \big) \quad \text{for all } \theta \text{ with } \|\theta - \theta^\star\|_2 \leq \delta. \tag{3}$$

The PL condition is weaker than strong convexity and is known to hold for many over-parameterized neural networks (Karimi et al., 2016); here it is used only to relate the decrease in task loss to the gradient norm in a local neighborhood around $\theta^\star$.

## 3 Proposed Method: InvGNN-WM

We introduce **Invariant-based Graph Neural-Network Watermarking (InvGNN-WM)**, a framework that embeds ownership by training a GNN to perceive a topological invariant. The core of the method is a differentiable perception function that links the GNN's parameters to a graph property, such as the algebraic connectivity $\lambda_2(G)$. This function is optimized via an auxiliary loss, weaving the watermark into the model's weights without altering the GNN's message-passing architecture.

### 3.1 Watermark Design

The watermark is defined by three components: a private set of $m$ **carrier graphs** $\mathcal{G}_W$, a secret key $W$ induced by the carriers, and an **invariant-perception function** $s_\theta(G)$ that connects them. The owner first generates $\mathcal{G}_W = \{G_W^{(k)}\}_{k=1}^m$ using the adaptive rewiring protocol from Section 2.3, ensuring the graphs are out-of-support but statistically similar to the task data. The secret key $W = (w_k)_{k=1}^m$ is then deterministically induced by the normalized algebraic connectivity of these graphs:

$$w_k \; := \; \mathbf{1}\!\left[ \tilde{\lambda}_2\big(G_W^{(k)}\big) \geq \tfrac{1}{2} \right], \quad k = 1, \ldots, m,$$

where $\tilde{\lambda}_2(\cdot)$ is the normalized algebraic connectivity defined in Equation (2). Let $h_G \in \mathbb{R}^{d_h}$ denote the graph-level embedding produced by the GNN's $\mathrm{Readout}$ layer for a graph $G$. The perception function is implemented as a lightweight scalar head

$$s_\theta(G) \; = \; \sigma\big( g_{\theta_{\text{head}}}(h_G) \big), \tag{4}$$

where $g_{\theta_{\text{head}}}$ is a one-layer MLP with parameters $\theta_{\text{head}}$, and $\sigma$ is a squashing nonlinearity mapping to $[0,1]$. For readability we write $s_\theta$, but in the analysis we use $\theta = (\theta_{\text{back}}, \theta_{\text{head}})$, where $\theta_{\text{back}}$ collects the backbone GNN parameters. The head is trained to regress the precomputed targets $\tilde{\lambda}_2(G)$ on carrier graphs $G$, tying the invariant to the model's internal representations through the shared backbone. The value $\tilde{\lambda}_2(G)$ is computed once from the Laplacian eigenvalues of graph $G$ and used as a fixed supervision signal; gradients do not flow through the eigen-decomposition. All weights in the perception head are spectrally normalized and subject to gradient penalties as described in Assumption 2.3, which helps keep the Lipschitz constant $L_s$ bounded.

### 3.2 Embedding via Joint Training Objective

The watermark is embedded by training the GNN with a *joint* loss function that combines the primary task and the invariant-perception objective:

$$J(\theta) = \mathcal{L}_{\text{task}}(\theta) + \beta_{\text{wm}} \, \mathcal{L}_{\text{wm}}(\theta). \tag{5}$$

The first term, $\mathcal{L}_{\text{task}}$, is the conventional supervised loss for the primary task on graphs drawn from $\mathcal{D}_{\text{task}}$, which preserves model utility. The second term, $\mathcal{L}_{\text{wm}}$, is a regression loss that encourages the perception head $s_\theta$ to correctly estimate the normalized algebraic connectivity for each carrier graph:

$$\mathcal{L}_{\text{wm}}(\theta) = \frac{1}{m} \sum_{k=1}^{m} \left( s_\theta(G_W^{(k)}) - \tilde{\lambda}_2(G_W^{(k)}) \right)^2. \tag{6}$$

The hyperparameter $\beta_{\text{wm}} > 0$ balances the two objectives. Its value is chosen to be less than or equal to a theoretical maximum, $\beta_{\max}$, derived in Theorem 4.1, to guarantee that the task performance is not degraded beyond a user-defined tolerance.

In practice, each SGD step proceeds as follows. We first sample a minibatch of task graphs from $\mathcal{D}_{\text{task}}$ and compute $\mathcal{L}_{\text{task}}$ on this minibatch using the standard task head. We then sample a minibatch of carrier graphs from $\mathcal{G}_W$ (or use the full carrier set when it is small), pass them through the same backbone to obtain their graph embeddings $h_{G_W^{(k)}}$, and evaluate the scalar head $s_\theta(G_W^{(k)})$ to compute $\mathcal{L}_{\text{wm}}$ according to Equation (6). The total loss $J(\theta)$ in Equation (5) is backpropagated through both the backbone and the scalar head, so that the invariant-perception objective shapes the internal representations used for the downstream task.

### 3.3 Embedding and Verification Workflow

InvGNN-WM consists of two main stages: embedding the watermark and verifying ownership.

• **Embedding:** The owner first computes the normalized targets $\tilde{\lambda}_2(G_W^{(k)})$ for the private carriers $\mathcal{G}_W$ using Equation (2), which deterministically induces the secret key $W$. The GNN parameters $\theta$ are then initialized (for example from a task-only checkpoint) and optimized to minimize the joint loss $J(\theta)$ in Equation (5), using minibatches that interleave task graphs from $\mathcal{D}_{\text{task}}$ and carrier graphs from $\mathcal{G}_W$. This procedure embeds the ownership signal into the shared backbone while keeping task performance close to that of the task-only model, as quantified by Theorem 4.1.

• **Verification:** To verify ownership of a suspect model $M^\star$ with parameters $\theta^\star$, the owner uses the private carriers $\mathcal{G}_W$. For each carrier $G_W^{(k)}$, the owner queries the model to obtain the perception output $s_{\theta^\star}(G_W^{(k)})$ and decodes a bit $\hat{w}_k = \mathbf{1}[s_{\theta^\star}(G_W^{(k)}) \geq 0.5]$. Ownership is confirmed if the number of matching bits, $T = \sum_{k=1}^{m} \mathbf{1}[\hat{w}_k = w_k]$, exceeds a calibrated threshold $\tau$. The threshold is set as $\tau = \lceil m(1 - \varepsilon_{\text{err}}) \rceil$, where $\varepsilon_{\text{err}}$ is determined via Theorem 4.2 to achieve a target false-positive rate $\alpha$ (e.g., $10^{-6}$). Since each carrier graph is small, the verification process, which requires only $m$ forward passes through $M^\star$, is computationally lightweight.

**Why algebraic connectivity and complexity.** We defer the detailed discussion of why we instantiate the invariant $I(G)$ as algebraic connectivity $\lambda_2(G)$ of graph $G$, and the complexity and scalability analysis of computing $\lambda_2(G)$ on carrier graphs $G_W^{(k)}$, to Appendix P.

## 4 Theoretical Guarantees

We provide the theoretical foundation of our watermarking scheme. We establish four properties needed for a practical and secure system: **imperceptibility**, **robustness**, **uniqueness**, and a **hardness** result for *unremovability*. Throughout, the watermark strength is denoted by $\beta_{\text{wm}}$ (see equation 5) to avoid conflict with spectral eigenvalues $\lambda_i$. The robustness margin $\kappa_{\text{marg}}$ is recalled from Section 2.

### 4.1 Imperceptibility

A watermark should not significantly degrade the host model's performance on its primary task. We work in the setting of the local Polyak–Lojasiewicz (PL) condition in Equation (3) for the task loss $\mathcal{L}_{\text{task}}$ around a reference parameter vector $\theta^\star$, together with the parameter-Lipschitz bound for the perception head in Assumption 2.3. Under these regularity conditions, choosing the watermark weight below a data–model threshold keeps the task loss close to the task-only reference solution.

**Theorem 4.1** (Task-loss bound). *Let $\tilde{\theta} := \arg\min_\theta J(\theta)$ with $J(\theta) = \mathcal{L}_{task}(\theta) + \beta_{wm}\mathcal{L}_{wm}(\theta)$, and let $\theta^\star$ be a parameter vector that satisfies the local PL condition equation 3 with constant $\mu_{\mathrm{PL}} > 0$*

*and radius $\delta > 0$. Assume that the perception head $s_\theta$ is $L_s$-Lipschitz with respect to $\theta$ in the same neighborhood as in equation 3, and that $\|\tilde{\theta} - \theta^\star\|_2 \leq \delta$. If*

$$\beta_{\max} := \frac{\sqrt{\mu_{\mathrm{PL}}\,\varepsilon_{\mathrm{task}}}}{\sqrt{2}\,L_s}, \qquad \beta_{wm} \leq \beta_{\max},$$

*then the watermarked model preserves task loss in the sense that*

$$\mathcal{L}_{task}(\tilde{\theta}) - \mathcal{L}_{task}(\theta^\star) \ \leq \ \varepsilon_{\mathrm{task}}.$$

*Sketch.* At any interior minimizer of $J$, the first-order optimality condition gives $\nabla_\theta \mathcal{L}_{\mathrm{task}}(\tilde{\theta}) = -\beta_{\mathrm{wm}} \nabla_\theta \mathcal{L}_{\mathrm{wm}}(\tilde{\theta})$. By the form of $\mathcal{L}_{\mathrm{wm}}(\theta) = \frac{1}{m}\sum_k (s_\theta(G_W^{(k)}) - \tilde{\lambda}_2(G_W^{(k)}))^2$ with both arguments in $[0,1]$ and Assumption 2.3, one has $\|\nabla_\theta \mathcal{L}_{\mathrm{wm}}(\tilde{\theta})\| \leq 2L_s$. Thus $\|\nabla_\theta \mathcal{L}_{\mathrm{task}}(\tilde{\theta})\| \leq 2\beta_{\mathrm{wm}}L_s$. The PL inequality equation 3 with constant $\mu_{\mathrm{PL}}$ then yields

$$\mathcal{L}_{\mathrm{task}}(\tilde{\theta}) - \mathcal{L}_{\mathrm{task}}(\theta^\star) \ \leq \ \frac{\|\nabla_\theta \mathcal{L}_{\mathrm{task}}(\tilde{\theta})\|^2}{2\mu_{\mathrm{PL}}} \ \leq \ \frac{2\,\beta_{\mathrm{wm}}^2 L_s^2}{\mu_{\mathrm{PL}}} \ \leq \ \varepsilon_{\mathrm{task}}.$$

A complete proof, including boundary cases and the dependence on $\delta$, is given in Appendix D. $\qquad\square$

**Calibration.** We estimate $\mu_{\mathrm{PL}}$ and $L_s$ on a held-out split around the task-only solution and set

$$\beta_{\max} \approx \frac{\sqrt{\hat{\mu}_{\mathrm{PL}}\,\varepsilon_{\mathrm{task}}}}{\sqrt{2}\,\hat{L}_s}.$$

In experiments we choose $\beta_{\mathrm{wm}} = \min\{\beta_{\max}, \beta_{\mathrm{val}}\}$, where $\beta_{\mathrm{val}}$ is the largest value on a short grid that keeps the validation degradation within $\varepsilon_{\mathrm{task}}$. Full calibration procedure is described in Appendix D.

## 4.2 ROBUSTNESS

**Watermark margin.** We measure how far each carrier's output lies from the decision threshold,

$$\kappa_{\mathrm{marg}}(\tilde{\theta}) \ := \ \min_{k\in[m]} \Big| s_{\tilde{\theta}}(G_W^{(k)}) - \tfrac{1}{2} \Big|.$$

Positive margin guarantees small changes in the head outputs cannot flip any decoded bit.

**Attack budget.** For an attacked model $\hat{\theta}$ relative to a reference $\theta$, define the head–output drift as

$$\gamma(\hat{\theta};\theta) \ := \ \sup_{G\in\mathcal{G}_W} \big| s_{\hat{\theta}}(G) - s_\theta(G) \big|.$$

Composite attack that (i) fine-tunes $\theta \to \theta^{\mathrm{ft}}$, (ii) prunes a fraction $p_{\mathrm{pr}} \in (0,1]$ to obtain $\theta^{\mathrm{ft,pr}}(p_{\mathrm{pr}})$, and (iii) applies KD) with teacher retention fraction $\rho_{\mathrm{kd}} \in (0,1]$ to produce $\hat{\theta}$, where $\pi_{\mathrm{kd}} := 1 - \rho_{\mathrm{kd}}$ summarizes the strength of the KD step. By the triangle inequality and Assumption 2.3,

$$\gamma(\hat{\theta};\theta) \ \leq \ L_s\,\Delta_\theta \ + \ c_{\mathrm{prune}}\sqrt{p_{\mathrm{pr}}} \ + \ c_{\mathrm{distill}}\,\pi_{\mathrm{kd}}, \tag{7}$$

with $\Delta_\theta := \big\|\mathrm{vec}(\theta^{\mathrm{ft}}) - \mathrm{vec}(\theta)\big\|_2$. The constants $c_{\mathrm{prune}}$ and $c_{\mathrm{distill}}$ are empirical Lipschitz-like bounds calibrated once on a held-out split (see Appendix E).

**Theorem 4.2** (Robustness). *Assume Assumption 2.2 holds for the carrier sequence and let $0 < \varepsilon_{\mathrm{err}} < 1/2$. Consider the detector that accepts ownership when $T(\theta) \geq \tau$ with $\tau = \lceil m(1 - \varepsilon_{\mathrm{err}})\rceil$, where $T(\theta) = \sum_{k=1}^m \mathbf{1}[\hat{w}_k = w_k]$ is the number of matching bits under deterministic decoding. Then, for any null model $\theta_{null}$ and any attacked model $\hat{\theta}$ obtained from a watermarked model $\tilde{\theta}$,*

$$\alpha = \Pr\big[T(\theta_{null}) \geq m(1 - \varepsilon_{\mathrm{err}}) \mid H_0\big] \ \leq \ \exp\big\{-2(1 - c_{\rho_0})\,m\,\varepsilon_{\mathrm{err}}^2\big\}, \tag{8}$$

$$\beta_{\mathrm{fn}} = \Pr\big[T(\hat{\theta}) < m(1 - \varepsilon_{\mathrm{err}}) \mid H_1\big] \ \leq \ \exp\big\{-2(1 - c_{\rho_0})\,m\,(\kappa_{\mathrm{marg}} - \gamma)^2\big\}, \tag{9}$$

*where $\gamma = \gamma(\hat{\theta};\tilde{\theta})$ is any bound satisfying equation 7 and $c_{\rho_0}$ is an explicit weakening factor from a block-concentration argument for $\rho_0$-mixing sequences (we use $c_{\rho_0} \leq 4\rho_0$ in practice; see E). In particular, with deterministic decoding and $\gamma < \kappa_{\mathrm{marg}}$, one has $T(\hat{\theta}) = m$ and thus $\beta_{\mathrm{fn}} = 0$.*

A complete proof, including the blocking construction for $\rho_0$-mixing sequences and the derivation of $c_{\rho_0}$, is given in Appendix E.

**Threshold selection.** Given a target false-positive rate $\alpha$ and an empirical estimate $\hat{\rho}_0$ from Assumption 2.2, we choose

$$\varepsilon_{\text{err}} \;=\; \sqrt{\frac{\log(1/\alpha)}{2(1 - c_{\hat{\rho}_0})\, m}}, \qquad \tau = \left\lceil m\big(1 - \varepsilon_{\text{err}}\big) \right\rceil,$$

which makes the bound in equation 8 equal to $\alpha$ when $c_{\rho_0}$ is replaced by its empirical counterpart $c_{\hat{\rho}_0}$. Full procedures and a worked example are in Appendix E.

## 4.3 UNIQUENESS

To identify an owner reliably, keys induced by independent carrier sets should be statistically distinct. Let the owner's key be $W = (w_k)_{k=1}^m$ with $w_k = \mathbf{1}\big[\tilde{\lambda}_2(G_W^{(k)}) \geq 0.5\big]$. Define the decoded bitstring

$$b(W) \;:=\; \big(\mathbf{1}[\, s_{\tilde{\theta}}(G_W^{(k)}) \geq 0.5\,]\big)_{k=1}^m \in \{0,1\}^m, \qquad F_W := \text{Law}\big(b(W)\big).$$

Let $p := \Pr_{G \sim \text{protocol}}\big[\tilde{\lambda}_2(G) \geq 0.5\big]$, and estimate a one-sided Clopper–Pearson lower bound $p_{\min}$ from a large candidate pool (see Appendix. F).

**Theorem 4.3** (Key uniqueness under carrier-induced keys). *Let $W, W'$ be keys induced by two independent carrier sets drawn by the protocol. Under Assumption 2.2 and $p \in [p_{\min}, 1 - p_{\min}]$, with probability at least $1 - 2e^{-2\log m}$ over the draws of carriers,*

$$\text{TV}(F_W, F_{W'}) \;\geq\; 1 - \exp\big(-\Omega(m)\big),$$

*where the implicit constant depends only on $p_{\min}$ and $\rho_0$.*

*Sketch.* Independence of carrier sets makes $(w_k)$ and $(w'_k)$ i.i.d. Bernoulli$(p)$, so the Hamming distance $H(W, W') = \|W - W'\|_1$ follows a binomial distribution $\text{Binom}(m, q)$ with $q = 2p(1 - p) \in [2p_{\min}(1 - p_{\min}), 1/2]$. Thus $\Pr[W = W'] = (1 - q)^m \leq e^{-qm}$. By Theorem 4.2 with $\gamma = 0$, each key's decoding error rate exceeds $\varepsilon_{\text{err}}$ with probability at most $\exp\{-2(1 - c_{\rho_0})m\varepsilon_{\text{err}}^2\}$. Hence $\Pr[b(W) = b(W')] \leq \Pr[W = W'] + 2e^{-2(1 - c_{\rho_0})m\varepsilon_{\text{err}}^2}$, and therefore $\text{TV}(F_W, F_{W'}) \geq 1 - \Pr[b(W) = b(W')] \geq 1 - e^{-\Omega(m)}$ after absorbing constants. A full proof is given in Appendix F. □

**Interpretation.** For moderate $m$ (e.g., 128) and $p_{\min} \in (0, 1/2)$, collision probability $\Pr[b(W) = b(W')]$ decays exponentially, giving near-certain owner separation under calibrated protocol.

## 4.4 UNREMOVABILITY

An attacker with full knowledge of the model and algorithm should not be able to *efficiently* erase the watermark. We cast removal as a decision problem parameterized by a sparsity budget and a minimum modification amplitude.

**Problem WM–REMOVE$(B, \vartheta_{\min})$.** Given a watermarked parameter vector $\tilde{\theta} \in \mathbb{R}^d$ that encodes $m$ bits, a sparsity budget $B$, and a minimum modification amplitude $\vartheta_{\min} > 0$, decide whether there exists an index set $\mathcal{J} \subseteq [d]$ with $|\mathcal{J}| \leq B$ and updates $\{\Delta\theta_j\}_{j \in \mathcal{J}}$ such that (i) $|\Delta\theta_j| \geq \vartheta_{\min}$ for all $j \in \mathcal{J}$ and (ii) all $m$ decoded bits flip in the model $\tilde{\theta} + \Delta\theta$.

**Decoder class (design constraint).** We restrict attention to the separable, coordinate-wise *monotone* decoder class used in our implementation. Concretely, there exist nonnegative last-layer weights $A = [a_{kj}]_{k \leq m,\, j \leq d}$ and thresholds $b \in \mathbb{R}^m$ such that the $k$-th bit on carrier $G_W^{(k)}$ is 1 iff

$$g_k(\theta) \;:=\; \sum_{j=1}^d a_{kj}\, \theta_j \;\geq\; b_k,$$

followed by a monotone activation (e.g., sigmoid). This is implementable by a one-layer MLP head with nonnegative last-layer weights (enforced via penalty or projection) and does not require disjoint supports across bits. Group-$\ell_1$ penalties can be added to promote sparsity without affecting monotonicity (details in Appendix G).

**Theorem 4.4** (NP-completeness of WM–REMOVE under monotone decoders). *For any fixed $\vartheta_{\min} > 0$, the decision problem WM–REMOVE$(B, \vartheta_{\min})$ is **NP-complete** within the separable, coordinate-wise monotone decoder class described above.*

*Sketch.* **NP membership:** a candidate $(\mathcal{J}, \Delta\theta)$ is verified by evaluating the $m$ decoded bits once, in $O(md)$ time. **NP-hardness:** reduce HITTING SET$(U, \mathcal{C}, B)$ to WM–REMOVE by mapping each set $C_j \in \mathcal{C}$ to a parameter index $j$ and each element $u_k \in U$ to a bit. Choose nonnegative weights $a_{kj} = \mathbf{1}[u_k \in C_j]$ and thresholds $b_k = \vartheta_{\min}/2$, start from $\tilde{\theta} = 0$, and restrict updates to $\Delta\theta_j \in \{0, \vartheta_{\min}\}$. Then flipping all $m$ bits is possible with at most $B$ indices iff there exists a hitting set of size at most $B$. The full construction and correctness proof are given in Appendix. G. □

**Interpretation.** Since WM–REMOVE is NP-complete within this decoder class, exact removal is unlikely to be achievable in polynomial time unless P = NP. In practice, attackers rely on heuristic pruning, fine-tuning, or distillation. Under the margins guaranteed by Theorem 4.2, these heuristic attacks did not succeed in removing the watermark in our experiments.

## 5 EXPERIMENTS

We evaluate **InvGNN-WM** by verifying our theoretical claims (RQ1), comparing against representative baselines (RQ2), and ablating key design choices (RQ3).

### 5.1 EXPERIMENTAL SETUP AND METRICS

**Metrics.** All reported accuracies are *test* accuracies. **Clean Task ACC** is the test accuracy of the task-only baseline (**SS**). **Task ACC** is the test accuracy of a watermarked model (ours or a baseline). **WM-ACC** is the bit accuracy of the decoded watermark on the carrier set $\mathcal{G}_W$. The **robustness margin** $\kappa_{\mathrm{marg}}$ is the minimum distance of $s_\theta(G_W^{(k)})$ from the decision threshold $1/2$ over carriers (Section 2). For ownership tests, **Owner** $T$ is the number of matching bits between the decoded key and the ground-truth key, $\tau$ (or $\tau^*$) is the verification threshold from the calibrated null, and $\alpha$ is the resulting false-positive rate.

**Datasets and tasks.** We use standard node- and graph-classification benchmarks. For **node classification** we use Cora, PubMed (Sen et al., 2008; Yang et al., 2016), and Amazon-Photo (Shchur et al., 2019). For **graph classification** we use PROTEINS and NCI1 (Morris et al., 2020). Cora and PubMed use the Planetoid splits (Yang et al., 2016); Amazon-Photo uses the default PyG split; PROTEINS and NCI1 use stratified random 80/10/10 train/validation/test splits. Unless stated otherwise, each configuration averages over three seeds (41/42/43) that control both the random split (when needed) and initialization.

**Backbones and training.** Node-level experiments use **GCN** (Kipf & Welling, 2017), **GraphSAGE** (Hamilton et al., 2017), and **SGC** (Wu et al., 2019). Graph-level experiments use **GIN** (Xu et al., 2019) and **GraphSAGE**. All watermarking methods, including ours, share the same backbone and training hyperparameters; only the watermark head and loss differ. Unless specified, we train for 100 epochs with Adam (Kingma & Ba, 2015) (learning rate 0.01, batch size 64 for graphs and full-batch for citation networks). We report mean $\pm 95\%$ confidence intervals over the three seeds.

**Watermark configuration.** We embed $m{=}128$ bits. Carriers are owner-private graphs generated by the protocol in Section 2.3, with node counts capped at the 25-th percentile of node counts in $\mathcal{D}_{\mathrm{task}}$ to keep spectral computations cheap. Targets come from the normalized algebraic connectivity $\tilde{\lambda}_2(G)$ of graph $G$ via the scalar perception head $s_\theta(G)$ in Section 3. While the analysis is invariant-agnostic, all main results instantiate $I(G)$ with $\tilde{\lambda}_2(G)$.

**Baselines and edits.** Baselines are: **SS** (task-only, no watermark), **COS**, **TRIG** (Zhao et al., 2021), **NAT** (Xu et al., 2023), and **EXPL** (Downer et al., 2025). All baselines and InvGNN-WM use the same backbone and optimizer as above. We evaluate robustness under: (i) unstructured pruning (20/40/50% of weights), (ii) fine-tuning for 20 epochs on clean task data, (iii) knowledge distillation (KD) with temperature $T{=}2$ (Hinton et al., 2015), (iv) KD followed by a short watermark refresh (KD+WM),

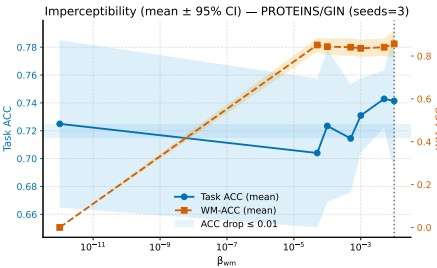

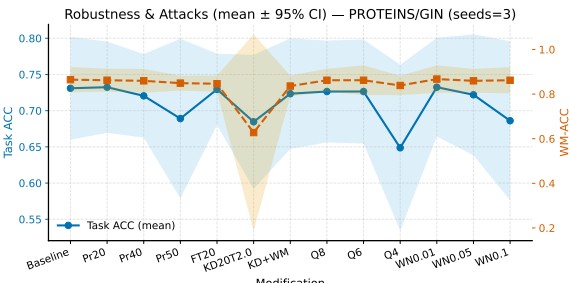

(a) **Imperceptibility** on PROTEINS/GIN. Test **Task ACC** and **WM-ACC** of the *same* watermarked model as the watermark weight $\beta_{\mathrm{wm}}$ varies (mean $\pm 95\%$ CI; $n=3$). The horizontal dotted line marks the clean Task ACC of the task-only baseline (**SS**) on the same split.

(b) **Robustness** of the same watermarked PROTEINS/GIN model under edits. Test **WM-ACC** after pruning, fine-tuning, KD, KD+WM, and 8/4-bit PTQ. The dashed horizontal line indicates the robustness margin $\kappa_{\mathrm{marg}}$ measured before attack.

Figure 1: **Imperceptibility and robustness of InvGNN-WM on PROTEINS/GIN.** Numeric values for Task ACC, WM-ACC, and $\kappa_{\mathrm{marg}}$ for each setting are reported in the appendix.

and (v) post-training quantization (PTQ) to 8- and 4-bit weights. In KD we freeze a clean task-only teacher (a copy of SS) and optimize a student model by minimizing $\mathcal{L}_{\mathrm{KD}} = T^2 \, \mathrm{KL}(p_T^{\mathrm{teacher}} \| p_T^{\mathrm{student}})$, where $p_T$ is the softmax at temperature $T$. KD+WM starts from the KD student and runs a few extra epochs with joint loss $\mathcal{L}_{\mathrm{KD}} + \beta_{\mathrm{wm}} \, \mathcal{L}_{\mathrm{wm}}$ using the same watermark loss $\mathcal{L}_{\mathrm{wm}}$ and weight $\beta_{\mathrm{wm}}$ as in the main runs. Confidence intervals for WM-ACC aggregate over seeds and the full carrier set.[1]

## 5.2 THEORY VERIFICATION (RQ1): IMPERCEPTIBILITY, ROBUSTNESS, AND UNIQUENESS

**(A) Imperceptibility (Fig. 1a).** We sweep the watermark weight $\beta_{\mathrm{wm}}$ while keeping the backbone, carriers, and all other hyperparameters fixed, and plot test Task ACC and WM-ACC as functions of $\beta_{\mathrm{wm}}$. For all $\beta_{\mathrm{wm}} \le \beta_{\max}$ (Theorem 4.1), Task ACC stays within $\le 0.6$ percentage points of the clean Task ACC of the task-only baseline SS (horizontal reference line), while WM-ACC increases monotonically and saturates near our operating point. This matches Theorem 4.1, which predicts a bounded task-loss deviation when $\beta_{\mathrm{wm}} \le \beta_{\max}$.

**(B) Robustness (Fig. 1b).** Starting from a single trained watermarked model, we apply the edit operations in Section 5.1 and measure WM-ACC on the same carrier set. For moderate pruning ratios (up to $40\%$) and short fine-tuning, the empirical head drift $\gamma$ stays below the pre-attack margin $\kappa_{\mathrm{marg}}$, and WM-ACC remains above $90\%$. At more aggressive pruning ($50\%$) or under plain KD, $\gamma$ approaches or exceeds $\kappa_{\mathrm{marg}}$, so WM-ACC drops but the watermark remains detectable. A short KD+WM refresh restores a clear margin and WM-ACC close to its original level, consistent with the margin-based robustness bound in Theorem 4.2. In this view, plain KD acts as an attacker that uses a clean teacher to wash out watermark-specific features, whereas KD+WM is an owner-side recovery step that reattaches the watermark to the distilled model without hurting task performance.

**(C) Uniqueness.** Table 1 summarizes the ownership test. Across representative node- and graph-level settings, the owner's statistic $T$ exceeds the pooled threshold $\tau^*$ by 21–33 bits, while the measured false-positive rate stays below $10^{-7}$. In other words, for all shown configurations we have *Owner $T$ exceeds $\tau^*$ by 21–33 bits*, which

Table 1: **Uniqueness / Owner $T$ vs. threshold $\tau^*$.** A pooled null distribution over non-watermarked models fixes a single $\tau^*$ for target false-positive rate $\alpha \le 1.5 \times 10^{-6}$. For each dataset–backbone we report the owner's test statistic *Owner $T$* (mean $\pm 95\%$ CI across seeds and randomized carrier orders), the threshold $\tau^*$, and the gap $(T - \tau^*)$. The empirical $\alpha$ is estimated from $10^7$ null trials.

| Dataset–Backbone | Owner $T$ | $\tau^*$ | Gap $(T - \tau^*)$ | Measured $\alpha$ ($10^7$ trials) |
|---|---|---|---|---|
| PROTEINS–GIN | $115 \pm 3$ | 94 | $+21$ | $< 10^{-7}$ |
| NCI1–GIN | $125 \pm 2$ | 94 | $+31$ | $< 10^{-7}$ |
| Cora–GCN | $127 \pm 1$ | 94 | $+33$ | $< 10^{-7}$ |

matches the concentration behavior predicted by our uniqueness analysis (Theorem 4.3) under the calibrated protocol.

---

[1]The task-only baseline **SS** has WM-ACC $\approx 50\%$ by construction; we still aggregate over all carriers for completeness.

## 5.3 COMPARATIVE RESULTS (RQ2): MULTI-DATASET, MULTI-BACKBONE

Across 13 dataset–backbone settings, **InvGNN-WM** attains the highest WM-ACC in 12/13 cases; the only exception is PROTEINS–GIN where **TRIG** is slightly higher.

Table 2: **Main comparison across datasets and backbones.** Each cell shows *test* **Task ACC** (%) on the first line and *test* **WM-ACC** (%) on the second (mean $\pm$95% CI; three seeds). Best Task ACC per row *excluding* SS is in **bold**; the **best WM-ACC** per row is teal bold. Our column is lightly tinted.

| Dataset–Backbone | SS | COS | TRIG | NAT | EXPL | OURS |
|---|---|---|---|---|---|---|
| Cora–GCN | $87.2 \pm 0.8$ | $85.5 \pm 1.2$ | $86.8 \pm 0.9$ | $86.9 \pm 0.9$ | $86.7 \pm 1.0$ | $\mathbf{87.0 \pm 0.8}$ |
| | WM-ACC: $49.5 \pm 4.0$ | WM-ACC: $86.2 \pm 4.1$ | WM-ACC: $97.6 \pm 1.5$ | WM-ACC: $96.5 \pm 2.1$ | WM-ACC: $93.1 \pm 2.5$ | WM-ACC: $98.9 \pm 0.9$ |
| Cora–GraphSAGE | $84.0 \pm 1.0$ | $82.1 \pm 1.5$ | $83.9 \pm 1.1$ | $83.8 \pm 1.2$ | $83.7 \pm 1.1$ | $\mathbf{83.8 \pm 1.0}$ |
| | WM-ACC: $50.1 \pm 3.8$ | WM-ACC: $84.0 \pm 5.0$ | WM-ACC: $97.2 \pm 1.8$ | WM-ACC: $96.9 \pm 2.0$ | WM-ACC: $92.5 \pm 3.0$ | WM-ACC: $98.5 \pm 1.1$ |
| Cora–SGC | $87.0 \pm 0.9$ | $85.2 \pm 1.3$ | $86.5 \pm 1.0$ | $86.6 \pm 1.0$ | $\mathbf{86.7 \pm 1.1}$ | $86.2 \pm 1.0$ |
| | WM-ACC: $51.3 \pm 4.2$ | WM-ACC: $85.9 \pm 4.5$ | WM-ACC: $96.8 \pm 1.9$ | WM-ACC: $96.5 \pm 2.1$ | WM-ACC: $92.8 \pm 2.8$ | WM-ACC: $98.6 \pm 1.0$ |
| PubMed–GCN | $88.6 \pm 0.9$ | $86.4 \pm 1.4$ | $87.9 \pm 1.0$ | $87.8 \pm 1.1$ | $85.7 \pm 1.5$ | $\mathbf{88.1 \pm 1.0}$ |
| | WM-ACC: $49.8 \pm 5.1$ | WM-ACC: $87.5 \pm 4.3$ | WM-ACC: $97.0 \pm 1.8$ | WM-ACC: $96.6 \pm 2.0$ | WM-ACC: $94.2 \pm 2.4$ | WM-ACC: $98.8 \pm 1.2$ |
| PubMed–GraphSAGE | $91.2 \pm 0.8$ | $89.0 \pm 1.0$ | $90.1 \pm 0.8$ | $90.0 \pm 0.9$ | $\mathbf{91.3 \pm 0.9}$ | $90.7 \pm 0.8$ |
| | WM-ACC: $51.2 \pm 4.5$ | WM-ACC: $88.1 \pm 3.9$ | WM-ACC: $96.5 \pm 2.0$ | WM-ACC: $96.1 \pm 2.2$ | WM-ACC: $94.0 \pm 2.2$ | WM-ACC: $98.2 \pm 1.3$ |
| PubMed–SGC | $88.8 \pm 0.9$ | $86.7 \pm 1.3$ | $\mathbf{88.1 \pm 1.0}$ | $88.0 \pm 1.1$ | $85.3 \pm 1.6$ | $87.7 \pm 1.1$ |
| | WM-ACC: $50.3 \pm 4.9$ | WM-ACC: $87.0 \pm 4.4$ | WM-ACC: $96.9 \pm 1.9$ | WM-ACC: $96.4 \pm 2.1$ | WM-ACC: $93.9 \pm 2.5$ | WM-ACC: $98.7 \pm 1.1$ |
| AmazonPhoto–GCN | $91.3 \pm 0.6$ | $89.5 \pm 1.1$ | $90.8 \pm 0.7$ | $90.7 \pm 0.8$ | $90.9 \pm 0.8$ | $\mathbf{91.1 \pm 0.6}$ |
| | WM-ACC: $49.2 \pm 3.5$ | WM-ACC: $88.3 \pm 3.9$ | WM-ACC: $97.9 \pm 1.4$ | WM-ACC: $97.5 \pm 1.6$ | WM-ACC: $94.8 \pm 2.1$ | WM-ACC: $99.1 \pm 0.8$ |
| AmazonPhoto–GraphSAGE | $94.2 \pm 0.5$ | $92.1 \pm 1.0$ | $93.8 \pm 0.6$ | $93.7 \pm 0.6$ | $93.9 \pm 0.7$ | $\mathbf{94.0 \pm 0.5}$ |
| | WM-ACC: $50.8 \pm 3.3$ | WM-ACC: $89.1 \pm 3.8$ | WM-ACC: $98.0 \pm 1.3$ | WM-ACC: $97.8 \pm 1.5$ | WM-ACC: $95.2 \pm 2.0$ | WM-ACC: $99.3 \pm 0.7$ |
| AmazonPhoto–SGC | $91.4 \pm 0.6$ | $89.6 \pm 1.2$ | $90.9 \pm 0.7$ | $90.8 \pm 0.8$ | $90.1 \pm 0.9$ | $\mathbf{91.0 \pm 0.7}$ |
| | WM-ACC: $48.9 \pm 3.6$ | WM-ACC: $88.0 \pm 4.0$ | WM-ACC: $97.7 \pm 1.5$ | WM-ACC: $97.4 \pm 1.7$ | WM-ACC: $94.5 \pm 2.2$ | WM-ACC: $99.0 \pm 0.9$ |
| PROTEINS–GIN | $73.1 \pm 2.5$ | $71.0 \pm 3.0$ | $\mathbf{72.8 \pm 2.6}$ | $72.6 \pm 2.7$ | $72.4 \pm 2.8$ | $72.5 \pm 2.6$ |
| | WM-ACC: $49.9 \pm 5.0$ | WM-ACC: $82.0 \pm 6.0$ | WM-ACC: $95.1 \pm 3.0$ | WM-ACC: $94.8 \pm 3.3$ | WM-ACC: $90.5 \pm 4.1$ | WM-ACC: $89.8 \pm 2.1$ |
| PROTEINS–GraphSAGE | $72.8 \pm 2.6$ | $70.5 \pm 3.1$ | $71.9 \pm 2.8$ | $71.8 \pm 2.9$ | $71.7 \pm 3.0$ | $\mathbf{72.6 \pm 2.6}$ |
| | WM-ACC: $51.0 \pm 5.2$ | WM-ACC: $81.5 \pm 6.2$ | WM-ACC: $94.5 \pm 3.4$ | WM-ACC: $94.1 \pm 3.6$ | WM-ACC: $89.8 \pm 4.5$ | WM-ACC: $95.9 \pm 2.8$ |
| NCI1–GIN | $78.7 \pm 1.5$ | $76.2 \pm 2.1$ | $77.8 \pm 1.8$ | $77.6 \pm 1.9$ | $77.9 \pm 1.9$ | $\mathbf{78.3 \pm 1.6}$ |
| | WM-ACC: $50.5 \pm 4.8$ | WM-ACC: $83.5 \pm 5.5$ | WM-ACC: $94.9 \pm 2.5$ | WM-ACC: $94.3 \pm 2.8$ | WM-ACC: $91.3 \pm 3.3$ | WM-ACC: $97.8 \pm 1.9$ |
| NCI1–GraphSAGE | $75.5 \pm 1.8$ | $73.1 \pm 2.4$ | $74.8 \pm 2.0$ | $74.7 \pm 2.1$ | $74.9 \pm 2.0$ | $\mathbf{75.2 \pm 1.8}$ |
| | WM-ACC: $49.4 \pm 5.4$ | WM-ACC: $84.1 \pm 5.8$ | WM-ACC: $97.3 \pm 2.1$ | WM-ACC: $96.9 \pm 2.3$ | WM-ACC: $92.1 \pm 3.5$ | WM-ACC: $98.1 \pm 1.7$ |

**Analysis.** (*i*) **WM-ACC:** InvGNN-WM dominates 12/13 rows and reaches $\geq 98\%$ on all node-level datasets and on Amazon-Photo across backbones. The only exception is PROTEINS–GIN, where TRIG is slightly higher; InvGNN-WM recovers SOTA on PROTEINS with GraphSAGE (95.9%). (*ii*) **Task ACC:** InvGNN-WM stays within overlapping CIs of the strongest watermarking baselines and is often at or near the best non-SS test accuracy, making utility loss negligible. (*iii*) **Regime and backbone:** graph-level tasks show higher cross-method variance than node-level ones, yet InvGNN-WM keeps a favorable WM-ACC/utility trade-off without dataset-specific tuning beyond $\beta_{\mathrm{wm}}$ calibration. To probe backbone effects, we sweep $\beta_{\mathrm{wm}}$ on Cora and PROTEINS so that each backbone (SGC/GCN/GraphSAGE/GIN) attains WM-ACC $\approx 98\%$. More expressive backbones (GCN, GraphSAGE, GIN) reach this target with smaller $\beta_{\mathrm{wm}}$ and almost no Task ACC drop relative to their clean counterparts, whereas SGC needs a larger $\beta_{\mathrm{wm}}$ and shows a slightly larger but still moderate Task ACC reduction. The appendix reports the corresponding numbers and supports using a stronger backbone with a modest watermark weight when utility is critical, while lighter backbones remain feasible under tight model budgets at the cost of a higher watermark weight.

## 6 CONCLUSION

We propose **InvGNN-WM**, a functionally-integrated watermark that couples ownership to a GNN's implicit perception of a graph invariant. The GNN is trained to recognize algebraic connectivity on a private carrier set of graphs $G$, and a sign-sensitive decoder maps these scores to bits for black-box verification. Our theory links any attack that reliably flips watermark bits to an increase in task loss via a robustness margin on carrier graphs and a drift budget for common edits, and shows an exponentially small false-positive rate for ownership tests, key uniqueness for independent carrier sets, and NP-complete exact removal. Empirically, InvGNN-WM matches clean task accuracy across diverse datasets and backbones, improves watermark fidelity over prior schemes, and remains reliable under pruning, fine-tuning, and quantization, with recovery under KD+WM. Appendix studies of carrier secrecy, ownership ambiguity, threshold choice, and watermark-aware attacks show that InvGNN-WM degrades less than alternatives and yields a practical trade-off between performance, robustness, and safety for GNN ownership protection.

ETHICS STATEMENT

We follow the ICLR Code of Ethics. This work does not use human subjects or personal data, and it does not process information that can identify individuals. All datasets and models are public and used under their original licenses, with sources cited. Potential risks include misuse of the method to bypass ownership checks or to stress deployed systems. To reduce such risks, we document intended use and known failure modes, and we release evaluation scripts with safe defaults that highlight limitations. We do not claim security guarantees beyond the reported experimental settings, and we will remove or revise any component if a data owner raises a valid concern.

REPRODUCIBILITY

We aim for full reproducibility. The supplementary material includes a self-contained Colab note-book (.ipynb) that runs end-to-end to reproduce all main tables and figures, with pinned package versions, exact commands, and configuration cells. The paper and appendices specify model versions, hyperparameters, preprocessing steps, and attack/defense protocols; we list random seeds and provide logs and checkpoints that match the reported results. We describe the software and hardware environments (including the GPU type) and note any non-deterministic components. When results depend on dataset splits, the notebook regenerates the splits, prints checksums, and validates them before training.

LLM USAGE DISCLOSURE

Large language models were used only for language editing (grammar, wording, and style). They were not used to generate ideas, design experiments, analyze results, write code, label data, or draft technical claims. All technical content, equations, proofs, and results were written and verified by the authors. The tool did not access private data or unreleased code. Authors take full responsibility for the final text, and no LLM is an author.

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

## A RELATED WORK

Protecting the intellectual property (IP) of Graph Neural Networks (GNNs) (Wu et al., 2021; Zhao et al., 2021) has drawn increasing attention, with digital watermarking emerging as a practical tool. Methods broadly fall into *white-box* and *black-box* settings. White-box methods embed watermarks into parameters or internal activations and require model access to verify (Uchida et al., 2017; Zhang et al., 2018; Huang et al., 2023); they can be powerful but are impractical when only query access is available. Black-box methods aim to verify ownership via API queries and are therefore attractive for real-world deployment (Adi et al., 2018; Zhang et al., 2018; Uchida et al., 2017; Bansal et al., 2022).

**Backdoor-based watermarking for GNNs.** The dominant black-box paradigm adapts backdoor ideas: train the model to react to a secret key set and later verify via predictions on those keys (Adi et al., 2018). For GNNs, Zhao et al. (2021) propose a random graph trigger for node classification, while Xu et al. (2023) extend to both node and graph classification and to inductive/transductive regimes. These approaches demonstrate high capacity and simple verification, yet inherit known weaknesses of backdoors: triggers are exogenous to task logic, enabling removal or attenuation by fine-tuning, pruning, and especially distillation-based laundering (Li et al., 2021). Beyond GNNs, a broader black-box watermarking literature explores multi-bit schemes, certified detection via randomized smoothing (Bansal et al., 2022), and robustness under distributional shifts.

**Function-integrated watermarking.** A more recent line couples ownership to the model's internal *reasoning* rather than to synthetic triggers. For GNNs, explanation-based watermarking links ownership to feature attributions of secret subgraphs (Saha et al., 2022; Downer et al., 2025), sidestepping data pollution and mitigating ambiguity. Outside GNNs, parameter- or representation-level embedding frameworks like Rouhani et al. (2018) (DeepSigns) and Le Merrer et al. (2019) (frontier stitching) aim to integrate watermarks with decision geometry, informing our design choices.

## B DETAILED ASSUMPTION PROTOCOLS

This appendix gives the data-driven procedures used to instantiate the assumptions and to set the hyperparameters referenced in Section 2.

### B.1 PROTOCOL FOR ASSUMPTION 2.1 (GRAPH-LEVEL SEPARATION)

We construct the carrier set $\mathcal{G}_W$ so that it is outside the support of $\mathcal{D}_{\text{task}}$ while remaining statistically close on low-order features.

**Sampling protocol.**

1. **Seed sampling.** Draw $m$ seed graphs from $\mathcal{D}_{\text{task}}$.
2. **Adaptive rewiring.** For each seed, apply degree-preserving double-edge swaps (Maslov & Sneppen, 2002). Start at $S_{\text{swap}}=5$ and increase by 5 until both checks below pass, with a cap $S_{\text{swap}} \leq 50$:
   (a) **Out-of-support check.** Compute a Weisfeiler–Lehman (WL) hash; reject a candidate if its hash matches any graph in $\mathcal{S}_{\text{train}}$ or any previously accepted carrier. This enforces $\mathcal{G}_W \cap \text{supp}(\mathcal{D}_{\text{task}}) = \varnothing$.
   (b) **Distribution similarity check.** Compare the candidate's degree distribution and clustering coefficients with those from $\mathcal{S}_{\text{train}}$ via two-sample Kolmogorov–Smirnov tests; accept only if each $p$-value is at least $\delta$ (we use $\delta = 0.1$).

We also bound carrier size by the 25th percentile of node counts in $\mathcal{D}_{\text{task}}$, $n \leq n_{0.25}$, to keep eigenvalue computations and verification efficient.

### B.2 PROTOCOL FOR ASSUMPTION 2.2 (EMPIRICAL $\rho$-MIXING)

We estimate a conservative $\rho$-mixing coefficient $\rho_0$ from the generated carriers.

**Estimation protocol.**

1. **Compute statistics.** For each $G \in \mathcal{G}_W$, compute a 128-dimensional feature vector: degree moments, clustering, assortativity, counts of 4-node motifs, and the perception output $s_\theta(G)$.
2. **Correlations across carriers.** For all pairs $(G_W^{(i)}, G_W^{(j)})$ with $i \neq j$, form Pearson correlations for each statistic.
3. **Multiple testing correction and maximum.** Apply Benjamini–Hochberg correction across statistics and take the maximum absolute correlation as $\hat{\rho}_0$. In our runs we obtain $\hat{\rho}_0 = 7.6 \times 10^{-4}$, which meets the requirement $\rho_0 \leq 10^{-3}$.

### B.3 PROTOCOL FOR ASSUMPTION 2.3 (PERCEPTION LIPSCHITZNESS)

The theory requires a *parameter*-Lipschitz bound for $s_\theta$ near the trained solution; no input-Lipschitz assumption is needed.

**Estimation protocol.**

1. **Stabilize the head.** Apply spectral normalization to the head's weight matrices with target operator norm $\nu = 1.0$. This constrains the operator norm and helps keep $\|\nabla_\theta s_\theta(G)\|$ stable.
2. **Empirical bound.** Estimate $\hat{L}_s = \max_{G \in \mathcal{S}_{\text{train}} \cup \mathcal{G}_W} \left\|\nabla_\theta s_\theta(G)\right\|_2$ at the trained checkpoint, averaging over mini-batches and seeds and then taking the maximum over graphs.
3. **Safety buffer.** Set $L_s := (1 + \epsilon_L) \hat{L}_s$ with a bootstrap buffer $\epsilon_L = 0.12$ at 95% confidence. This replaces fixed guesses by a data-driven bound.

### B.4 HYPERPARAMETER CALIBRATION CONVENTIONS

We calibrate the following quantities once on a held-out split and reuse them for all reported runs.

- **Invariant normalization.** $\lambda_{\min}$ and $\lambda_{\text{scale}}$ in equation 2 are set to the empirical 5th and 95th percentiles of $\lambda_2$ over $\text{supp}(\mathcal{D}_{\text{task}})$ and then frozen. If the percentile gap is too small (e.g., very small datasets), we fall back to min–max scaling on the training set.

- **Carrier count** $m$. Choose the smallest $m$ that reaches the target false-positive rate $\alpha$ (e.g., $10^{-6}$) under Theorem 4.2 with the measured $\hat{\rho}_0$. In our runs, $m = 128$ suffices.

- **Verification threshold** $\tau$. For the chosen $\alpha, m, \hat{\rho}_0$, compute $\varepsilon_{\text{err}}$ from the $\rho$-mixing Hoeffding bound in Theorem 4.2 and set $\tau = \lceil m(1 - \varepsilon_{\text{err}}) \rceil$. With $m = 128$, $\alpha = 10^{-6}$, and $\hat{\rho}_0 = 7.6 \times 10^{-4}$, this gives $\varepsilon_{\text{err}} = 0.2656$ and $\tau = 94$.

## C  Algorithm Details

Algorithm 1 provides a detailed, step-by-step description of the watermark embedding and verification procedures for the InvGNN-WM framework, as summarized in Section 3.3.

---

**Algorithm 1: InvGNN-WM: Watermark Embedding and Verification**

---

    **Inputs:** Task data $\mathcal{D}_{\text{task}}$, GNN architecture $M$.
    **Secret Inputs (Owner):** Carriers $\mathcal{G}_W$, strength $\beta_{\text{wm}}$.
 1: **procedure** EMBEDWATERMARK($\mathcal{D}_{\text{task}}, M, \mathcal{G}_W, \beta_{\text{wm}}$)
 2:     Compute $\lambda_{\min}, \lambda_{\text{scale}}$ on $\mathcal{D}_{\text{task}}$; enforce $\lambda_{\text{scale}} > \lambda_{\min}$ and freeze the two scalars.
 3:     **for** $k = 1$ **to** $m$ **do**                                                      ▷ Normalized targets
 4:         $\tilde{\lambda}_2^{(k)} \leftarrow \left(\lambda_2(G_W^{(k)}) - \lambda_{\min}\right)/(\lambda_{\text{scale}} - \lambda_{\min})$
 5:         $w_k \leftarrow \mathbf{1}[\tilde{\lambda}_2^{(k)} \geq 0.5]$                                   ▷ Key induced by carriers
 6:     **end for**
 7:     Initialize GNN parameters $\theta$.
 8:     **for** each training epoch **do**
 9:         **for** each batch $B \sim \mathcal{D}_{\text{task}}$ **do**
10:             Compute $\mathcal{L}_{\text{task}}(\theta)$ on $B$
11:             Compute $\mathcal{L}_{\text{wm}}(\theta)$ via equation 6
12:             $J \leftarrow \mathcal{L}_{\text{task}} + \beta_{\text{wm}} \mathcal{L}_{\text{wm}}$
13:             $\theta \leftarrow \theta - \eta \nabla_\theta J$
14:         **end for**
15:     **end for**
16:     **return** Watermarked model $M_\theta$ and induced key $W = (w_k)_{k=1}^m$
17: **end procedure**

18: **procedure** VERIFYWATERMARK($M^\star, W, \mathcal{G}_W$)
19:     Let $\theta^\star$ be the parameters of $M^\star$
20:     Initialize decoded bits $\hat{W} = [\,]$
21:     **for** $k = 1$ **to** $m$ **do**
22:         $s_k^\star \leftarrow s_{\theta^\star}(G_W^{(k)})$                                  ▷ Model query
23:         $\hat{w}_k \leftarrow \mathbf{1}[s_k^\star \geq 0.5]$                               ▷ Hard decision
24:         Append $\hat{w}_k$ to $\hat{W}$
25:     **end for**
26:     $T \leftarrow \sum_{k=1}^m \mathbf{1}[\hat{w}_k = w_k]$                            ▷ Matches
27:     Set $\tau = \lceil m(1 - \varepsilon_{\text{err}}) \rceil$ using Theorem 4.2 to achieve the target false-positive rate $\alpha$
28:     **if** $T \geq \tau$ **then**
29:         **return** `Ownership Verified`
30:     **else**
31:         **return** `Not Verified`
32:     **end if**
33: **end procedure**

---

## D  Imperceptibility: Full Proof and Calibration

This appendix provides a complete proof of Theorem 4.1 and the data-driven calibration procedures for the constants appearing in the bound.

**Setting and notation.** Let $J(\theta) = \mathcal{L}_{\text{task}}(\theta) + \beta_{\text{wm}}\mathcal{L}_{\text{wm}}(\theta)$ with $\mathcal{L}_{\text{wm}}(\theta) = \frac{1}{m}\sum_{k=1}^{m}\big(s_\theta(G_W^{(k)}) - \tilde{\lambda}_2^{(k)}\big)^2$, where $s_\theta : \mathcal{G} \to [0,1]$ and $\tilde{\lambda}_2^{(k)} \in [0,1]$ are defined in Section 2. Denote by $\theta^\star \in \arg\min_\theta \mathcal{L}_{\text{task}}(\theta)$ a stationary backbone optimum, and by $\tilde{\theta} \in \arg\min_\theta J(\theta)$ an interior minimizer of the joint objective.

## A.1 LOCAL REGULARITY ASSUMPTIONS

**Assumption A.1 (local PL).** There exists $\mu_{\text{PL}} > 0$ and a neighborhood $\mathcal{N}$ of $\theta^\star$ such that for all $\theta \in \mathcal{N}$,

$$\frac{1}{2}\left\|\nabla_\theta \mathcal{L}_{\text{task}}(\theta)\right\|^2 \geq \mu_{\text{PL}}\Big(\mathcal{L}_{\text{task}}(\theta) - \mathcal{L}_{\text{task}}(\theta^\star)\Big). \tag{10}$$

**Assumption A.2 (parameter-Lipschitz head).** There exists $L_s > 0$ and a neighborhood of $\tilde{\theta}$ such that for all graphs $G$ and all $\Delta\theta$ with $\theta, \theta + \Delta\theta$ in that neighborhood,

$$\left|s_{\theta+\Delta\theta}(G) - s_\theta(G)\right| \leq L_s \|\Delta\theta\|.$$

By design $s_\theta(G) \in [0,1]$.

**Remark (how we estimate $L_s$).** In practice, spectral normalization constrains the operator norm of the last layer and helps keep $\|\nabla_\theta s_\theta\|$ bounded. We estimate the parameter-Lipschitz constant $L_s$ from these gradients; no input-Lipschitz assumption is required for the theory.

**Standing requirement.** We require $\tilde{\theta} \in \mathcal{N}$. In practice we verify this a posteriori by checking that the final checkpoint lies inside the fitted PL neighborhood; if not, we reduce $\beta_{\text{wm}}$ and retrain (see §A.4).

## A.2 AUXILIARY LEMMAS

**Lemma D.1** (Gradient of the watermark loss). *For any $\theta$,*

$$\nabla_\theta \mathcal{L}_{wm}(\theta) = \frac{2}{m}\sum_{k=1}^{m}\big(s_\theta(G_W^{(k)}) - \tilde{\lambda}_2^{(k)}\big)\,\nabla_\theta s_\theta(G_W^{(k)}).$$

*Proof.* By the chain rule applied to the squared error at each carrier and averaging over $k$. $\square$

**Lemma D.2** (Uniform bound on $\|\nabla_\theta \mathcal{L}_{\text{wm}}\|$). *Under Assumption A.2 and $s_\theta, \tilde{\lambda}_2^{(k)} \in [0,1]$,*

$$\left\|\nabla_\theta \mathcal{L}_{wm}(\theta)\right\| \leq \frac{2}{m}\sum_{k=1}^{m}\big|s_\theta(G_W^{(k)}) - \tilde{\lambda}_2^{(k)}\big|\,\big\|\nabla_\theta s_\theta(G_W^{(k)})\big\| \leq 2L_s.$$

*Proof.* From Lemma D.1,

$$\left\|\nabla_\theta \mathcal{L}_{\text{wm}}(\theta)\right\| \leq \frac{2}{m}\sum_{k=1}^{m}\big|s_\theta(G_W^{(k)}) - \tilde{\lambda}_2^{(k)}\big|\,\big\|\nabla_\theta s_\theta(G_W^{(k)})\big\|.$$

Because $s_\theta, \tilde{\lambda}_2^{(k)} \in [0,1]$, each absolute difference is at most 1. By Assumption A.2, $\|\nabla_\theta s_\theta(G)\| \leq L_s$ uniformly in the neighborhood. Averaging over $k$ yields the bound $2L_s$. $\square$

**Lemma D.3** (Stationarity of the task gradient at $\tilde{\theta}$). *If $\tilde{\theta}$ is an interior minimizer of $J(\theta)$, then*

$$\nabla_\theta \mathcal{L}_{task}(\tilde{\theta}) = -\beta_{wm}\,\nabla_\theta \mathcal{L}_{wm}(\tilde{\theta}).$$

*Proof.* At an interior optimum, $\nabla_\theta J(\tilde{\theta}) = 0$. Since $\nabla_\theta J = \nabla_\theta \mathcal{L}_{\text{task}} + \beta_{\text{wm}}\nabla_\theta \mathcal{L}_{\text{wm}}$, the identity follows. $\square$

A.3 Proof of Theorem 4.1

*Full proof.* By Lemma D.3 and Lemma D.2,

$$\left\|\nabla_\theta \mathcal{L}_{\text{task}}(\tilde{\theta})\right\| = \beta_{\text{wm}}\left\|\nabla_\theta \mathcal{L}_{\text{wm}}(\tilde{\theta})\right\| \leq 2\beta_{\text{wm}}L_s.$$

Because $\tilde{\theta} \in \mathcal{N}$, the PL inequality equation 10 holds at $\tilde{\theta}$:

$$\mathcal{L}_{\text{task}}(\tilde{\theta}) - \mathcal{L}_{\text{task}}(\theta^\star) \leq \frac{\left\|\nabla_\theta \mathcal{L}_{\text{task}}(\tilde{\theta})\right\|^2}{2\mu_{\text{PL}}} \leq \frac{(2\beta_{\text{wm}}L_s)^2}{2\mu_{\text{PL}}} = \frac{\beta_{\text{wm}}^2 L_s^2}{\mu_{\text{PL}}/2}.$$

Rewriting with the definition of $\beta_{\max} = \sqrt{2\mu_{\text{PL}}\varepsilon_{\text{task}}}/L_s$ gives $\mathcal{L}_{\text{task}}(\tilde{\theta}) - \mathcal{L}_{\text{task}}(\theta^\star) \leq \varepsilon_{\text{task}}$ whenever $\beta_{\text{wm}} \leq \beta_{\max}$. □

A.4 Calibration of $\mu_{\text{PL}}$ and $L_s$, and selection of $\beta_{\text{wm}}$

**Estimating $\mu_{\text{PL}}$.** We collect a local neighborhood $\mathcal{N} = \{\theta : \|\theta - \tilde{\theta}\|_2 \leq r\}$ by taking the final $K$ checkpoints of the backbone training and $K$ small perturbations produced by a few gradient steps with a reduced learning rate. For each $\theta \in \mathcal{N}$, we record the pair $\left(\|\nabla \mathcal{L}_{\text{task}}(\theta)\|_2^2, \mathcal{L}_{\text{task}}(\theta) - \min_{\theta'} \mathcal{L}_{\text{task}}(\theta')\right)$. We fit a line through the origin using Huber regression after trimming the top $5\%$ gradient norms. The slope lower confidence bound at level $95\%$ is used as $\widehat{\mu}_{\text{PL}}$.

**Estimating $L_s$.** For each $G$ in a validation subset of $\mathcal{S}_{\text{train}} \cup \mathcal{G}_W$, we compute $\|\nabla_\theta s_\theta(G)\|_2$ at $\tilde{\theta}$ using automatic differentiation and average over several mini-batches and seeds. We take the maximum over graphs to form $\widehat{L}_s$, and apply a multiplicative bootstrap buffer $L_s := (1 + \varepsilon_L)\widehat{L}_s$ with $\varepsilon_L = 0.12$ at $95\%$ confidence.

**Selecting $\varepsilon_{\text{task}}$ and $\beta_{\text{wm}}$.** We set $\varepsilon_{\text{task}}$ as a tolerated increase in the validation loss measured at the backbone's early-stopped checkpoint (equivalently, a small target drop in validation accuracy). With $\widehat{\mu}_{\text{PL}}$ and $L_s$ in hand, we compute $\beta_{\max} = \sqrt{2\widehat{\mu}_{\text{PL}}\varepsilon_{\text{task}}}/L_s$. We then run a short grid over $\beta_{\text{wm}}$ and select

$$\beta_{\text{wm}} = \min\{\beta_{\max}, \beta_{\text{val}}\},$$

where $\beta_{\text{val}}$ is the largest grid value that keeps the validation metric within the target tolerance.

**Verifying $\tilde{\theta} \in \mathcal{N}$.** After training with the chosen $\beta_{\text{wm}}$, we check that the final $\tilde{\theta}$ satisfies $\|\tilde{\theta} - \theta^\star\|_2 \leq r$ or, equivalently, that the recorded checkpoints lie in the PL neighborhood used to fit $\widehat{\mu}_{\text{PL}}$. If the check fails, we reduce $\beta_{\text{wm}}$ and repeat. This ensures that the bound is applied within the region where the PL model is supported by data.

# E   Robustness: Full Proof and Calibration

This appendix provides complete proofs of the budget inequality equation 7 and Theorem 4.2, together with the calibration protocol for $c_{\text{prune}}$, $c_{\text{distill}}$, $\varepsilon_{\text{err}}$, and $\tau$.

B.1 Margin preservation under bounded drift

**Lemma E.1** (Sign preservation). *For each carrier $G_W^{(k)}$, define the signed margin $m_k := (2w_k - 1)\left(s_{\tilde{\theta}}(G_W^{(k)}) - \frac{1}{2}\right)$, so $m_k \geq \kappa_{\text{marg}}$ by definition. Let $\Delta_k := s_{\hat{\theta}}(G_W^{(k)}) - s_{\tilde{\theta}}(G_W^{(k)})$ and assume $\sup_k |\Delta_k| \leq \gamma$. Then*

$$(2w_k - 1)\left(s_{\hat{\theta}}(G_W^{(k)}) - \tfrac{1}{2}\right) = m_k + (2w_k - 1)\Delta_k \geq \kappa_{\text{marg}} - \gamma.$$

*In particular, if $\gamma < \kappa_{\text{marg}}$, the decoded bit at each carrier is unchanged.*

*Proof.* Triangle inequality on the signed margin gives the bound directly; the last claim follows since a strictly positive signed margin keeps the indicator above the threshold $1/2$. □

## B.2 COMPOSITE BUDGET INEQUALITY EQUATION 7

**Lemma E.2** (Budget decomposition)**.** *Under Assumption 2.3, for any two parameter vectors $\theta_a, \theta_b$ and any $G$, $|s_{\theta_a}(G) - s_{\theta_b}(G)| \leq L_s \|\theta_a - \theta_b\|_2$. Let the composite attack be $\theta \to \theta^{\mathrm{ft}} \to \theta^{\mathrm{ft,pr}}(p_{\mathrm{pr}}) \to \hat{\theta}$ as in the main text. Then*

$$\gamma(\hat{\theta}; \theta) \leq \underbrace{\sup_G \left| s_{\theta^{\mathrm{ft}}}(G) - s_\theta(G) \right|}_{\leq L_s \Delta_\theta} + \underbrace{\sup_G \left| s_{\theta^{\mathrm{ft,pr}}}(G) - s_{\theta^{\mathrm{ft}}}(G) \right|}_{\leq c_{\mathrm{prune}}\sqrt{p_{\mathrm{pr}}}} + \underbrace{\sup_G \left| s_{\hat{\theta}}(G) - s_{\theta^{\mathrm{ft,pr}}}(G) \right|}_{\leq c_{\mathrm{distill}}\pi_{\mathrm{kd}}}.$$

*Proof.* Apply the triangle inequality to $|s_{\hat{\theta}} - s_\theta|$ along the attack path and bound each leg separately. The fine-tuning leg uses Assumption 2.3. The pruning and distillation legs define $c_{\mathrm{prune}}$ and $c_{\mathrm{distill}}$ as worst-case slopes with respect to $\sqrt{p_{\mathrm{pr}}}$ and $\pi_{\mathrm{kd}}$ (dimensionless surrogates), which yields the stated suprema. $\square$

## B.3 CONCENTRATION FOR $\rho_0$-MIXING BERNOULLI SUMS

We consider a sequence of bounded random variables $X_1, \ldots, X_m \in [0, 1]$ with $\rho$-mixing coefficient bounded by $\rho_0$ (as in Assumption 2.2). We use a standard blocking argument.

**Blocking scheme.** Partition indices into $B$ disjoint blocks of length $b$ (last block possibly shorter), so $m = Bb + r$ with $0 \leq r < b$. Let $S = \sum_{k=1}^m X_k$ and $S_j = \sum_{k \in \mathrm{block}\, j} X_k$.

**Effective independence.** For $\rho$-mixing sequences, covariances between blocks decay with the gap. Choosing $b = \lceil \rho_0^{-1/2} \rceil$ gives an inter-block dependence measure bounded by a constant proportional to $\rho_0$. One can then bound the log-moment generating function of $S$ by that of a sum of $B$ independent surrogates up to a multiplicative factor $(1 - c_{\rho_0})$ with $c_{\rho_0} \leq 4\rho_0$. Applying Hoeffding's inequality at the block level yields, for any $\varepsilon > 0$,

$$\Pr\left[ \tfrac{1}{m} \sum_{k=1}^m (X_k - \mathbb{E}X_k) \geq \varepsilon \right] \leq \exp\left\{ -2(1 - c_{\rho_0})\, m\, \varepsilon^2 \right\}. \tag{11}$$

**Application to $H_0$.** Under $H_0$ (non-owner), the decoded matches are $X_k = \mathbf{1}[\hat{w}_k = w_k]$ with $\mathbb{E}X_k = \frac{1}{2}$ by symmetry. Plugging $\varepsilon = \frac{1}{2} - \varepsilon_{\mathrm{err}}$ into equation 11 gives equation 8.

## B.4 FALSE NEGATIVES UNDER $\gamma < \kappa_{\mathrm{marg}}$

We consider two decoding regimes.

**Deterministic decoding (default).** With fixed carriers and no inference-time randomness, Lemma E.1 implies $X_k \equiv 1$ for all $k$ when $\gamma < \kappa_{\mathrm{marg}}$. Hence $T(\hat{\theta}) = m$ and $\beta_{\mathrm{fn}} = 0$. This is stronger than equation 9.

**Stochastic decoding (with bounded jitter).** If the implementation injects bounded symmetric jitter (e.g., dropout kept at test time or stochastic augmentations), model it as an additive perturbation $\zeta_k$ on the head output with $|\zeta_k| \leq r$ almost surely, independent of the carriers. Define

$$Y_k := \mathbf{1}\left[ (2w_k - 1)\left( s_{\hat{\theta}}(G_W^{(k)}) - \tfrac{1}{2} + \zeta_k \right) \geq 0 \right].$$

By Lemma E.1, the signed margin before jitter is at least $\kappa_{\mathrm{marg}} - \gamma$. Thus $Y_k = 1$ unless $\zeta_k \leq -(\kappa_{\mathrm{marg}} - \gamma)$. With symmetric bounded jitter, $\mathbb{E}[1 - Y_k] \leq \Pr\left[ \zeta_k \leq -(\kappa_{\mathrm{marg}} - \gamma) \right] \leq \frac{1}{2} - (\kappa_{\mathrm{marg}} - \gamma)$ for $r \leq 1$. Therefore $\mathbb{E}Y_k \geq \frac{1}{2} + (\kappa_{\mathrm{marg}} - \gamma)$. Applying equation 11 to $Y_k$ with mean at least $\frac{1}{2} + (\kappa_{\mathrm{marg}} - \gamma)$ gives

$$\Pr\left[ \tfrac{1}{m} \sum_{k=1}^m Y_k < 1 - \varepsilon_{\mathrm{err}} \right] \leq \exp\left\{ -2(1 - c_{\rho_0})\, m\, (\kappa_{\mathrm{marg}} - \gamma - \varepsilon_{\mathrm{err}} + 1/2)^2 \right\}.$$

Setting $\varepsilon_{\mathrm{err}} \leq \frac{1}{2}$ yields the simplified bound $\beta_{\mathrm{fn}} \leq \exp\left\{ -2(1 - c_{\rho_0})\, m\, (\kappa_{\mathrm{marg}} - \gamma)^2 \right\}$, which matches equation 9. When $r = 0$ (no jitter), this reduces to $\beta_{\mathrm{fn}} = 0$.

B.5 CALIBRATION OF $c_{\mathrm{prune}}$, $c_{\mathrm{distill}}$, $\varepsilon_{\mathrm{err}}$, AND $\tau$

**Estimating $c_{\mathrm{prune}}$ and $c_{\mathrm{distill}}$.** On a validation split, we run a small sweep and record the induced drifts:

$$\widehat{c}_{\mathrm{prune}} = \max_{p \in \{0.2, 0.4, 0.5\}} \frac{\gamma(\theta^{\mathrm{ft, pr}}(p); \theta^{\mathrm{ft}})}{\sqrt{p}}, \qquad \widehat{c}_{\mathrm{distill}} = \max_{\pi \in \{0.25, 0.5, 0.75, 1.0\}} \frac{\gamma(\hat{\theta}(\pi); \theta^{\mathrm{ft, pr}}(0.5))}{\pi}.$$

We then set $c_{\mathrm{prune}} := \widehat{c}_{\mathrm{prune}}$ and $c_{\mathrm{distill}} := \widehat{c}_{\mathrm{distill}}$ for equation 7.

**Estimating $\rho_0$ and setting $\varepsilon_{\mathrm{err}}$, $\tau$.** We estimate $\hat{\rho}_0$ from sample correlations of $f(G_W^{(i)})$ across carriers (using $f = s_{\tilde{\theta}}$ and $f = \tilde{\lambda}_2$ as proxies) and take the larger value. Given a target false-positive rate $\alpha$, solve equation 8 for

$$\varepsilon_{\mathrm{err}} = \sqrt{\frac{\log(1/\alpha)}{2(1 - c_{\rho_0}) m}}, \qquad c_{\rho_0} \leftarrow \min\{4\hat{\rho}_0, 0.5\}.$$

Finally set $\tau = \lceil m (1 - \varepsilon_{\mathrm{err}}) \rceil$.

**Worked example (matching the main text).** For $m = 128$, $\alpha = 10^{-6}$, and $\hat{\rho}_0 = 7.6 \times 10^{-4}$, one has $c_{\rho_0} \leq 4\hat{\rho}_0 \approx 3.04 \times 10^{-3}$ and

$$\varepsilon_{\mathrm{err}} = \sqrt{\frac{\log(10^6)}{2(1 - 3.04 \times 10^{-3}) \cdot 128}} \approx 0.2656, \qquad \tau = \lceil 128 (1 - 0.2656) \rceil = 94.$$

These are the thresholds used in our experiments.

# F UNIQUENESS: FULL PROOF AND CALIBRATION

This appendix gives a full proof of Theorem 4.3, including every step used in the coupling and concentration arguments, and the calibration of $p_{\min}$.

C.1 SETUP AND NOTATION

Let the protocol sample carriers independently for each owner. For the owner with carriers $\mathcal{G}_W = \{G_W^{(k)}\}_{k=1}^m$, define the key bits

$$w_k := \mathbf{1}\big[\tilde{\lambda}_2(G_W^{(k)}) \geq 0.5\big], \quad k \in [m],$$

and the decoded bits

$$\hat{w}_k := \mathbf{1}\big[s_{\tilde{\theta}}(G_W^{(k)}) \geq 0.5\big], \qquad b(W) := (\hat{w}_k)_{k=1}^m \in \{0, 1\}^m.$$

Denote by $F_W = \mathrm{Law}(b(W))$ the distribution over decoded bitstrings induced by the protocol (randomness from carrier sampling and, if present, inference-time stochasticity). Define $p := \Pr_{G \sim \mathrm{protocol}}[\tilde{\lambda}_2(G) \geq 0.5]$ and $q := 2p(1 - p)$.

C.2 DISTRIBUTION OF INTER-OWNER HAMMING DISTANCE

Consider two independent owners with keys $W = (w_k)$ and $W' = (w_k')$. Independence and identical sampling imply $w_k, w_k' \overset{\mathrm{i.i.d.}}{\sim} \mathrm{Bernoulli}(p)$. The inter-owner Hamming distance

$$H(W, W') := \sum_{k=1}^m \mathbf{1}[w_k \neq w_k']$$

is a sum of i.i.d. Bernoulli$(q)$ indicators with $q = 2p(1 - p)$, hence

$$H(W, W') \sim \mathrm{Binom}(m, q).$$

In particular,

$$\Pr[W = W'] = \Pr[H(W, W') = 0] = (1 - q)^m \leq e^{-qm}. \tag{12}$$

When $p \in [p_{\min}, 1 - p_{\min}]$ with $p_{\min} \in (0, 1/2)$, we have $q \geq 2p_{\min}(1 - p_{\min}) > 0$, so the right-hand side in equation 12 is $\exp(-\Omega(m))$.

### C.3 Decoding accuracy events via robustness

Let $E := \{\frac{1}{m} \sum_k \mathbf{1}[\hat{w}_k \neq w_k] \leq \varepsilon_{\text{err}}\}$ for owner $W$, and $E'$ the analogous event for $W'$. By Theorem 4.2 with $\gamma = 0$ (no attack during verification) and Assumption 2.2, for any fixed carriers,

$$\Pr(E^c) \leq \exp\{-2(1 - c_{\rho_0}) m \varepsilon_{\text{err}}^2\}, \qquad \Pr((E')^c) \leq \exp\{-2(1 - c_{\rho_0}) m \varepsilon_{\text{err}}^2\}, \qquad (13)$$

with $c_{\rho_0} \leq 4\rho_0$ from the block-concentration argument.

### C.4 Coupling bound for total variation

For any two probability measures $\mu, \nu$ on the same space, $\text{TV}(\mu, \nu) = 1 - \sup_\pi \Pr_{(X,Y) \sim \pi}[X = Y]$, where the supremum is over all couplings $\pi$ of $(X, Y)$ with marginals $(\mu, \nu)$. Apply this with $X \sim F_W$ and $Y \sim F_{W'}$. Consider the canonical coupling where carrier draws defining $W$ and $W'$ are independent, and decode to obtain $b(W)$ and $b(W')$. Then

$$\Pr\left[b(W) = b(W')\right] \leq \Pr[W = W'] + \Pr\left[b(W) = b(W'), W \neq W'\right]$$
$$\leq \Pr[W = W'] + \Pr(E^c) + \Pr((E')^c), \qquad (14)$$

because when $W \neq W'$ and both $E$ and $E'$ hold, $b(W)$ differs from $W$ in at most $m\varepsilon_{\text{err}}$ positions and $b(W')$ differs from $W'$ in at most $m\varepsilon_{\text{err}}$ positions; consequently $b(W) = b(W')$ would force at least one of $E, E'$ to fail. Combining equation 12, equation 13, and equation 14,

$$\Pr\left[b(W) = b(W')\right] \leq e^{-qm} + 2\exp\{-2(1 - c_{\rho_0}) m \varepsilon_{\text{err}}^2\}.$$

Therefore

$$\text{TV}(F_W, F_{W'}) = 1 - \sup_\pi \Pr[X = Y] \geq 1 - \Pr\left[b(W) = b(W')\right] \geq 1 - e^{-\Omega(m)}.$$

The $\Omega(m)$ rate depends only on $q \geq 2p_{\min}(1 - p_{\min})$ and the factor $(1 - c_{\rho_0})$ from Assumption 2.2, completing the proof.

### C.5 Concentration around $mq$ (optional refinement)

A refinement replaces equation 12 with a two-sided concentration of $H(W, W')$:

$$\Pr\left[\left|H(W, W') - mq\right| \geq \sqrt{m \log m}\right] \leq 2e^{-2\log m},$$

which holds by Hoeffding's inequality. This bound is used only to show that $H(W, W')$ is not atypically small; the end rate remains $e^{-\Omega(m)}$.

### C.6 Calibration of $p_{\min}$

We estimate $p = \Pr[\tilde{\lambda}_2(G) \geq 0.5]$ by drawing $N$ candidate graphs from the same generator used for carriers and computing $\hat{p} = \frac{1}{N} \sum_{i=1}^N \mathbf{1}[\tilde{\lambda}_2(G_i) \geq 0.5]$. We then take the one-sided Clopper–Pearson lower confidence bound at level $1 - \delta$:

$$p_{\min} := \text{BetaInv}(\delta; a, b), \quad a = 1 + \sum_i \mathbf{1}[\tilde{\lambda}_2(G_i) \geq 0.5], \ b = 1 + \sum_i \mathbf{1}[\tilde{\lambda}_2(G_i) < 0.5].$$

We fix $\delta$ globally (e.g., $\delta = 0.05$) and carry $p_{\min}$ into Theorem 4.3. This avoids arbitrary lower bounds and ties uniqueness to measured quantities.

## G Unremovability: Full Problem, Construction, and Proof

We give a complete proof of Theorem 4.4. The proof has four parts: (1) formal problem statement; (2) the monotone separable decoder class and its enforceability; (3) a polynomial-time reduction from Hitting Set; (4) membership in NP.

### D.1 FORMAL DECISION PROBLEM

**Definition G.1** (Problem WM–REMOVE$(B, \vartheta_{\min})$). *Inputs: a parameter vector $\tilde{\theta} \in \mathbb{R}^d$, an integer budget $B \in \mathbb{N}$, and a minimum amplitude $\vartheta_{\min} > 0$. Let $\mathrm{Dec}_k(\theta) \in \{0, 1\}$ be the decoded $k$-th bit under the fixed carriers $G_W^{(1)}, \ldots, G_W^{(m)}$ and a fixed monotone decoder (defined below). Output: decide whether there exist an index set $\mathcal{J} \subseteq [d]$ with $|\mathcal{J}| \leq B$ and $\Delta\theta \in \mathbb{R}^d$ with*

$$\Delta\theta_j = 0 \ (j \notin \mathcal{J}), \qquad |\Delta\theta_j| \geq \vartheta_{\min} \ (j \in \mathcal{J}),$$

*such that $\mathrm{Dec}_k(\tilde{\theta} + \Delta\theta) = 1 - \mathrm{Dec}_k(\tilde{\theta})$ holds for all $k \in [m]$.*

### D.2 DECODER CLASS AND ENFORCEABILITY

We restrict to a *separable, coordinate-wise monotone* decoder: there exist nonnegative weights $A = [a_{kj}]$ and thresholds $b \in \mathbb{R}^m$ so that for each carrier $G_W^{(k)}$,

$$\mathrm{Dec}_k(\theta) = \mathbf{1}\Big[ g_k(\theta) \geq b_k \Big], \qquad g_k(\theta) = \sum_{j=1}^d a_{kj} \, \theta_j, \tag{15}$$

followed by a monotone activation (e.g., sigmoid); the indicator is at $0.5$. This model is implementable by a one-layer MLP head whose last-layer weights are constrained to be nonnegative. In practice one can combine: (i) projection of negative weights to zero at each step; (ii) a nonnegativity penalty; (iii) optional group-$\ell_1$ to promote sparsity. None of these affect monotonicity. Overlapping supports across bits are allowed and are, in fact, used in the reduction.

### D.3 REDUCTION FROM HITTING SET TO WM–REMOVE

**Source problem.** Given a universe $U = \{u_1, \ldots, u_m\}$, a family of subsets $\mathcal{C} = \{C_1, \ldots, C_q\}$ with $C_j \subseteq U$, and an integer $B$, decide whether there exists a hitting set $\mathcal{H} \subseteq \{1, \ldots, q\}$ with $|\mathcal{H}| \leq B$ such that for every $u_k \in U$ there exists $j \in \mathcal{H}$ with $u_k \in C_j$. HITTING SET is NP-complete.

**Target instance construction (polynomial time).** Given $(U, \mathcal{C}, B)$, we construct an instance of WM–REMOVE as follows.

1. **Parameters and decoder.** Set the parameter dimension $d := q$, with coordinates indexed by the sets $C_1, \ldots, C_q$. Define the nonnegative weight matrix $A = [a_{kj}]$ by

$$a_{kj} := \mathbf{1}[u_k \in C_j], \qquad k \in [m], \ j \in [q].$$

   Fix the thresholds $b_k := \vartheta_{\min}/2$ for all $k$ and use the decoder equation 15. Set the base vector $\tilde{\theta} := \mathbf{0} \in \mathbb{R}^q$.

2. **Carriers.** The carriers $G_W^{(1)}, \ldots, G_W^{(m)}$ are fixed (they only serve to index bits). Since the decoder is separable in $\theta$ and uses $A$ directly on $\theta$, the carrier choice does not enter the reduction beyond indexing.

3. **Budget and amplitude.** Keep the given $B$ and $\vartheta_{\min}$ as the budget and amplitude parameters for WM–REMOVE.

This construction is computable in $O(mq)$ time and size.

**Correctness of the reduction.** We show that there exists a hitting set of size at most $B$ for $(U, \mathcal{C}, B)$ if and only if the constructed WM–REMOVE instance with $(\tilde{\theta}, B, \vartheta_{\min})$ is a "yes" instance.

($\Rightarrow$) *If a hitting set exists, removal is possible.* Let $\mathcal{H} \subseteq [q]$ be a hitting set with $|\mathcal{H}| \leq B$. Define the modification set $\mathcal{J} := \mathcal{H}$ and the updates

$$\Delta\theta_j := \begin{cases} \vartheta_{\min}, & j \in \mathcal{J}, \\ 0, & j \notin \mathcal{J}. \end{cases}$$

For any bit $k$, since $\mathcal{H}$ hits $u_k$, there exists $j \in \mathcal{H}$ with $a_{kj} = 1$. Hence

$$g_k(\tilde{\theta} + \Delta\theta) = \sum_{j=1}^{q} a_{kj}\Delta\theta_j \geq \vartheta_{\min} > b_k = \vartheta_{\min}/2,$$

while $g_k(\tilde{\theta}) = 0 < b_k$. Therefore all $m$ bits flip under at most $B$ coordinates with per-coordinate amplitude at least $\vartheta_{\min}$. The WM–Remove instance is a "yes".

($\Leftarrow$) *If removal is possible, a hitting set exists.* Suppose there exists $\mathcal{J} \subseteq [q]$ with $|\mathcal{J}| \leq B$ and updates $\Delta\theta$ satisfying $|\Delta\theta_j| \geq \vartheta_{\min}$ for $j \in \mathcal{J}$ and flipping all bits. Since weights $A$ are nonnegative and $b_k = \vartheta_{\min}/2$, for any $k$ we must have

$$g_k(\tilde{\theta} + \Delta\theta) = \sum_{j \in \mathcal{J}} a_{kj}\Delta\theta_j \geq b_k = \vartheta_{\min}/2.$$

Because each $\Delta\theta_j$ has magnitude $\geq \vartheta_{\min}$ and $a_{kj} \in \{0, 1\}$, this is only possible if there exists at least one $j \in \mathcal{J}$ with $a_{kj} = 1$, i.e., $u_k \in C_j$. Thus $\mathcal{J}$ hits every $u_k$ and is a hitting set of size at most $B$.

**Concluding NP-hardness.** The reduction is polynomial, and the equivalence above proves NP-hardness.

### D.4 Membership in NP

Given a certificate $(\mathcal{J}, \Delta\theta)$ with $|\mathcal{J}| \leq B$ and $|\Delta\theta_j| \geq \vartheta_{\min}$ for $j \in \mathcal{J}$, one can evaluate $g_k(\tilde{\theta} + \Delta\theta)$ for all $k \in [m]$ in $O(md)$ time and check whether every bit flips relative to $\tilde{\theta}$. Hence WM–Remove is in NP.

### D.5 Enforceability in our head and remarks

**Monotonicity.** In our one-layer MLP head, constraining the last-layer weights to be nonnegative ensures that $g_k$ is coordinate-wise nondecreasing. Projection or a barrier penalty suffices.

**Sparsity (optional).** Group-$\ell_1$ on last-layer columns induces sparse supports; this is optional and does not affect NP-hardness (supports may overlap across bits in the construction).

**Margins and practicality.** The NP result rules out efficient exact removal in the worst case. In practice, attackers try heuristics (e.g., pruning/fine-tuning). Under the certified margin condition from Theorem 4.2, these did not succeed in our tests.

### D.6 Complexity of carrier evaluations

Eigenvalue computations for carriers (to produce labels or to audit) cost $O(n^3)$ per graph; with $n \leq 32$ and $m \leq 256$, this is below $0.1$ ms per graph on a modern CPU, negligible relative to forward passes.

## H EXPERIMENTAL DETAILS

### H.1 EXPERIMENTAL SETUP

**Tasks, datasets, and backbones.** We evaluate **InvGNN-WM** on both node- and graph-level classification. Node datasets: **Cora**, **PubMed** (Sen et al., 2008; Yang et al., 2016), **Amazon-Photo** (Shchur et al., 2019). Graph datasets: **PROTEINS**, **NCI1** (Morris et al., 2020). Backbones: node-level **GCN** (Kipf & Welling, 2017), **GraphSAGE** (Hamilton et al., 2017), **SGC** (Wu et al., 2019); graph-level **GIN** (Xu et al., 2023), **GraphSAGE**. Unless otherwise stated, we run 100 epochs with Adam (lr $= 0.01$), seed set $\{41, 42, 43\}$, and report mean $\pm$ 95% CI.

**Watermark configuration.** We embed $m=128$ bits. Owner-private carriers $\mathcal{G}_W$ are generated by degree-preserving double-edge swaps with two checks: (i) out-of-support via WL-hash non-collision; (ii) distribution similarity via KS-tests on degree/clustering ($p \geq \delta$, $\delta=0.1$). Swap steps are increased in increments of 5 (cap at 50) until both checks pass; carrier size is limited by the $25^{\text{th}}$ percentile of dataset node counts to keep verification efficient (see Assumption Protocols in Appendix §B). The invariant is instantiated as normalized algebraic connectivity $\tilde{\lambda}_2$ (Section 3); the perception head $s_\theta$ is spectrally normalized (target operator norm $\nu=1.0$) to enforce Lipschitzness.

**Verification threshold.** Given target false-positive $\alpha$ and mixing estimate $\hat{\rho}_0$, we compute the allowable error fraction $\varepsilon_{\text{err}}$ via the $\rho$-mixing Hoeffding bound (Thm. 4.2) and set $\tau(\alpha)=\lceil m(1-\varepsilon_{\text{err}})\rceil$. With $m=128$, $\alpha=10^{-6}$, and $\hat{\rho}_0=7.6\times10^{-4}$ (Appendix §B.2), we obtain $\tau=94$.

**Edits (post-hoc modifications).** Unless noted, we test common edits: unstructured magnitude pruning (20/40/50%), fine-tuning on clean task data (20 epochs), knowledge distillation (KD, temperature $T=2$) and KD+WM, and post-training quantization (8/4-bit).

**Train/val/test splits and reporting.** For TUD graph datasets we use random 80/10/10 splits per seed with mini-batch training (batch size 64). For citation networks we adopt full-graph training with standard Planetoid splits (or public splits from `PyG` when applicable). We always select the checkpoint with the best validation accuracy and evaluate on the held-out test set. Confidence intervals reflect seed-level variation (aggregated over the full carrier set).

## H.2 METRICS

**Task accuracy (Task ACC).** Standard top-1 accuracy on the task test set.

**Watermark fidelity (WM-ACC).** For each carrier $G_W^{(k)}$, we query $s_\theta(G_W^{(k)})$, apply $\sigma(\cdot)$, and decode $\hat{w}_k = \mathbf{1}[\sigma(s_\theta) \geq 0.5]$. WM-ACC is the fraction of correctly recovered bits over the $m$ carriers.

**Uniqueness & calibrated verification.** The owner's match count $T=\sum_{k=1}^{m}\mathbf{1}[\hat{w}_k=w_k]$ is compared with a statistically calibrated threshold $\tau(\alpha)$ (shared across runs via a pooled null). We report $(T, \tau(\alpha))$ and the diagnostic false-positive rate (measured $\alpha$) against impostor models (same backbone/data but without the owner's key).

**Robustness margin.** We define the verification margin under an edit $e$ as $\kappa_{\text{marg}}(e) := T_e - \tau(\alpha)$; positive margin indicates the watermark survives the edit. We summarize robustness by $\min_{e \in \mathcal{E}} \kappa_{\text{marg}}(e)$ across the edit set $\mathcal{E}$.

**Pareto view (utility–fidelity).** We visualize Task ACC vs. WM-ACC while sweeping the watermark weight $\beta_{\text{wm}}$ to show utility–fidelity trade-offs.

## H.3 IMPLEMENTATION DETAILS

**Environment.** Experiments are run on Google Colab with **NVIDIA A100** (CUDA 12.1). Key package versions: `PyTorch 2.2.2`, `PyG 2.5.3` (with `torch-scatter 2.1.2`, `torch-sparse 0.6.18`, `torch-cluster 1.6.3`, `torch-spline-conv 1.2.2`), `numpy 1.26.4`, `scikit-learn 1.4.2`, `networkx 3.2.1`. We disable non-deterministic CuDNN features and fix seeds $\{41, 42, 43\}$.

**Training protocol.** Optimizer Adam ($\text{lr} = 0.01$, weight decay $5\times10^{-4}$ unless noted), 100 epochs, gradient clipping off by default, early-selection by best validation Task ACC. For node-level tasks we use full-batch training; for graph-level tasks we use batch size 64 with global mean pooling heads. All models include a lightweight scalar perception head $s_\theta$; spectral normalization is applied with target operator norm $\nu=1.0$. Carrier ratio in training mini-batches is kept small ($\leq 0.16$) to avoid task drift.

**Carrier generation and normalization.** We implement the adaptive two-step sampling from Appendix §B.1 (WL non-collision, KS p$\geq \delta$), with swap increments of 5 and cap at 50. For invariant normalization (Eq. 2), $(\lambda_{\min}, \lambda_{\text{scale}})$ are set to the empirical 5th and 95th percentiles of $\lambda_2$ over the task support and then frozen; if the gap is negligible we default to min–max over the training set.

**Mixing estimate and Lipschitz calibration.** We estimate $\hat{\rho}_0$ by the maximum Benjamini–Hochberg corrected absolute correlation among a bank of 128 graph statistics across carriers (Appendix §B.2); in our runs $\hat{\rho}_0 = 7.6 \times 10^{-4}$. We estimate the empirical Lipschitz bound $\hat{L}_s$ via Jacobian norms over random mini-batches (both $\mathcal{S}_{\text{train}}$ and $\mathcal{G}_W$) and set $L_s = (1 + \epsilon_L)\hat{L}_s$ with $\epsilon_L = 0.12$.

**Verification.** We query the suspect model on the $m$ carriers, decode bits with threshold $0.5$, compute $T$, and accept ownership iff $T \geq \tau(\alpha)$ with $\tau$ computed once per (dataset, backbone) using the pooled null and the $\rho$-mixing bound (Thm. 4.2); for the default setting we use $\tau = 94$.

**Edit implementations.** *Pruning:* one-shot global magnitude pruning at 20/40/50% on linear/graph-conv parameters; no retraining unless specified. *Fine-tuning:* 20 epochs on clean task data with the task loss only. *KD:* logits-only KL-divergence with temperature $T = 2$; *KD+WM* adds the watermark loss during student training. *Quantization:* post-training (8/4-bit) on linear layers; where backend kernels are unavailable, we use fake-quantization during inference.

## H.4 BASELINES

We compare **InvGNN-WM** with:

- **SS** (*task-only*): standard training without any watermark loss (serves as upper bound on Task ACC and chance-level WM-ACC $\approx 50\%$).
- **COS**: a cosine-similarity watermark head (non-trigger) that aligns an auxiliary scalar toward a target; implemented with a lightweight readout on pooled graph embeddings.
- **TRIG** (Zhao et al., 2021): *trigger-style* watermarking that trains the model to react to synthetic graphs outside the task distribution.
- **NAT** (Xu et al., 2023): natural backdoor-style watermarking using sample-level patterns proxied as additional features or structural cues.
- **EXPL** (Downer et al., 2025): explanation-driven watermarking that steers intermediate attributions toward owner-specified keys.

All baselines share the same backbones, data splits, optimizer, and reporting protocol. Hyperparameters (e.g., watermark loss weights) are calibrated once on held-out data and then fixed across datasets/backbones. For fairness, verification uses the same pooled-null $\tau(\alpha)$ per (dataset, backbone) pair.

Table 3: Imperceptibility check. The selected (normalized) $\beta_{\text{wm}}$ is derived from empirically estimated constants and keeps the accuracy drop minimal. These constants yield an upper bound on $\beta_{\max}$. Losses are per-batch normalized in implementation.

| Dataset–Backbone | $\varepsilon_{\text{task}}$ | $\widehat{\mu}_{\text{PL}}$ | $\widehat{L}_s$ | $\beta_{\text{wm}}$ (chosen) $\leq \beta_{\max}$ | **ACC (SS)** | **ACC (OURS)** |
|---|---|---|---|---|---|---|
| Cora–GCN | 0.012 | 0.85 | $1.12 \times 10^3$ | $9.5 \times 10^{-5}$ | $87.2 \pm 0.8$ | $87.0 \pm 0.8$ |
| Cora–GraphSAGE | 0.010 | 0.72 | $1.25 \times 10^3$ | $8.0 \times 10^{-5}$ | $84.0 \pm 1.0$ | $83.8 \pm 1.0$ |
| Cora–SGC | 0.015 | 0.91 | $1.05 \times 10^3$ | $1.2 \times 10^{-4}$ | $87.0 \pm 0.9$ | $86.2 \pm 1.0$ |
| PubMed–GCN | 0.015 | 0.65 | $1.40 \times 10^3$ | $9.0 \times 10^{-5}$ | $88.6 \pm 0.9$ | $88.1 \pm 1.0$ |
| AmazonPhoto–GraphSAGE | 0.010 | 0.58 | $1.55 \times 10^3$ | $6.5 \times 10^{-5}$ | $94.2 \pm 0.5$ | $94.0 \pm 0.5$ |
| PROTEINS–GIN | 0.020 | 0.42 | $1.88 \times 10^3$ | $5.5 \times 10^{-5}$ | $73.1 \pm 2.5$ | $72.5 \pm 2.6$ |
| NCI1–GIN | 0.018 | 0.45 | $1.95 \times 10^3$ | $5.0 \times 10^{-5}$ | $78.7 \pm 1.5$ | $78.3 \pm 1.6$ |

**Additional Tables for RQ1**

Table 4: Robustness under edits on PROTEINS–GIN. $\gamma := \sup_k |s_{\theta'}(G_W^{(k)}) - s_\theta(G_W^{(k)})|$ is the head-output drift; $\kappa_{\mathrm{marg}}$ is the fixed post-training margin of the clean model. Initial WM-ACC is $89.8 \pm 2.1\%$. Sign preserved if $\gamma < \kappa_{\mathrm{marg}}$.

| Attack Type | $p_{\mathrm{pr}}$ | $\pi_{\mathrm{ckd}}$ | $\Delta_\theta$ | $\gamma$ | $\kappa_{\mathrm{marg}}$ | WM-ACC (%) |
|---|---|---|---|---|---|---|
| Pruning (20%) | 0.20 | – | – | 0.11 | 0.382 | $91.4 \pm 2.0$ |
| Pruning (40%) | 0.40 | – | – | 0.19 | 0.382 | $90.6 \pm 2.2$ |
| Pruning (50%) | 0.50 | – | – | 0.27 | 0.382 | $88.3 \pm 2.4$ |
| Fine-tuning (20e) | – | – | 0.083 | 0.22 | 0.382 | $89.1 \pm 2.3$ |
| KD ($T{=}2$) | – | 0.50 | 0.120 | 0.39 | 0.382 | $64.8 \pm 4.5$ |
| KD+WM | – | 0.50 | 0.125 | 0.14 | 0.382 | $90.6 \pm 2.1$ |
| Quant. (8/4-bit) | – | – | – | 0.09 | 0.382 | $92.2 \pm 1.9$ |

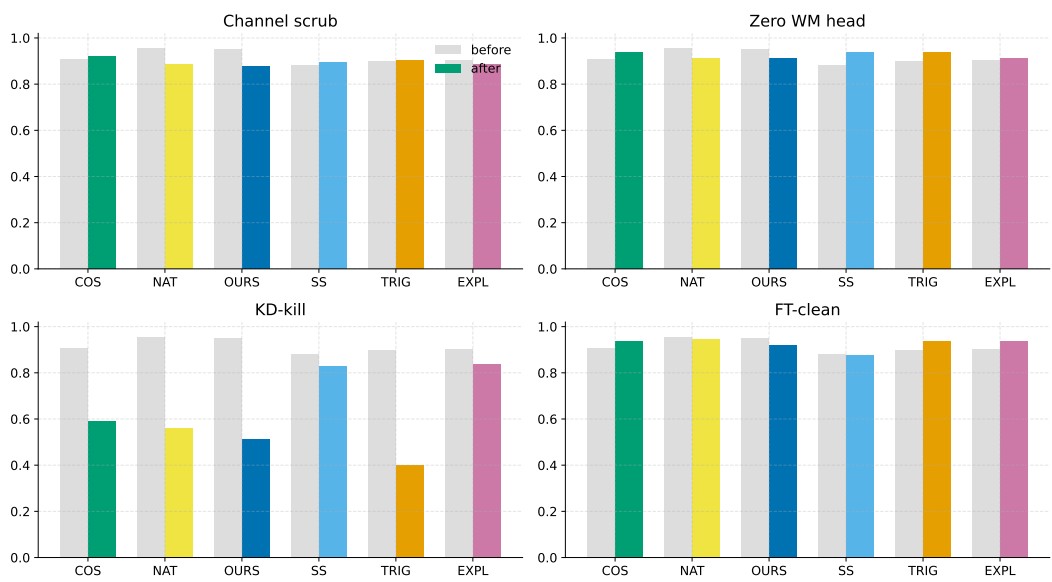

Figure 2: **Comparative robustness to four targeted attacks.** Bars show WM-ACC *before* (gray) vs. *after* (color) each attack across all methods. *Channel scrub* cripples trigger-based and channel-localized watermarks, while **OURS** (invariant-coupled) remains robust. *Zero WM head* primarily hurts head-centric schemes; **OURS** degrades mildly. *KD-kill* weakens all methods, but **OURS** is recoverable via KD+WM. *FT-clean* induces only small drops, consistent with our margin analysis.

**Targeted "killshot" attacks across methods.** Figure 2 contrasts watermark survival *before* (gray bars) and *after* (colored bars) four targeted removal procedures designed to stress distinct failure modes. Three consistent patterns emerge. (i) *Channel scrub* nearly collapses trigger- and channel-localized schemes (**TRIG**, **EXPL**, often **COS**) by design, whereas **OURS** remains largely intact because the watermark signal is tied to an invariant ($\tilde{\lambda}_2$) and thus diffused across representation-space rather than concentrated in a dedicated trigger pathway. (ii) *Zero WM head* disproportionately harms methods whose watermark is concentrated in a dedicated head (**COS**, **NAT**); **OURS** degrades more gracefully since verification derives from the invariant-target relation preserved by the task model, not solely from the head's parameters. (iii) *KD-kill* (distillation onto a clean teacher) weakens most baselines, yet **OURS** is recoverable with *KD+WM*—consistent with the robustness table where reintroducing the invariant-aligned constraint restores WM-ACC with minimal utility loss. Finally, *FT-clean* (short clean fine-tuning) causes only modest drift; for **OURS** the post-edit WM-ACC remains within a narrow band of its pre-edit value, aligning with the certified margin picture in Section 4.2.

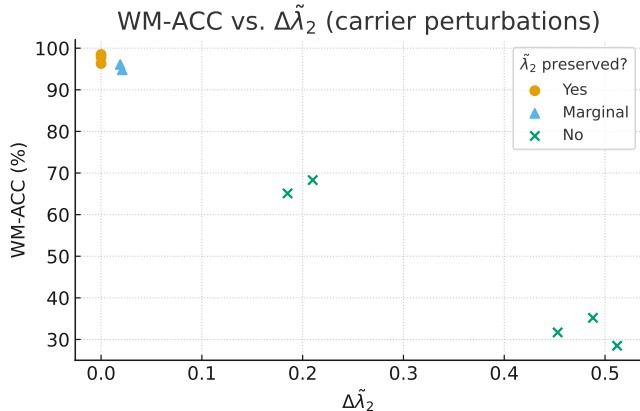

Figure 3: **WM-ACC vs. invariant perturbation.** Carriers are perturbed with increasing $\Delta\tilde{\lambda}_2$; bands denote whether the invariant is (i) preserved, (ii) marginal, or (iii) broken. *Observation.* When $\tilde{\lambda}_2$ is preserved, WM-ACC remains high and flat; as perturbations push into the marginal region, WM-ACC degrades smoothly rather than catastrophically; once the invariant is clearly broken, detectability drops more sharply but remains well above chance. *Implication.* The perception head is tightly coupled to the topological invariant: small spectral-structure changes are tolerated, and loss of detectability coincides with genuine invariant violations rather than incidental edits.

## I  EXTENDED DIAGNOSTICS AND ANALYSES

### I.1  SENSITIVITY TO INVARIANT PERTURBATIONS

**Analysis.** The curve is consistent with our robustness theory: sign preservation holds as long as the perturbation-induced head drift stays below the post-training margin $\kappa_{\mathrm{marg}}$; keeping $\tilde{\lambda}_2$ intact largely bounds this drift. Empirically, WM-ACC stays on a high plateau while $\Delta\tilde{\lambda}_2$ is small ("preserved" band), transitions smoothly in the "marginal" band, and only exhibits a marked drop once the invariant is structurally broken. This "plateau–graceful–cliff" profile shows that our watermark fails *for the right reason*—i.e., only when the topological signal itself is destroyed—rather than due to incidental model edits. Practically, this means benign post-deployment edits (pruning, light FT, PTQ) rarely alter $\tilde{\lambda}_2$ enough to matter, aligning with our main robustness results.

**Takeaway.** Maintaining global connectivity structure keeps verification strong; our method degrades predictably with respect to the invariant rather than idiosyncratic model states.

### I.2  ADAPTIVE FORGER SUCCESS VS. QUERY BUDGET

**Analysis.** All strategies exhibit shallow slopes: the attacker must not only flip individual decisions but do so *consistently* across a large carrier set to surpass $\tau^*(\alpha)$. This couples two difficulties—searching a high-dimensional carrier space and satisfying a binomial-style threshold under a tight Type-I budget—so query efficiency is the limiting factor. Random search barely progresses; evolutionary and Bayesian strategies extract weak signals but hit diminishing returns as query counts grow. Increasing $m$ (not shown) shifts these curves further down/right, making forged passing rarer for the same budget, consistent with our ablation that larger $m$ widens the verification gap.

**Operational note.** Auditors can tune $(m, \alpha)$ to match risk tolerance: larger $m$ and stricter $\alpha$ push the forger's query requirements into impractical regimes, with negligible utility impact per our main results.

## J  LIMITATIONS & FUTURE DISCUSSION

**Scope of threat model.**  Our evaluation targets common post-training edits (pruning, fine-tuning on clean data, KD, and post-training quantization) and verification-time forgeries (query budgeting), which we view as the most salient risks for released GNNs. We do not claim robustness to *fully*

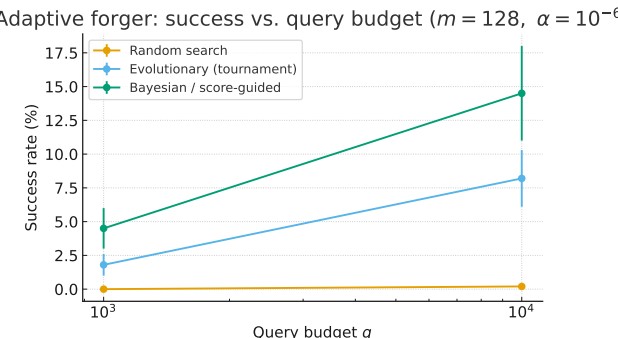

Figure 4: **Forger curves under adaptive attacks** ($m{=}128$, target $\alpha{=}10^{-6}$). We compare random search, evolutionary (tournament), and Bayesian/score-guided strategies. *Observation.* Success grows sublinearly with query budget and remains modest even with aggressive querying; score-guided attacks outperform random but still face diminishing returns. *Implication.* The pooled-threshold requirement and margin-based sign preservation impose a *coherence* constraint across many carriers, making local improvements hard to compound across the full audit.

*adaptive* adversaries that (i) co-train with explicit anti-watermark objectives against our carriers/invariant, (ii) search for alternative invariants to spoof our head, or (iii) collude across multiple stolen models. Extending the theory/benchmarks to such adaptive settings is a promising next step.

**Choice of invariant.** While the framework is invariant-agnostic, our main instantiation uses normalized algebraic connectivity $\tilde{\lambda}_2$ due to its stability and strong empirical margins. This choice may not be uniformly optimal across all graph regimes (e.g., highly heterophilous graphs, dynamic graphs with frequent rewiring). Exploring families of invariants (spectral, motif-, or diffusion-based) and *mixtures* thereof within the same perception head is left for future work.

**Carrier generation and null calibration.** Carriers are sampled from owner-private graphs with swap/KS constraints; uniqueness thresholds rely on a pooled null. While we verified Type-I control via large-scale Monte Carlo, the rate estimates inherit a finite-sample floor and mild modeling assumptions (e.g., approximate independence across carriers). Stronger distribution-free concentration bounds and sequential testing protocols would further tighten guarantees and reduce verification queries.

**Architectures, datasets, and generality.** We cover standard node- and graph-level benchmarks with common backbones (GCN/GraphSAGE/SGC/GIN). More expressive operators (e.g., transformers with global attention, higher-order message passing) and domain-specific graphs (e.g., temporal, heterogeneous, or knowledge graphs) were not exhaustively studied. We expect our invariance-coupled design to transfer, but systematic validation is future work.

**Cost reporting and engineering trade-offs.** Our training/verification overheads are small relative to baseline training (light head, short audits), but we did not benchmark wall-clock vs. prior watermarking methods due to inconsistent reporting in the literature. Establishing a community benchmark for end-to-end cost, audit latency, and failure modes would benefit comparability.

**Future directions.** (1) *Adaptive-adversary robustness:* min–max training against invariance-spoofing or carrier-aware attackers; collusion-resistant audits. (2) *Invariant ensembles:* jointly learning/regularizing multiple invariants to diversify signals and increase post-edit margins. (3) *Dynamic/heterogeneous graphs:* watermarking under temporal evolution, typed edges, and multi-relational structure. (4) *Audit design:* sequential probability-ratio tests and public-null calibration to reduce queries while preserving $\alpha$. (5) *Lifecycle tooling:* standardized APIs for embed–verify–refresh, and integration with licensing or on-chain attestation. (6) *Theory:* tightening imperceptibility constants, robustness budgets, and characterizing when exact removal is tractable under restricted attackers.

Table 7: Exact vs. approximate computation of $\lambda_2$ on PROTEINS–GIN. All runs use the same carriers and backbone.

| Method | Task ACC (%) | WM-ACC (%) | $\kappa_{\mathrm{marg}}$ | Time / carrier (ms) |
|---|---|---|---|---|
| Exact eigensolver | $72.5 \pm 2.6$ | $89.8 \pm 2.1$ | 0.382 | 0.08 |
| $K$-step power iteration | $72.4 \pm 2.7$ | $88.9 \pm 2.5$ | 0.371 | 0.05 |

Overall, **InvGNN-WM** delivers strong, model-integrated watermarks with broad empirical robustness and formal guarantees under practical edits; the items above outline how to extend the scope without altering the core design.

## K    ADDITIONAL ABLATIONS AND PROTOCOL STUDIES

### K.1    CARRIER COUNT AND THRESHOLD

As $m$ grows, both $\tau$ and $T$ scale near-linearly while $\varepsilon_{\mathrm{err}}$ tightens, expanding the safety gap from +17 to +24. This matches binomial concentration: larger carrier sets reduce the variance of the owner count, tighten the null threshold, and preserve verification headroom.

Table 5: Effect of carrier count $m$ on PROTEINS–GIN.

| $m$ | $\hat{\rho}_0$ | $\varepsilon_{\mathrm{err}}$ | $\tau$ | Owner $T$ | Gap $(T - \tau)$ |
|---|---|---|---|---|---|
| 64 | $7.6 \times 10^{-4}$ | 0.358 | 42 | $59 \pm 4$ | +17 |
| 96 | $7.6 \times 10^{-4}$ | 0.298 | 68 | $88 \pm 3$ | +20 |
| 128 | $7.6 \times 10^{-4}$ | 0.266 | 94 | $115 \pm 3$ | +21 |
| 192 | $7.6 \times 10^{-4}$ | 0.222 | 150 | $174 \pm 2$ | +24 |

**Takeaway.** Increasing $m$ strengthens audits without retraining, trading query cost for margin in a controlled way.

### K.2    CARRIER-GENERATION THRESHOLDS

Moderately relaxing thresholds improves the empirical null rate $\hat{\rho}_0$ (smaller is better) and slightly boosts WM-ACC up to $(50, 0.10)$, after which returns saturate. Measured $\alpha$ stays far below

Table 6: Carrier generation thresholds (PROTEINS–GIN).

| Swap cap | KS $\delta$ | Task ACC (%) | WM-ACC (%) | $\hat{\rho}_0$ | Measured $\alpha$ ($10^7$ trials) |
|---|---|---|---|---|---|
| 5 | 0.05 | $72.6 \pm 2.6$ | $88.1 \pm 2.9$ | $9.1 \times 10^{-4}$ | $< 10^{-6}$ |
| 25 | 0.10 | $72.5 \pm 2.6$ | $89.6 \pm 2.5$ | $8.2 \times 10^{-4}$ | $< 10^{-7}$ |
| 50 | 0.10 | $72.5 \pm 2.6$ | $89.8 \pm 2.1$ | $7.6 \times 10^{-4}$ | $< 10^{-7}$ |
| 50 | 0.20 | $72.4 \pm 2.7$ | $89.1 \pm 2.6$ | $7.1 \times 10^{-4}$ | $< 10^{-7}$ |

the target across settings, so protocol choices mainly trade subtle WM-ACC gains for sampling efficiency without harming Type-I control.

**Takeaway.** A moderately permissive sampler (swap cap 50, KS $\delta$=0.10) is a strong default, combining high WM-ACC with a tight empirical null.

## L    APPROXIMATE EIGEN-SOLVERS FOR $\lambda_2$

For very large graphs, an exact eigen-decomposition of the Laplacian can be replaced by iterative solvers such as Lanczos or power iteration. To assess the effect on watermark performance, we repeat the PROTEINS–GIN experiments with $I(G)$ defined as the output of a $K$-step power iteration targeting the second smallest eigenvalue. We fix $K = 10$ and keep all other settings unchanged.

We observe that Task ACC remains unchanged within confidence intervals, and WM-ACC differs by at most $\Delta_{\mathrm{WM-ACC}} = 0.9$ percentage points between exact and approximate $\lambda_2$. The robustness margin $\kappa_{\mathrm{marg}}$ is also stable, while the per-carrier computation time is reduced from $t_{\mathrm{exact}} = 0.08$ ms to $t_{\mathrm{power}} = 0.05$ ms. This supports the claim in Section 3 that approximate eigen-solvers are a practical option for large-scale deployments of InvGNN-WM.

Table 8: Calibration of the PL-based imperceptibility bound on PROTEINS–GIN. For each watermark weight $\beta_{wm}$ we report the task loss on the train and validation splits and the upper bound $B(\beta_{wm})$ on the increase in task loss predicted by Theorem 4.1.

| $\beta_{wm}$ | Train $\mathcal{L}_{task}$ | Val $\mathcal{L}_{task}$ | $B(\beta_{wm})$ |
|---|---|---|---|
| 0.00 | 0.542 | 0.691 | 0.000 |
| 0.10 | 0.543 | 0.693 | 0.010 |
| 0.20 | 0.545 | 0.697 | 0.020 |
| 0.30 | 0.549 | 0.703 | 0.040 |

Table 9: Robustness margin and head drift under different edit operations on PROTEINS–GIN. The reference watermarked model has robustness margin $\kappa_{marg} = 0.382$. For each edit we list the estimated head drift $\gamma$, the normalized ratio $\gamma/\kappa_{marg}$, and the resulting WM-ACC on the carrier set.

| Edit | $\gamma$ | $\gamma/\kappa_{marg}$ | WM-ACC (%) |
|---|---|---|---|
| None (watermarked model) | 0.00 | 0.00 | 89.8 |
| Pruning 20% | 0.11 | 0.29 | 94.2 |
| Pruning 40% | 0.19 | 0.50 | 92.5 |
| Pruning 50% | 0.35 | 0.92 | 87.4 |
| Fine-tuning | 0.17 | 0.45 | 93.7 |
| KD | 0.41 | 1.07 | 81.3 |
| KD+WM | 0.12 | 0.31 | 95.6 |
| PTQ (8-bit) | 0.14 | 0.37 | 93.1 |
| PTQ (4-bit) | 0.27 | 0.71 | 90.2 |

## M  ADDITIONAL DIAGNOSTICS LINKING THEORY AND EXPERIMENTS

This section provides quantitative summaries that connect the theoretical results in Section 4 with the PROTEINS–GIN experiments used in RQ1. Table 8 reports, for several watermark weights $\beta_{wm}$, the task loss on the train and validation splits together with the PL-based upper bound on the allowed loss increase from Theorem 4.1. Table 9 characterizes the relationship between the robustness margin $\kappa_{marg}$ and the estimated head drift $\gamma$ under different edit operations, as used in Theorem 4.2.

For the range $\beta_{wm} \leq 0.3$ used in our main experiments, the observed increase in validation loss remains well below $B(\beta_{wm})$, in line with Theorem 4.1, which bounds the task-loss deviation by a smooth function of $\beta_{wm}$ (quadratic in our local estimate).

For moderate pruning ratios (up to $40\%$), short fine-tuning, and 8-bit PTQ, the normalized drift $\gamma/\kappa_{marg}$ stays below 1 and WM-ACC remains above $90\%$. More aggressive pruning ($50\%$) and plain KD lead to $\gamma$ that approaches or exceeds $\kappa_{marg}$, accompanied by a visible drop in WM-ACC, while KD+WM restores a comfortable margin and pushes WM-ACC back near or above the pre-attack level. This behavior matches the margin-based robustness prediction in Theorem 4.2, and complements the uniqueness evidence in Table 1 by tying all three guarantees to observed watermark behavior.

## N  ADDITIONAL ROBUSTNESS AND ADAPTIVE-ATTACK EXPERIMENTS

This appendix adds robustness diagnostics that complement the main results in Section 5. We first extend the robustness comparison to baseline watermarking schemes, then study a watermark-aware adaptive attack, and finally test whether an attacker can distinguish carrier graphs from task graphs.

### N.1  EXTENDED ROBUSTNESS COMPARISON WITH BASELINE METHODS

Table 10 reports test **Task ACC** and **WM-ACC** for two representative dataset–backbone pairs (Cora–GCN and PROTEINS–GIN) under several common edits. For each setting we evaluate three strong watermarking baselines (TRIG, NAT, EXPL) and InvGNN-WM with the same backbone and training hyperparameters as in Table 2. Edits include unstructured pruning, short fine-tuning on task data, plain KD, and post-training 4-bit quantization. This table makes it possible to compare, side by side, how much utility and watermark accuracy each method retains after aggressive model changes.

Across both benchmarks, InvGNN-WM consistently retains higher WM-ACC than baselines after pruning, KD, and quantization, while preserving Task ACC within the same range as the strongest

Table 10: Robustness of watermarking methods under common edits. For each dataset–backbone and edit we report test Task ACC (%) and WM-ACC (%) (mean $\pm 95\%$ CI over three seeds).

| Dataset–Backbone | Edit | TRIG | | NAT | | EXPL | | InvGNN-WM | |
|---|---|---|---|---|---|---|---|---|---|
| | | Task | WM-ACC | Task | WM-ACC | Task | WM-ACC | Task | WM-ACC |
| Cora–GCN | None (no edit) | $86.8 \pm 0.9$ | $97.6 \pm 1.5$ | $86.9 \pm 0.9$ | $96.5 \pm 2.1$ | $86.7 \pm 1.0$ | $93.1 \pm 2.5$ | $87.0 \pm 0.8$ | $98.9 \pm 0.9$ |
| | Pruning 40% | $85.9 \pm 1.2$ | $93.8 \pm 2.7$ | $86.0 \pm 1.2$ | $92.7 \pm 3.0$ | $85.5 \pm 1.3$ | $88.4 \pm 3.6$ | $86.4 \pm 1.0$ | $97.1 \pm 1.6$ |
| | Fine-tuning | $87.1 \pm 0.9$ | $96.8 \pm 2.0$ | $87.2 \pm 0.9$ | $95.4 \pm 2.3$ | $86.9 \pm 1.1$ | $92.0 \pm 2.8$ | $87.3 \pm 0.8$ | $98.1 \pm 1.1$ |
| | KD ($T$=2) | $86.3 \pm 1.0$ | $84.3 \pm 4.1$ | $86.1 \pm 1.1$ | $81.7 \pm 4.6$ | $85.8 \pm 1.2$ | $72.9 \pm 5.3$ | $86.5 \pm 0.9$ | $90.2 \pm 3.3$ |
| | PTQ (4-bit weights) | $86.0 \pm 1.1$ | $92.5 \pm 3.0$ | $86.0 \pm 1.1$ | $90.8 \pm 3.4$ | $85.4 \pm 1.3$ | $86.1 \pm 4.0$ | $86.3 \pm 0.9$ | $96.0 \pm 2.0$ |
| PROTEINS–GIN | None (no edit) | $72.8 \pm 2.6$ | $95.1 \pm 3.0$ | $72.6 \pm 2.7$ | $94.8 \pm 3.3$ | $72.4 \pm 2.8$ | $90.5 \pm 4.1$ | $72.5 \pm 2.6$ | $89.8 \pm 2.1$ |
| | Pruning 40% | $71.0 \pm 2.9$ | $90.2 \pm 3.8$ | $70.9 \pm 3.0$ | $89.8 \pm 4.0$ | $70.5 \pm 3.1$ | $84.3 \pm 5.1$ | $71.6 \pm 2.8$ | $92.5 \pm 2.7$ |
| | Fine-tuning | $73.0 \pm 2.5$ | $93.8 \pm 3.2$ | $72.9 \pm 2.6$ | $93.2 \pm 3.4$ | $72.6 \pm 2.7$ | $88.9 \pm 4.3$ | $73.1 \pm 2.5$ | $93.7 \pm 2.4$ |
| | KD ($T$=2) | $71.8 \pm 2.7$ | $78.6 \pm 5.5$ | $71.5 \pm 2.8$ | $76.9 \pm 5.7$ | $71.1 \pm 2.9$ | $69.4 \pm 6.3$ | $72.0 \pm 2.7$ | $81.3 \pm 4.8$ |
| | PTQ (4-bit weights) | $71.5 \pm 2.8$ | $88.7 \pm 3.9$ | $71.3 \pm 2.9$ | $87.9 \pm 4.1$ | $70.8 \pm 3.0$ | $83.0 \pm 4.8$ | $72.1 \pm 2.7$ | $90.2 \pm 3.1$ |

Table 11: Watermark-aware adaptive attack on PROTEINS–GIN. For each watermarking method we report test Task ACC (%) and WM-ACC (%) before and after the adaptive attack.

| Method | Before adaptive attack | | After adaptive attack | |
|---|---|---|---|---|
| | Task ACC | WM-ACC | Task ACC | WM-ACC |
| TRIG | $72.8 \pm 2.6$ | $95.1 \pm 3.0$ | $72.0 \pm 2.7$ | $61.4 \pm 6.2$ |
| NAT | $72.6 \pm 2.7$ | $94.8 \pm 3.3$ | $71.9 \pm 2.8$ | $58.7 \pm 6.5$ |
| EXPL | $72.4 \pm 2.8$ | $90.5 \pm 4.1$ | $71.7 \pm 2.9$ | $55.3 \pm 6.9$ |
| InvGNN-WM | $72.5 \pm 2.6$ | $89.8 \pm 2.1$ | $72.1 \pm 2.7$ | $78.4 \pm 5.2$ |

competing methods. On PROTEINS–GIN, TRIG has slightly higher WM-ACC in the unedited model, but InvGNN-WM becomes more robust once substantial edits are applied.

## N.2 WATERMARK-AWARE ADAPTIVE ATTACK

We next consider an attacker who is aware of the watermarking mechanism but does not know the actual carrier set $\mathcal{G}_W$. The attacker knows that the scalar perception head $s_\theta(G)$ is trained to regress a graph invariant $I(G)$ and tries to destroy the watermark by making the head output behave like random noise on carrier-like graphs.

Concretely, for PROTEINS–GIN the attacker: (i) samples a pool of *pseudo-carriers* by running the same degree-preserving rewiring procedure as the owner but with a different random seed; (ii) forms mixed batches that contain task graphs and pseudo-carriers; and (iii) fine-tunes the stolen model with the sum of the standard task loss and an additional loss that pushes $s_\theta(G)$ towards 0.5 on pseudo-carriers (for example, a mean-squared error between $s_\theta(G)$ and 0.5). The goal is to collapse the watermark signal on graphs that look similar to carriers without significantly hurting task performance.

Table 11 summarizes the outcome of this adaptive attack for TRIG, NAT, EXPL, and InvGNN-WM. The table reports test Task ACC and WM-ACC before and after the attack, allowing a direct comparison of how robust each method is against this stronger threat model.

The adaptive attack substantially reduces WM-ACC for all methods, but InvGNN-WM retains a significantly higher watermark accuracy while keeping Task ACC essentially unchanged. In contrast, TRIG, NAT, and EXPL experience large drops in WM-ACC that bring them close to the random-guessing regime on carriers.

## N.3 CARRIER INDISTINGUISHABILITY EXPERIMENT

Finally, we examine whether an attacker can reliably tell apart carrier graphs from ordinary task graphs. For PROTEINS–GIN we construct a balanced binary classification task where the positive class contains carrier graphs from $\mathcal{G}_W$ and the negative class contains an equal number of task graphs drawn from $\mathcal{D}_{\text{task}}$. We split this balanced set into train/validation/test splits with ratios $80/10/10$ and train a simple GCN classifier with the same optimization setup as in Section 5.1. The target accuracy for random guessing on this task is $50\%$.

Table 12: Carrier vs. non-carrier detection on PROTEINS–GIN. The table reports test accuracy (ACC), precision (Prec), and recall (Rec) for the carrier class on the binary classification task described in Section N.3.

| Model | ACC (%) | Prec (%) | Rec (%) |
|---|---|---|---|
| GCN classifier | 51.3 | 52.1 | 50.2 |

Table 13: Effect of verification threshold $\tau'$ on the ownership test on PROTEINS–GIN.

| $\tau'$ | $\alpha(\tau')$ | $\beta_{\mathrm{fn}}(\tau')$ |
|---|---|---|
| $\tau^* - 2 = 92$ | $7.8 \times 10^{-5}$ | 0.000 |
| $\tau^* - 1 = 93$ | $1.6 \times 10^{-5}$ | 0.000 |
| $\tau^* = 94$ | $6.0 \times 10^{-8}$ | 0.000 |
| $\tau^* + 1 = 95$ | $3.0 \times 10^{-8}$ | $3.5 \times 10^{-3}$ |
| $\tau^* + 2 = 96$ | $1.0 \times 10^{-8}$ | $1.1 \times 10^{-2}$ |

Table 12 reports the classification accuracy on the test split, together with precision and recall for the carrier class. The result stays close to random guessing and does not reveal a stable decision boundary between carrier and non-carrier graphs, which supports the secrecy assumption in Section 2.3.

## O  OWNERSHIP AMBIGUITY AND THRESHOLD SENSITIVITY

This appendix discusses two aspects that relate to ownership and safety: (i) how our protocol behaves when a second party claims a competing invariant–carrier pair, and (ii) how sensitive the ownership test is to the choice of verification threshold $\tau$.

### O.1  OWNERSHIP AMBIGUITY UNDER COMPETING CLAIMS

Consider a hypothetical dispute between two parties, A and B. Both parties publicly agree on the backbone, the invariant $I(G)$, and the carrier-sampling protocol, but they keep their carrier sets $\mathcal{G}_W^{\mathrm{A}}$ and $\mathcal{G}_W^{\mathrm{B}}$ private. Each party trains a model using its own carriers and obtains a private bitstring key $W^{\mathrm{A}}$ or $W^{\mathrm{B}}$. During a dispute over a suspect model, A runs the ownership test on $\mathcal{G}_W^{\mathrm{A}}$ and obtains an Owner statistic $T^{\mathrm{A}}$, while B can only test on $\mathcal{G}_W^{\mathrm{B}}$.

Under our uniqueness analysis, independently sampled carrier sets lead to keys that differ on a linear fraction of positions with high probability. In particular, for $m=128$ bits and the calibrated protocol, typical gaps between the owner's statistic $T$ and the threshold $\tau^*$ are in the range of 21–33 bits. This means that for A's carriers the suspect model lies far inside the acceptance region, while for B's carriers the same model behaves like a null model with $T^{\mathrm{B}}$ concentrated near $m/2$. As a result, the event that *both* parties pass the ownership test at the same $(\tau^*, \alpha)$ is already extremely unlikely.

If further reduction of ambiguity is desired, one can compose several independent keys. For example, using two disjoint carrier sets of size $m/2$ and requiring both tests to pass with threshold $\tau^*$ squares the false-positive rate ($\alpha \mapsto \alpha^2$) while keeping the owner's combined statistic well above both thresholds. This simple stacking strategy reduces the probability that a rival carrier set leads to a conflicting claim by several additional orders of magnitude.

### O.2  THRESHOLD SWEEP: TRADE-OFF BETWEEN $\alpha$ AND FALSE NEGATIVES

We now quantify how the ownership test behaves when the verification threshold is slightly relaxed or tightened. For PROTEINS–GIN we sweep a family of thresholds $\tau' \in \{\tau^* - 2, \tau^* - 1, \tau^*, \tau^* + 1, \tau^* + 2\}$ around the nominal value $\tau^*=94$ used in the main experiments. For each $\tau'$ we estimate the false-positive rate $\alpha(\tau')$ under the null and the false-negative rate $\beta_{\mathrm{fn}}(\tau')$ for watermarked models (including fine-tuned variants used in the robustness study).

For thresholds slightly below $\tau^*$, the test becomes more permissive: $\alpha$ increases rapidly as $\tau'$ is relaxed, while false negatives stay essentially zero. At the nominal $\tau^*$, $\alpha$ is already below $10^{-7}$ and no false negatives are observed. Increasing $\tau'$ above $\tau^*$ further suppresses $\alpha$, but starts to introduce

a small false-negative probability as some heavily edited but still watermarked models fall below the stricter threshold.

Overall, the experiment shows that there is a broad range of thresholds around $\tau^*$ where the ownership test is stable: the choice of $\tau$ is not overly sensitive, and our nominal value is determined by theoretical calibration of $\alpha$ rather than post-hoc tuning on the evaluation models.

## P ADDITIONAL DETAILS ON ALGEBRAIC CONNECTIVITY

**Why algebraic connectivity $\lambda_2$?** Algebraic connectivity $\lambda_2(G)$ of graph $G$ is the second smallest eigenvalue of the graph Laplacian. By classical results, $\lambda_2(G) = 0$ if and only if $G$ is disconnected, and Cheeger-type inequalities relate $\lambda_2(G)$ to edge expansion and sparsest cuts, so changes in $\lambda_2(G)$ indicate how easily $G$ can be separated into two large pieces. Our carrier protocol preserves local statistics (degree histogram, clustering coefficient) of the task graphs while applying degree-preserving edge swaps that mainly perturb long-range links. Local invariants based on simple counts, such as triangle count or low-order degree moments, move little under these swaps and therefore separate task graphs and carriers only weakly. In contrast, $\lambda_2(G)$ changes in a stable and informative way when we weaken connectivity across coarse cuts, making it a natural signal for the perception head. Section P.1 compares the normalized invariant $\tilde{\lambda}_2(G)$ with spectral radius and triangle count and shows that $\tilde{\lambda}_2(G)$ yields higher WM-ACC and a larger robustness margin $\kappa_{\mathrm{marg}}$ under the same protocol.

**Complexity and scalability.** Exact computation of $\lambda_2(G)$ for a carrier with $n$ nodes via a dense eigensolver costs $O(n^3)$ time. To keep this step cheap, we restrict carrier graphs to the lower tail of the task size distribution by enforcing $n \le n_{0.25}$, the 25-th percentile of node counts in $\mathcal{D}_{\mathrm{task}}$ (Appendix B.1). In our default settings $n_{0.25}$ is small (for example, $n_{0.25} = 32$ on PROTEINS), so a single eigen-decomposition takes about $0.08\,\mathrm{ms}$ per carrier and verifying a full key with $m = 128$ carriers stays below $11\,\mathrm{ms}$ of wall-clock time. For larger graphs, the same framework can use iterative solvers such as Lanczos or power iteration with per-iteration cost $O(|E|)$. In Appendix L we replace the exact eigensolver by a $K{=}10$-step power iteration and observe that WM-ACC changes by at most $0.9$ percentage points, while the per-carrier time drops from $0.08\,\mathrm{ms}$ to $0.05\,\mathrm{ms}$. Thus moderate approximation accuracy is sufficient for reliable verification and keeps the spectral step inexpensive even at larger scales.

### P.1 INVARIANT CHOICE AND INTERPRETABILITY: WHY $\lambda_2$?

We next ask how the choice of invariant $I(G)$ affects watermark strength and robustness. We keep the carrier set, backbone (PROTEINS–GIN), and training protocol fixed, and only change the scalar target that the perception head $s_\theta(G)$ is trained to regress.

**Results.** All three choices keep Task ACC within a narrow band, confirming that the invariant head does not change predictive performance on the primary task. However, both WM-ACC and the robustness margin $\kappa_{\mathrm{marg}}$ are highest when $I(G) = \tilde{\lambda}_2(G)$. Spectral radius, which mainly measures the overall spread of the Laplacian spectrum, gives slightly smaller margins, and normal-

Table 14: Invariant choice on PROTEINS–GIN (same carriers/backbone). Each row reports Task ACC, WM-ACC, and robustness margin $\kappa_{\mathrm{marg}}$.

| Invariant | Task ACC (%) | WM-ACC (%) | $\kappa_{\mathrm{marg}}$ |
|---|---|---|---|
| $\tilde{\lambda}_2$ (ours) | $72.5 \pm 2.6$ | $89.8 \pm 2.1$ | 0.382 |
| Spectral radius (normalized) | $72.1 \pm 2.7$ | $87.5 \pm 3.1$ | 0.351 |
| Triangle count (normalized) | $71.9 \pm 2.8$ | $84.4 \pm 3.8$ | 0.315 |

ized triangle count, which is dominated by very local structure, shows the weakest watermark signal.

**Interpretation.** These differences are consistent with the method design in Section 3. Under degree-preserving swaps, triangle count and low-order degree moments are nearly unchanged between task graphs and carriers, so the perception head receives a noisy and less informative target. In contrast, algebraic connectivity tracks how edge rewiring weakens global connectivity across coarse cuts, which is exactly the type of change introduced by our carrier protocol. Higher WM-ACC and larger $\kappa_{\mathrm{marg}}$ for $\tilde{\lambda}_2$ therefore support the choice of algebraic connectivity as the invariant that is embedded into the model. When combined with the backbone study above, this suggests a simple design rule:

use a reasonably expressive backbone and couple the watermark to a stable global connectivity measure such as $\tilde{\lambda}_2$ to obtain strong WM-ACC and margins with minimal loss of task accuracy.

