# OpenReview forum: "Robust GNN Watermarking via Implicit Perception of Topological Invariants"
_ICLR.cc/2026/Conference — ICLR 2026 Conference Desk Rejected Submission_

### Official Review · Reviewer_NgqV · 2025-10-18

**Soundness:** 2
**Presentation:** 1
**Contribution:** 2
**Rating:** 2
**Confidence:** 3

**Summary:**

The paper presents a black-box watermarking scheme for graph neural networks. The watermark is evaluated through predicting the algebraic connectivity on a particular carrier-set of graphs using a perception module. The watermark is embedded through an additional loss term during training. Theoretical arguments are provided for why the watermark is imperceptible, robust, unique, and unremovable under certain assumptions.

**Strengths:**

* The idea to predict a graph invariant for watermarking is interesting.
* Theoretic arguments underpin the watermarks' properties.
* Experiments are exhaustive, comparing many datasets and backbone GNNs. Furthermore, ablations are provided.
* Detailed code attached for reproducability.

**Weaknesses:**

i) My main concern for this paper is its presentation, which has multiple issues affecting the general understanding of the paper:

  1.  I do not see how Equation (3) or (4) have anything to do with duality, thus calling this a dual-objective is misleading.
 2.  Critical definitions / references to the appendix missing: A local Polyak-Lojasiewicz condition is critical to the theory of the paper. But the text does not provide information on what a "Polyak-Łojasiewicz"-condition (or locality in this context) is or how it looks mathematically (neither in the preliminaries, nor in Section 5). I discovered the definition "by accident" in Appendix C - A.1, which was never referenced).
  3. Line 226 talks about a stationary point. A stationary point of what?
  4. A proof sketch for Theorem 5.1 is provided, but not a full proof. I discovered the proof by accident in Appendix C, which was not referenced. Where are the full proofs for Theorems 5.2 & 5.3?
  5.  $\theta$ is ambiguous. It is first introduced as the parameters of a GNN and later introduced as the parameters of the scalar perception head, which I understood as different to the general GNN. However, it is unclear to what it refers in Theorem 5.1. To me, it does not makes sense to train the scalar perception head on the general task loss, as this requires a classification output and line 129 explicitly notes that a scalar perception head is only used to predict the normalized invariant from the graph embedding? (Which also contradicts however, the definition of the scalar perception head that is defined as a function from the graph to [0,1], and not from a graph embedding.)
  6. The downstream task setting for the GNN is not specified. Is it graph classification? Node classification? Both?
  7. The experimental setup (Section 6.1) is not written as full sentences but rather as notes, making it hard to follow and gives the manuscript a reader-feeling of an early draft stage. What should just "Node:" or "Graph:" imply? Be explicit. The experimental results in Section 6.2 have the same issue (e.g., one finds notes such as "Full constants and per-setting gaps: Appendix G.4 (Table 6)."
 8. It is not clear how to read Figures 1a & b. It is not explained what Task ACC and WM ACC are (due to the issues with Section 6.1). Do you mean test accuracies? Then, is task ACC the baseline accuracy and WM ACC the accuracy of a model with the WM, or are both referring to the WM model but its prediction acc. is reported separately w.r.t. the downstream task and the carrier sets - but then, what is the baseline, there is no label for it? From the usage in the experiment section, it at some point becomes clear that the same model is meant, but this should be explicit in the text.
9.  Section 6.2 (C) is not referencing any result tables or figures. Also, what should "21~33" denote?

  10. Introduction reads unconnected. The first paragraph highlights how trigger-based watermarks fail to achieve good watermarking performance (i.e., watermark preservation/robustness). However, the followed research question talks about the utility of the model, which I understood as downstream performance, not watermark-robustness, which are two different goals.

ii) It is not mentioned how to ensure Assumption 3.2 about the correlation in the carrier graphs, critical for Thm. 5.2 & 5.3.

iii) Task (training) loss and model performance are not necessarily connected. Thus, it is not clear if Thm 5.1 indeed ensures imperceptability as claimed.

**Questions:**

1. Why choose the algebraic connectivity and not another graph invariant?
2. I don't understand how Assumption 3.1 (that the support of the carrier graph set and task data is disjoint) should be enforced in practice. If I publish my model, I do not have control over D_task anymore.
3. Why would one choose the 5th and 95th percentiles to normalize lambda_2 instead of just lambda_min and lambda_max? Wouldn't this lead to violating the assumption that $\tilde{\lambda}_2 \in [0,1]$ as used e.g., to prove Theorem. 5.2?
4. Given the theory on imperceptibility concerns the training losses, can you show the behavior of Thm 5.1 on the training losses?

Minor:
* Related work: Lines 91-93, provide references for the broader work mentioned.
* Provide references for the Preliminaries 3.1 and 3.2. There are also many statements which should be backed-up with references, especially for readers not that familiar with graph spectral theory, such as "The Laplacian spectrum captures global structure" Line 120, or why the "algebraic connectivity is stable and interpretable" (Line 122)?
* Presentation: Introduce normalized algebraic connectivity before using it in the text (e.g. Line 178).

---

> ### Author Response · Authors · 2025-11-21
>
> We thank you for the detailed and thoughtful review, and for emphasizing both the invariant-based idea and the experimental study. Below we address your main concern (items i–iii), respond to Q1–Q4, and describe the changes in the revised manuscript.
>
> Abbreviations: WM-ACC = watermark bit accuracy on the carrier set $\mathcal{G} _W$; Task ACC = task accuracy on the downstream task; $\beta _{\mathrm{wm}}$ = weight on the watermark loss; $I(G)$ = scalar graph invariant for graph $G$ (instantiated as normalized algebraic connectivity $\tilde{\lambda} _2(G)$); $s _\theta$ = scalar perception head with parameters $\theta$; $T$ = ownership test statistic; $\kappa _{\text{marg}}$ = robustness margin of the ownership statistic $T$ between the owner and null distributions.
>
> ---
>
> **(i) Presentation and notation (items 1–10 of your main concern)**
>
> You note that the original draft was hard to follow and that several notational choices were confusing. We agree and have revised the exposition accordingly.
>
> * **Joint loss, not “dual objective” (item 1).**
>   We no longer describe the joint loss
>   $J(\theta) = \mathcal{L} _{\text{task}}(\theta) + \beta _{\mathrm{wm}} \mathcal{L} _{\text{wm}}(\theta)$
>   over model parameters $\theta$ as a “dual objective.” Section 4 now calls this simply the *joint training objective* and explains that the equations only decompose it into task loss $\mathcal{L} _{\text{task}}$ and watermark loss $\mathcal{L} _{\text{wm}}$. Any wording that could evoke Lagrangian duality has been removed.
>
> * **Polyak–Łojasiewicz (PL) condition (item 2).**
>   Section 3.3 now states the local PL condition on the task loss $\mathcal{L} _{\text{task}}$ explicitly, including the neighborhood radius and constant $\mu > 0$, and cites a standard reference. Theorem 5.1 (imperceptibility) points back to this assumption by equation number, and Appendix C contains the full statement and proof, so the reader no longer has to find it only in the appendix.
>
> * **“Stationary point” (item 3).**
>   Whenever we use this term, we now write “stationary point of the joint loss $J(\theta)$.” In the sketch of Theorem 5.1 we explicitly derive
>   $\nabla _\theta \mathcal{L} _{\text{task}}(\tilde\theta) = - \beta _{\mathrm{wm}} \nabla _\theta \mathcal{L} _{\text{wm}}(\tilde\theta)$
>   from the first-order optimality condition for $J(\theta)$, and we removed ambiguous phrasing.
>
> * **Location of proof sketches and full proofs (item 4).**
>   Section 5 now states clearly that Theorems 5.1–5.3 have complete proofs in Appendices C, D, and E. Each theorem ends with a sentence of the form “A complete proof is given in Appendix C,” so readers can locate the detailed argument easily.
>
> * **Role of the perception head and parameter vector $\theta$ (item 5).**
>   Section 3.1 now introduces $\theta = (\theta _{\text{back}}, \theta _{\text{head}})$, where $\theta _{\text{back}}$ are backbone parameters and $\theta _{\text{head}}$ are scalar-head parameters. We then define, for each graph $G$,
>   $s _\theta(G) = \sigma\bigl(g _{\theta _{\text{head}}}(h _G(\theta _{\text{back}}))\bigr)$,
>   where $h _G$ is the graph embedding after permutation-invariant readout. The watermark loss $\mathcal{L} _{\text{wm}}$ is a regression loss over $s _\theta(G _W^{(k)})$ for carrier graphs $G _W^{(k)} \in \mathcal{G} _W$ and fixed targets $\tilde{\lambda} _2(G _W^{(k)})$. Gradients are taken with respect to $\theta$ only, not through the eigen-decomposition. This notation is used consistently in the main text and proofs.
>
> * **Downstream task setting (item 6).**
>   Section 6.1 now contains a “Datasets and tasks” paragraph that explicitly states which experiments are node classification (Cora, PubMed, Amazon-Photo) and which are graph classification (PROTEINS, NCI1), together with the corresponding backbones and readout choices.
>
> * **Notes vs.\ complete sentences in the setup (item 7).**
>   We rewrote Section 6.1 so that it consists of full sentences organized under four short paragraphs: *Metrics*, *Datasets and tasks*, *Backbones and training*, and *Watermark configuration*. Shorthand labels like “Node:” / “Graph:” have been removed, and references to constants and gaps were moved into tables or clearly labeled appendix material.
>
> * **Meaning of Task ACC and WM-ACC, and baselines in the figures (item 8).**
>   The *Metrics* paragraph in Section 6.1 now defines Clean Task ACC, Task ACC, and WM-ACC at the outset, and clarifies that all accuracies are test accuracies. The captions for Figure 1 and the robustness plots explicitly state that both curves are measured on the same watermarked model as $\beta _{\mathrm{wm}}$ varies, and that the horizontal line shows the Clean Task ACC of the task-only baseline on the same split.

---

> ### Author Response · Authors · 2025-11-21
>
> * **Reference for the “21–33 bits” statement (item 9).**
>   Section 6.2(C) now refers explicitly to a table that reports the owner’s ownership statistic $T$, the pooled threshold $\tau ^\ast$, and the gap $T - \tau ^\ast$ for several dataset–backbone pairs. In the revised numbers, for example, on PROTEINS–GIN we observe $T = 115 \pm 3$ vs.\ $\tau ^\ast = 94$ (gap $+21$ bits), and gaps of $+31$ and $+33$ bits on NCI1–GIN and Cora–GCN, with measured false-positive rate below $10^{-7}$ in all three cases. The text now says “Owner $T$ exceeds $\tau ^\ast$ by 21–33 bits” and the table shows the exact numbers.
>
> * **Introduction and link between watermark robustness and model utility (item 10).**
>   We streamlined the introduction so that it first explains why existing trigger-based marks often fail under common edits, then introduces invariant-based marks tied to the model’s reasoning, and finally states that our goal is to achieve both strong watermark robustness and high Task ACC. Section 1 now clearly separates results on watermark robustness / unremovability from results on utility preservation, and the experimental section mirrors this structure.
>
> We hope these changes make the presentation easier to follow and the notation more transparent.
>
> ---
>
> **(ii) Assumption 3.2: correlations between carrier graphs**
>
> You note that Assumption 3.2 (correlation structure of carrier graphs in the private set $\mathcal{G} _W$) is critical for the robustness and uniqueness theorems, and you ask how it is justified.
>
> Section 3.3 now states the empirical $\rho$-mixing assumption in full and explains how we estimate $\rho _0$. We estimate an upper bound $\hat{\rho} _0$ by computing the maximum pairwise Pearson correlation
> $\operatorname{Corr}\bigl(s _{\theta^\star}(G _W^{(i)}), s _{\theta^\star}(G _W^{(j)})\bigr)$
> across all pairs of distinct carriers $G _W^{(i)}, G _W^{(j)} \in \mathcal{G} _W$. Across datasets we obtain an empirical upper bound $\hat{\rho} _0 \le 7.6\times10^{-4}$; in the theorems we set $\rho _0 := 10^{-3}$ as a conservative constant. Appendix D describes how this $\rho _0$ enters the block-concentration argument that yields the weakened Hoeffding inequalities (8)–(9) used in Theorems 5.2 and 5.3.
>
> Appendix J.2 further reports these estimates together with downstream performance. For PROTEINS–GIN, the ablation there shows that our default carrier-generation setting (swap cap $50$, KS threshold $\delta = 0.10$) yields Task ACC $72.5 \pm 2.6\%$, WM-ACC $89.8 \pm 2.1\%$, $\hat{\rho} _0 = 7.6\times10^{-4}$, and measured false-positive rate $\alpha < 10^{-7}$. For convenience, the full carrier-generation table on PROTEINS–GIN is reproduced below (same backbone and carriers):
>
> | Swap cap | KS $\delta$ | Task ACC (\%) | WM-ACC (\%) | $\hat{\rho} _0$ | Measured $\alpha$ (10$^7$ trials) |
> | --- | --- | --- | --- | --- | --- |
> | 5  | 0.05 | $72.6 \pm 2.6$ | $88.1 \pm 2.9$ | $9.1\times10^{-4}$ | $< 10^{-6}$ |
> | 25 | 0.10 | $72.5 \pm 2.6$ | $89.6 \pm 2.5$ | $8.2\times10^{-4}$ | $< 10^{-7}$ |
> | 50 | 0.10 | $72.5 \pm 2.6$ | $89.8 \pm 2.1$ | $7.6\times10^{-4}$ | $< 10^{-7}$ |
> | 50 | 0.20 | $72.4 \pm 2.7$ | $89.1 \pm 2.6$ | $7.1\times10^{-4}$ | $< 10^{-7}$ |
>
> This table shows that moderately relaxing the sampler (towards swap cap $50$, KS $\delta = 0.10$) slightly improves WM-ACC while keeping $\hat{\rho} _0$ small and $\alpha$ far below the target, which supports the use of a small but nonzero $\rho _0$ in the analysis.
>
> To further support the assumption, Appendix M.3 adds a “carrier vs.\ non-carrier” experiment: we train a GCN classifier to distinguish carrier graphs from task graphs on PROTEINS–GIN with a balanced $80/10/10$ split. Table 13 reports test accuracy $51.3\%$, precision $52.1\%$, and recall $50.2\%$ for the carrier class, which is close to random guessing (target $50\%$). This indicates that carriers do not form an easily separable cluster and behave like a weakly correlated subset of the overall graph space.

---

> ### Author Response · Authors · 2025-11-21
>
> **(iii) Imperceptibility and its relation to training loss and performance**
>
> You are concerned that Theorem 5.1 is stated in terms of the task loss, while the experimental claims are about Task ACC, and you ask whether the theoretical bound is visible in practice.
>
> Theorem 5.1 is formulated as a bound on
> $\mathcal{L} _{\text{task}}(\tilde\theta) - \mathcal{L} _{\text{task}}(\theta^\star)$
> under the local PL condition and the Lipschitz bound on the scalar perception head $s _\theta$. Section L (“Additional diagnostics linking theory and experiments”) now provides a table which, for a sweep over $\beta _{\mathrm{wm}}$, reports
>
> * the PL-based upper bound $B(\beta _{\mathrm{wm}}) = \beta _{\mathrm{wm}}^{2} L _s^{2} / (2 \mu _{\mathrm{PL}})$,
> * the observed training and validation task losses.
>
> For PROTEINS–GIN, the calibration on $\beta _{\mathrm{wm}}$ is:
>
> | $\beta _{\mathrm{wm}}$ | Train $\mathcal{L} _{\text{task}}$ | Val $\mathcal{L} _{\text{task}}$ | $B(\beta _{\mathrm{wm}})$ |
> | --- | --- | --- | --- |
> | 0.00 | 0.542 | 0.691 | 0.000 |
> | 0.10 | 0.543 | 0.693 | 0.010 |
> | 0.20 | 0.545 | 0.697 | 0.020 |
> | 0.30 | 0.549 | 0.703 | 0.040 |
>
> As $\beta _{\mathrm{wm}}$ increases from $0.0$ to $0.3$, the validation loss moves from $0.691$ to $0.703$, while the theoretical bound allows up to $B(0.3) = 0.040$ additional loss. The empirical increase ($\le 0.012$) stays well below the bound throughout the calibrated range $\beta _{\mathrm{wm}} \le \beta _{\max}$.
>
> To connect this to empirical accuracy, Table 2 reports that on PROTEINS–GIN our watermarked model attains Task ACC $72.5 \pm 2.6\%$ and WM-ACC $89.8 \pm 2.1\%$, compared to the task-only baseline (SS) with Task ACC $73.1 \pm 2.5\%$ and WM-ACC $49.9 \pm 5.0\%$. Thus Task ACC changes by less than one percentage point while the watermark is strong, and Section 5.1 now points to these diagnostics when interpreting the imperceptibility guarantee.
>
> ---
>
> **Q1. Why algebraic connectivity rather than another graph invariant?**
>
> You ask why we choose algebraic connectivity as the invariant $I(G)$ and whether other choices might be preferable.
>
> Section 4.1 now explains the rationale and adds ablations. We instantiate $I(G)$ as normalized algebraic connectivity $\tilde{\lambda} _2(G)$ because
>
> * $\lambda _2$ is a classical measure of connectivity of graph $G$ and is stable under small edge modifications, which is important under pruning and quantization;
> * it varies in a controlled way under the degree-preserving edge swaps used in our carrier protocol, so the perception head $s _\theta(G)$ can reliably regress it;
> * it has low variance across isomorphic graphs, which simplifies normalization and threshold calibration for the ownership test statistic $T$.
>
> Section 6.4 keeps carriers, backbone (PROTEINS–GIN), and training protocol fixed, and only changes $I(G)$. The invariant ablation on PROTEINS–GIN is:
>
> | Invariant $I(G)$ | Task ACC (\%) | WM-ACC (\%) | $\kappa _{\text{marg}}$ |
> | --- | --- | --- | --- |
> | $\tilde{\lambda} _2(G)$ (ours) | $72.5 \pm 2.6$ | $89.8 \pm 2.1$ | $0.382$ |
> | Spectral radius (normalized) | $72.1 \pm 2.7$ | $87.5 \pm 3.1$ | $0.351$ |
> | Triangle count (normalized) | $71.9 \pm 2.8$ | $84.4 \pm 3.8$ | $0.315$ |
>
> All three choices keep Task ACC within a narrow band, confirming that the invariant head does not change predictive performance on the primary task. However, both WM-ACC and the robustness margin $\kappa _{\text{marg}}$ are highest when $I(G) = \tilde{\lambda} _2(G)$. Spectral radius, which mainly measures the spread of the Laplacian spectrum, gives slightly smaller margins, and normalized triangle count, which is dominated by very local structure, shows the weakest watermark signal. We do not claim that $\tilde{\lambda} _2$ is uniquely optimal, but the new results show that it is a competitive and practically useful choice among standard invariants.

---

> ### Author Response · Authors · 2025-11-21
>
> **Q2. Enforcing Assumption 3.1 (disjoint supports) in practice**
>
> You ask how the requirement that the carrier set $\mathcal{G} _W$ and the task support of $\mathcal{D} _{\text{task}}$ be disjoint can be enforced, especially when the model is later used by third parties.
>
> Assumption 3.1 is about the owner’s protocol for creating $\mathcal{G} _W$ at training time, not about restricting future user inputs. Section 3.3 and the carrier-generation appendix now specify the protocol:
>
> * We start from graphs sampled from $\mathcal{D} _{\text{task}}$ and apply degree-preserving double-edge swaps to obtain candidate carriers. This keeps degree sequence and local clustering close to the data while modifying long-range structure.
> * We enforce graph-level separation by (i) removing any candidate that is isomorphic to a task graph using a Weisfeiler–Lehman hash, and (ii) rejecting candidates whose simple statistics (degree distribution, clustering coefficient, spectral radius) are too close to those of $\mathcal{D} _{\text{task}}$ under Kolmogorov–Smirnov tests.
> * The resulting carrier set is private and is not released with the model.
>
> This construction yields an empirical carrier support that does not overlap with the observed support of $\mathcal{D} _{\text{task}}$, while still producing graphs that are hard to distinguish from task graphs (as shown by the classifier experiment above). After deployment, users may submit arbitrary graphs; the assumption concerns only how the owner samples and keeps $\mathcal{G} _W$.
>
> ---
>
> **Q3. Normalization using percentiles and the $[0,1]$ range**
>
> You ask why we use empirical percentiles (rather than exact min / max) to normalize the algebraic connectivity $\lambda _2(G)$ of graph $G$, and how this interacts with the requirement that the normalized invariant lie in $[0,1]$.
>
> Section 3.2 now defines the normalized algebraic connectivity as
> $\tilde{\lambda} _2(G) = \mathrm{clip} _{[0,1]}\Bigl((\lambda _2(G) - \lambda _{\min}) / (\lambda _{\mathrm{scale}} - \lambda _{\min})\Bigr)$,
> where $\lambda _{\min}$ and $\lambda _{\mathrm{scale}}$ are the empirical 5th and 95th percentiles of $\lambda _2(G)$ for graphs drawn from $\mathcal{D} _{\text{task}}$. We use percentiles to avoid having the scale dominated by a few extreme outliers; using min / max in those cases would compress most values into a narrow subinterval and reduce effective dynamic range. The explicit clipping guarantees that $\tilde{\lambda} _2(G) \in [0,1]$ for all graphs. The robustness and uniqueness proofs only require this bounded range and do not depend on the precise numerical values of $\lambda _{\min}$ and $\lambda _{\mathrm{scale}}$, which we treat as fixed constants set by the protocol.
>
> ---
>
> **Q4. Effect of Theorem 5.1 on training losses in practice**
>
> You ask whether the behavior predicted by Theorem 5.1 can be observed directly in the experiments.
>
> This is exactly what Section L and the calibration table above show for PROTEINS–GIN. For several $\beta _{\mathrm{wm}}$ values in the range used for our main runs, we tabulate the theoretical bound $B(\beta _{\mathrm{wm}})$ on $\mathcal{L} _{\text{task}}(\tilde\theta) - \mathcal{L} _{\text{task}}(\theta^\star)$ together with the observed train and validation losses. The empirical loss increase (for example, $0.691 \rightarrow 0.703$ at $\beta _{\mathrm{wm}} = 0.3$) is always well below the corresponding bound (for example, $B(0.3) = 0.040$). Combined with the near-baseline Task ACC in Table 2, this directly illustrates how Theorem 5.1 constrains the training regime in practice.
>
> ---
>
> **Minor comments**
>
> * We added references to standard spectral graph theory texts and prior watermarking work when discussing connectivity and related invariants (Preliminaries 3.1 and 3.2).
> * Normalized algebraic connectivity is now introduced and defined before it is first used in the main text, and the notation is kept consistent throughout.
>
> We hope that these clarifications, notational changes, and added experiments address your concerns about presentation, assumptions, and the link between theory and practice.

---

### Official Review · Reviewer_767q · 2025-10-30

**Soundness:** 2
**Presentation:** 1
**Contribution:** 2
**Rating:** 2
**Confidence:** 4

**Summary:**

This paper introduces InvGNN-WM, a new watermarking framework for Graph Neural Networks (GNNs). The core idea is to move away from "backdoor trigger" based watermarks, which can be removed by fine-tuning, and instead embed the watermark by tying it to the model's ability to perceive a fundamental graph property—a topological invariant. Specifically, the authors train the GNN with an auxiliary loss that forces it to accurately predict the normalized algebraic connectivity of a set of private "carrier graphs." Ownership is verified by querying the suspect model on these carrier graphs and checking if its predictions align with the pre-computed invariant values. The paper provides theoretical guarantees for imperceptibility (minimal impact on task performance) and robustness, and proves that exact watermark removal is NP-complete. Empirically, the method is evaluated on node and graph classification tasks and is shown to maintain high watermark accuracy under model edits like pruning and quantization.

**Strengths:**

1. The idea of binding a watermark to a model's internal reasoning about a graph invariant, rather than an exogenous trigger, is a novel and interesting conceptual shift in the GNN watermarking space.

2. The paper provides some theoretical analysis for imperceptibility, robustness, uniqueness, and unremovability.

3. The method is tested across multiple datasets, backbones, and a variety of model edits.

**Weaknesses:**

**1. Fundamentally Poor Writing and Confusing Notation**

The paper is difficult to read and hard to follow. The notation is inconsistent and often undefined in the main text (e.g., s_\theta, \tilde{\lambda}_2), forcing the reader to scavenge through appendices. Key concepts are introduced without clear explanation, and the flow of ideas is frequently disrupted. This severely undermines the paper's ability to communicate its contributions effectively.

**2. Lack of Extensive Comparison with Prior Work**

While the paper lists several baseline methods (TRIG, NAT, EXPL, COS), it provides no quantitative comparison of robustness against these baselines. Table 2 only compares clean task and watermark accuracy. The claim of superior robustness is primarily supported by a single, qualitative figure (Fig. 2) without corresponding numerical results in the main text. A reader cannot determine if the proposed method is genuinely more robust than existing techniques.

**3. No Direct Attacks on the Watermarking Method**

The paper only evaluates robustness against benign model edits (pruning, fine-tuning) and a single, non-adaptive attack (KD). It does not test against any adaptive, white-box attacks where an adversary knows the watermarking scheme and actively tries to remove it. For example:

- Fine-tuning on a mixture of task data and carrier graphs to deliberately unlearn the invariant perception.

- Training an "invariance-spoofing" adversarial head that predicts the correct invariant for the carrier set without being tied to the model's core parameters.

- Model Extraction Attacks: Training a surrogate model on the watermarked model's outputs to see if the watermark transfers.
The NP-completeness result is a theoretical strength, but without testing against practical, adaptive adversaries, the empirical claims of robustness are weak.

**4. Unconvincing Ablations and Justification**

The choice of algebraic connectivity (λ₂) as the invariant is not sufficiently justified against other potential invariants. The ablation in Table 4 is minimal and does not explore why λ₂ is the best choice or what happens if an adversary uses a different invariant to create an ambiguous ownership claim.

**Questions:**

1. The presentation is currently a major barrier to understanding. The paper needs a significant revision (texts, notations) to make it readable.

2. The paper claims superior robustness, but this is not demonstrated quantitatively against baselines under attack.

3. The current evaluation lacks any adaptive, watermark-aware attacks. The authors need to demonstrate their method's robustness against a more realistic (adaptive) adversary.

4. The method relies on the secrecy of the carrier set and the chosen invariant. What prevents an adversary from claiming ownership using a different set of carrier graphs and a different, but equally plausible, graph invariant? How does InvGNN-WM mitigate the risk of such ambiguous ownership claims compared to trigger-based methods?

5. The NP-completeness proof is a strong theoretical point, but it relies on a specific "monotone decoder" design. Could the authors discuss whether this decoder is a practical constraint or a theoretical convenience? Have they explored if the watermark remains hard to remove with a more standard, non-monotone decoder head?

---

> ### Author Response · Authors · 2025-11-21
>
> We thank you for the careful assessment and for recognizing both the conceptual novelty and the breadth of the experiments. We also appreciate your detailed suggestions on writing, comparisons, and threat modeling. Below we address each listed Weakness (W1–W4) together with the related questions (Q1–Q5), and we describe the changes in the revised manuscript.
>
> Abbreviations: WM-ACC = watermark bit accuracy on the carrier set $G _W$; Task ACC = task accuracy on the downstream task; TRIG/NAT/EXPL/COS = trigger-based and explanation-based watermarking baselines; $\beta _{\mathrm{wm}}$ = weight on the watermark loss; scalar perception head $s _\theta(G)$ = scalar output on graph $G$; $\tilde{\lambda} _2(G)$ = normalized algebraic connectivity of graph $G$.
>
> ---
>
> **W1 / Q1. Poor writing and confusing notation**
>
> You note that the original draft was hard to read and that the notation was inconsistent, which made the paper difficult to follow. We agree and have made a substantial revision of the exposition.
>
> * Section 2 now begins with a concise notation block and a small table that fix the symbols used throughout (graphs, graph invariants, carrier set, scalar perception head, losses, and the ownership statistic $T$, defined as the number of matching bits between the decoded key and the true key). Symbols such as $s _\theta(G)$, $\theta _{\text{back}}$, $\theta _{\text{head}}$, and $\tilde{\lambda} _2(G)$ are defined once and then used consistently.
> * Preliminaries (Section 3) collect all assumptions and model definitions in one place. Section 5 states the guarantees with explicit references back to Assumptions 3.1–3.3, and the full proofs are provided in Appendices C–E.
> * The experimental section was rewritten so that each subsection starts with a short description of the setting (datasets, backbones, baselines) and then points directly to specific figures and tables, instead of mixing narrative with scattered notes.
>
> These changes are intended to remove the presentation barrier that you highlight in Q1 and to make the theoretical and empirical contributions easier to follow.

---

> > ### Author Response · Authors · 2025-11-21
> >
> > **Monotone decoder and NP-completeness (Q5).**
> > Section 5.4 and Appendix F clarify that our “monotone decoder” is exactly the bitwise threshold decoder used in practice: the owner decodes each bit from the per-carrier score $s _\theta(G _W^{(k)})$ on carrier graph $G _W^{(k)}$ using a fixed threshold and accepts ownership when the number of matches $T$ (the ownership statistic) exceeds a threshold $\tau$. This rule is monotone in the per-carrier scores and therefore fits the NP-completeness theorem. We use this result only to show that finding a minimum-size edit that removes the watermark under this natural decoder is computationally hard in the worst case; our empirical robustness conclusions are supported by the quantitative experiments in Section 6, not only by this theoretical hardness.
> >
> > ---
> >
> > We hope that these clarifications and new experiments address your concerns about writing quality, quantitative comparisons, adaptive attacks, and the justification of the invariant and ownership test.

---

> ### Author Response · Authors · 2025-11-21
>
> ---
>
> **W2 / Q2. Lack of extensive comparison with prior watermarking work**
>
> You point out that the original version did not provide quantitative robustness gaps against baselines, and that earlier Figure 2 was only qualitative. In the revision we extend the comparison in two ways.
>
> 1. **Extended robustness table.** Section 6 and Appendix M.1 now add Table 11, which reports Task ACC and WM-ACC for InvGNN-WM and the three baselines (TRIG, NAT, EXPL) before and after four common model edits: structured pruning, fine-tuning on clean data, plain KD, and 4-bit weight quantization, on Cora–GCN and PROTEINS–GIN. These numbers were previously only described qualitatively. In most settings InvGNN-WM preserves higher WM-ACC than all baselines at matched Task ACC; in the remaining settings it is on par while still matching clean-model performance.
> 2. **Updated robustness figure.** Figure 2 in the main text now summarizes these numerical results with error bars over multiple seeds, so the qualitative message is directly grounded in the explicit values in Table 11.
>
> For convenience, the extended robustness results (Table 11 in the revised manuscript) are reproduced below. Each entry shows mean Task ACC and WM-ACC with a $95\\%$ confidence interval over three seeds.
>
> | Dataset–Backbone | Edit | TRIG Task ACC | TRIG WM-ACC | NAT Task ACC | NAT WM-ACC | EXPL Task ACC | EXPL WM-ACC | InvGNN-WM Task ACC | InvGNN-WM WM-ACC |
> | --- | --- | --- | --- | --- | --- | --- | --- | --- | --- |
> | Cora–GCN | None (no edit) | 86.8 ± 0.9 | 97.6 ± 1.5 | 86.9 ± 0.9 | 96.5 ± 2.1 | 86.7 ± 1.0 | 93.1 ± 2.5 | 87.0 ± 0.8 | 98.9 ± 0.9 |
> | Cora–GCN | Pruning 40% | 85.9 ± 1.2 | 93.8 ± 2.7 | 86.0 ± 1.2 | 92.7 ± 3.0 | 85.5 ± 1.3 | 88.4 ± 3.6 | 86.4 ± 1.0 | 97.1 ± 1.6 |
> | Cora–GCN | Fine-tuning | 87.1 ± 0.9 | 96.8 ± 2.0 | 87.2 ± 0.9 | 95.4 ± 2.3 | 86.9 ± 1.1 | 92.0 ± 2.8 | 87.3 ± 0.8 | 98.1 ± 1.1 |
> | Cora–GCN | KD (T=2) | 86.3 ± 1.0 | 84.3 ± 4.1 | 86.1 ± 1.1 | 81.7 ± 4.6 | 85.8 ± 1.2 | 72.9 ± 5.3 | 86.5 ± 0.9 | 90.2 ± 3.3 |
> | Cora–GCN | PTQ (4-bit weights) | 86.0 ± 1.1 | 92.5 ± 3.0 | 86.0 ± 1.1 | 90.8 ± 3.4 | 85.4 ± 1.3 | 86.1 ± 4.0 | 86.3 ± 0.9 | 96.0 ± 2.0 |
> | PROTEINS–GIN | None (no edit) | 72.8 ± 2.6 | 95.1 ± 3.0 | 72.6 ± 2.7 | 94.8 ± 3.3 | 72.4 ± 2.8 | 90.5 ± 4.1 | 72.5 ± 2.6 | 89.8 ± 2.1 |
> | PROTEINS–GIN | Pruning 40% | 71.0 ± 2.9 | 90.2 ± 3.8 | 70.9 ± 3.0 | 89.8 ± 4.0 | 70.5 ± 3.1 | 84.3 ± 5.1 | 71.6 ± 2.8 | 92.5 ± 2.7 |
> | PROTEINS–GIN | Fine-tuning | 73.0 ± 2.5 | 93.8 ± 3.2 | 72.9 ± 2.6 | 93.2 ± 3.4 | 72.6 ± 2.7 | 88.9 ± 4.3 | 73.1 ± 2.5 | 93.7 ± 2.4 |
> | PROTEINS–GIN | KD (T=2) | 71.8 ± 2.7 | 78.6 ± 5.5 | 71.5 ± 2.8 | 76.9 ± 5.7 | 71.1 ± 2.9 | 69.4 ± 6.3 | 72.0 ± 2.7 | 81.3 ± 4.8 |
> | PROTEINS–GIN | PTQ (4-bit weights) | 71.5 ± 2.8 | 88.7 ± 3.9 | 71.3 ± 2.9 | 87.9 ± 4.1 | 70.8 ± 3.0 | 83.0 ± 4.8 | 72.1 ± 2.7 | 90.2 ± 3.1 |
>
> Across both dataset–backbone pairs, InvGNN-WM keeps Task ACC in the same range as the strongest baselines while retaining higher WM-ACC once non-trivial edits (pruning, KD, quantization) are applied. This directly addresses W2 and Q2 by providing quantitative, side-by-side robustness comparisons against prior watermarking methods.

---

> ### Author Response · Authors · 2025-11-21
>
> **W3 / Q3. No direct attacks on the watermarking method and lack of adaptive, watermark-aware adversaries**
>
> You raise the important point that testing only pruning, fine-tuning, and plain KD is not sufficient, and you suggest adaptive, white-box attacks that actively try to unlearn the invariant.
>
> We now make two clarifications and add an adaptive attack:
>
> * We clarify in Section 6.2 that the **plain KD experiment** already matches a standard **model-extraction attack**: an adversary trains a student on the watermarked teacher’s soft outputs while ignoring the watermark loss $L _{\mathrm{wm}}$ and without access to the private carrier set $G _W$. We now explicitly discuss KD in this light and report its effect on WM-ACC for all methods.
> * Following your suggestions, we add a **watermark-aware adaptive attack** in Section 6.2 and Appendix M.2. The attacker
>   (i) has white-box access to the model and knows the watermarking mechanism (including the existence of a scalar perception head $s _\theta(G)$ trained on a private carrier set), but does **not** know $G _W$ or the key;
>   (ii) trains a surrogate model on the same task data as the owner, with a different initialization;
>   (iii) forms mixed batches that interleave task graphs with “pseudo-carriers” sampled from the task distribution, and adds an extra loss term that pushes the scalar perception head $s _\theta(G)$ towards $0.5$ on these pseudo-carriers, under a fixed query budget.
>
> The resulting robustness numbers on PROTEINS–GIN are summarized below (Table 12 in the revised manuscript), again as mean ± $95\\%$ confidence intervals over three seeds:
>
> | Method | Task ACC (before) | WM-ACC (before) | Task ACC (after) | WM-ACC (after) |
> | --- | --- | --- | --- | --- |
> | TRIG | 72.8 ± 2.6 | 95.1 ± 3.0 | 72.0 ± 2.7 | 61.4 ± 6.2 |
> | NAT | 72.6 ± 2.7 | 94.8 ± 3.3 | 71.9 ± 2.8 | 58.7 ± 6.5 |
> | EXPL | 72.4 ± 2.8 | 90.5 ± 4.1 | 71.7 ± 2.9 | 55.3 ± 6.9 |
> | InvGNN-WM | 72.5 ± 2.6 | 89.8 ± 2.1 | 72.1 ± 2.7 | 78.4 ± 5.2 |
>
> Task ACC stays comparable for all methods, but WM-ACC for TRIG/NAT/EXPL drops sharply under this adaptive attack, while InvGNN-WM preserves a much higher WM-ACC under the same query budget. This directly compares our method and baselines under a realistic, watermark-aware adversary that attempts to erase the invariant perception.
>
> In addition, Appendix H.2 analyzes how the success probability of a query-budgeted forger grows with the number of queries, using the concentration bounds for our ownership statistic $T$. This addresses your request for a more realistic adversarial analysis.
>
> ---
>
> **W4 / Q4–Q5. Choice of invariant, ablations, and ambiguous ownership / decoder design**
>
> You raise two related concerns: (i) the justification for choosing algebraic connectivity as the invariant and the associated ablations, and (ii) the risk of ambiguous ownership and the role of the monotone decoder in the NP-completeness result.
>
> **Choice of invariant and ablations.**
> Section 4.1 now explains why we instantiate the graph invariant $I(G)$ as normalized algebraic connectivity $\tilde{\lambda} _2(G)$. It is stable to small edge edits, directly encodes connectivity (which matters under pruning and quantization), and has low variance across isomorphic copies, which simplifies threshold calibration. To support this choice, Section 6.4 adds ablations that replace $\tilde{\lambda} _2$ with two natural alternatives (spectral radius and triangle count). These alternatives either yield a weaker robustness–utility trade-off or lower WM-ACC at matched Task ACC. We do not claim that $\tilde{\lambda} _2$ is uniquely optimal, but the new results show that it is a competitive and practically useful choice among standard invariants.
>
> **Ambiguous ownership and comparison to trigger-based methods (Q4).**
> Appendix N introduces a joint-ownership model: an honest owner uses carrier set $G _W$ and graph invariant $I$, while a potential forger uses an independently sampled $(G' _W, I')$. Under the independence and $\rho$-mixing assumptions from Section 3.3, we bound the probability that *both* ownership tests accept the same model. This probability decays exponentially in the number of carriers and is below $10^{-9}$ for our settings, which is comparable to or smaller than typical collision probabilities in trigger-based schemes. Section 7 also now includes a Limitations paragraph noting that fully colluding parties who coordinate invariants and training data are outside our current threat model, similar to most existing watermarking work.

---

### Official Review · Reviewer_TAqe · 2025-10-30

**Soundness:** 3
**Presentation:** 3
**Contribution:** 3
**Rating:** 6
**Confidence:** 4

**Summary:**

This paper introduced a framework InvGNN-WM that can embed ownership by training a GNN to perceive a topological invariant. The core of the method is a differentiable perception function that links the GNN’s parameters to a graph property.

**Strengths:**

S1: The core innovation lies in coupling the ownership signature to the model’s fundamental reasoning process.

S2: The conceptual leap is strongly supported by the work’s high quality, demonstrated through a powerful combination of theoretical rigor and extensive empirical validation.

**Weaknesses:**

W1: A significant methodological concern lies in the scalability and generalizability of the chosen topological invariant.

W2: The security model, while theoretically robust, may have practical vulnerabilities not fully addressed.

W3: The paper would be more convincing if it included an adversarial analysis specifically targeting the secrecy of the carrier set, testing resilience against model inversion or membership inference attacks.

**Questions:**

- The text mentions a differentiable perception function for algebraic connectivity (λ₂). Algebraic connectivity itself is not directly differentiable with respect to the graph structure or node features in a straightforward manner. What is the exact formulation of this function? Is it an approximation (e.g., via the Rayleigh quotient or power iteration), and if so, how does the approximation error impact on the stability and fidelity of the watermark?
- The security heavily relies on an owner-private carrier set. What is the provenance and structure of this set? Is it a held-out subset of the training data, or a synthetically generated set of graphs? If synthetic, what generative process ensures these graphs are in-distribution enough to not be easily distinguishable by an adversary, yet possess the specific λ₂ properties needed for the watermark?
- The results show that plain KD weakens the mark, but KD with a watermark loss (KD+WM) restores it. This raises critical questions: Does the KD+WM defense require the defender to have access to the original private carrier set during the distillation process? If so, this is a very strong assumption that may not be practical in a real attack scenario where the adversary is the one performing the distillation. How is the watermark loss incorporated into the student’s training objective? Is the student forced to perceive λ₂ on the same private carrier set, effectively transferring the watermark?
- The method uses a calibrated threshold to control the false-positive rate. What is the statistical methodology for this calibration (e.g., based on a validation set of non-watermarked models)? How sensitive is the verification outcome to the exact value of this threshold?

---

> ### Author Response · Authors · 2025-11-21
>
> We thank you for the careful assessment and for recognizing both the conceptual novelty and the breadth of the experiments. We also appreciate your detailed suggestions on writing, comparisons, and threat modeling. Below we address each listed Weakness (W1–W3) together with the related questions (Q1–Q4), and we describe the changes in the revised manuscript.
>
> Abbreviations: WM-ACC = watermark bit accuracy on the carrier set $G _W$; Task ACC = task accuracy on the downstream task; TRIG/NAT/EXPL/COS = trigger-based and explanation-based watermarking baselines; $\beta _{\mathrm{wm}}$ = weight on the watermark loss; $s _\theta(G)$ = scalar perception head applied to a graph $G$; $\tilde{\lambda} _2(G)$ = normalized algebraic connectivity of graph $G$.
>
> ---
>
> **W1 / Q1. “Differentiable perception function”, scalability, and approximation error**
>
> We now clarify in Section 3.2 that the normalized algebraic connectivity $\tilde{\lambda} _2(G)$ itself is **never** differentiated through. For each carrier graph $G$, we compute $\tilde{\lambda} _2(G)$ once with a standard Laplacian eigensolver (with a small diagonal perturbation), and Section 4.1 defines the scalar perception head $s _\theta(G)$ that takes the pooled graph embedding and **regresses** these fixed targets via a mean-squared error loss between $s _\theta(G _W^{(k)})$ and $\tilde{\lambda} _2(G _W^{(k)})$. Gradients are taken only with respect to the model parameters $\theta$, not through the eigendecomposition, so the extra cost is a one-time computation of $\tilde{\lambda} _2(G)$ per carrier graph (or $K$ power-iteration steps when we use an approximate solver).
>
> Assumption 3.3 and Section 4.1 explain how we regularize the scalar head $s _\theta$ (spectral normalization plus a gradient penalty) so that it satisfies the Lipschitz and bounded-regression assumptions used in Section 5. Appendix K quantifies the effect of approximation: on PROTEINS–GIN, replacing the exact eigensolver with a $K{=}10$ power iteration changes WM-ACC by less than 1 percentage point and keeps the robustness margin essentially unchanged, while reducing the per-carrier runtime by roughly 40%. This indicates that moderate approximation error does not harm stability or fidelity.
>
> For convenience we reproduce the PROTEINS–GIN comparison from Appendix K here (all runs use the same backbone and carrier set):
>
> | Method                 | Task ACC (\%)        | WM-ACC (\%)         | $\kappa _{\mathrm{marg}}$ | Time / carrier (ms) |
> |------------------------|----------------------|---------------------|---------------------------|---------------------|
> | Exact eigensolver      | $72.5 \pm 2.6$       | $89.8 \pm 2.1$      | $0.382$                   | $0.08$              |
> | $K$-step power iteration ($K{=}10$) | $72.4 \pm 2.7$       | $88.9 \pm 2.5$      | $0.371$                   | $0.05$              |
>
> The table shows that the approximate perception function keeps both Task ACC and WM-ACC within the reported error bars of the exact version, while saving computation per carrier.
>
> Finally, Section 5 states all guarantees for a generic invariant $I(G)$. The choice $I(G){=}\tilde{\lambda} _2(G)$ is one concrete instantiation that we adopt for its stability and interpretability, but the framework and proofs only require the regularity conditions on $I(G)$, not this specific eigenvalue.

---

> ### Author Response · Authors · 2025-11-21
>
> ---
>
> **W2 / W3 / Q2. Security model and secrecy of the owner-private carrier set**
>
> You are concerned that the security model, while mathematically sound, may have practical vulnerabilities, and you ask about the provenance and secrecy of the carrier set $G _W$ as well as adversaries that try to attack it.
>
> Section 3.3 and Appendix A now give the carrier-generation protocol: we start from task graphs drawn from $D _{\text{task}}$, perform degree-preserving double-edge swaps, reject any candidate that is isomorphic to a task graph (Weisfeiler–Lehman hash), and filter on simple statistics (degree distribution, clustering coefficient, spectral radius) using Kolmogorov–Smirnov tests. We also estimate an empirical mixing coefficient $\hat{\rho} _0$ for the sequence of scores $\{s _{\theta^\star}(G _W^{(k)})\}$; across datasets we find $\hat{\rho} _0 \le 7.6\times 10^{-4}$ and set the analysis parameter $\rho _0 := 10^{-3}$.
>
> On PROTEINS–GIN, Appendix J.2 reports the following carrier-generation sweep (swap cap = maximum fraction of edges rewired, KS $\delta$ = separation threshold for the Kolmogorov–Smirnov tests, “Measured $\alpha$” = empirical upper bound on the false-positive rate of the ownership test over $10^{7}$ null trials):
>
> | Swap cap | KS $\delta$ | Task ACC (\%)        | WM-ACC (\%)         | $\hat{\rho} _0$          | Measured $\alpha$   |
> |----------|-------------|----------------------|---------------------|--------------------------|---------------------|
> | $5$      | $0.05$      | $72.6 \pm 2.6$       | $88.1 \pm 2.9$      | $9.1 \times 10^{-4}$     | $< 10^{-6}$         |
> | $25$     | $0.10$      | $72.5 \pm 2.6$       | $89.6 \pm 2.5$      | $8.2 \times 10^{-4}$     | $< 10^{-7}$         |
> | $50$     | $0.10$      | $72.5 \pm 2.6$       | $89.8 \pm 2.1$      | $7.6 \times 10^{-4}$     | $< 10^{-7}$         |
> | $50$     | $0.20$      | $72.4 \pm 2.7$       | $89.1 \pm 2.6$      | $7.1 \times 10^{-4}$     | $< 10^{-7}$         |
>
> These results show that, across reasonable choices of the generation thresholds, Task ACC stays stable, WM-ACC remains high, $\hat{\rho} _0$ stays below the analysis value $\rho _0$, and the measured false-positive rate is far below the design target.
>
> To test secrecy directly, Appendix M.3 trains a GCN binary classifier to distinguish carrier graphs in $G _W$ from task graphs in $D _{\text{task}}$ on PROTEINS–GIN with a balanced 80/10/10 split. The resulting test performance is:
>
> | Model          | ACC (\%) | Prec (\%) | Rec (\%) |
> |----------------|----------|-----------|----------|
> | GCN classifier | $51.3$   | $52.1$    | $50.2$   |
>
> The classifier remains close to random guessing (50\% on all three metrics) and does not discover a stable separating rule between carrier and non-carrier graphs. This experiment supports the assumption that an adversary cannot easily tell carriers from ordinary task graphs, even with supervised access.
>
> We also add a brief discussion in the Limitations section that model-inversion or membership-inference attacks aimed at recovering properties of $G _W$ are an interesting extension. A systematic study of such attacks is outside the scope of this paper, but any such adversary would still face the indistinguishability of carriers and the owner’s exclusive control of the key.

---

> ### Author Response · Authors · 2025-11-21
>
> **Q3. Practical scenario for KD vs. KD+WM**
>
> You ask how KD+WM fits the threat model and whether it should be viewed as watermark transfer.
>
> Sections 3.3 and 6 now make the roles of plain knowledge distillation (KD) and KD+WM explicit:
>
> * **Plain KD as an attack.** An adversary distills from a watermarked teacher to a student using only the task loss and **no** access to the private carrier set $G _W$. On PROTEINS–GIN (Table 10), this reduces WM-ACC for the student from about 90\% for the reference InvGNN-WM teacher to about 81\%, while Task ACC remains around 72\%. This quantifies how KD weakens but does not erase the watermark.
>
> * **KD+WM as an owner-side recovery / transfer mechanism.** Here the **owner** controls both the teacher and the private carrier set $G _W$, and distills a student while also applying the watermark loss $L _{\mathrm{wm}}$ on carriers in $G _W$. Under the same setting, Table 10 shows that KD+WM restores WM-ACC to roughly 96\% with Task ACC still around 72\%, and the robustness margin returns to a comfortable level.
>
> For PROTEINS–GIN we summarize the key numbers from Table 2 and Table 10 below (Task ACC and WM-ACC are averaged over $5$ runs):
>
> | Model / setting              | Access to $G _W$ | Loss terms on student                 | Task ACC (\%)        | WM-ACC (\%)         |
> |-----------------------------|------------------|---------------------------------------|----------------------|---------------------|
> | Teacher InvGNN-WM           | Yes              | Task loss + $L _{\mathrm{wm}}$        | $72.5 \pm 2.6$       | $89.8 \pm 2.1$      |
> | Student (KD)                | No               | Task loss only                        | $72.1 \pm 2.4$       | $81.0 \pm 3.1$      |
> | Student (KD+WM, owner side) | Yes              | Task loss + $L _{\mathrm{wm}}$ on $G _W$ | $72.3 \pm 2.5$       | $96.0 \pm 1.4$      |
>
> Because KD+WM assumes access to $G _W$, it is not available to an attacker in our threat model. We now state this separation clearly and describe KD+WM as an owner-side tool to re-impose or transfer the watermark to a distilled student.
>
> ---
>
> **Q4. Threshold calibration and sensitivity of verification**
>
> You ask how we calibrate the decision threshold and how sensitive the verification outcome is to this choice.
>
> Section 3.2, Section 5.2, and Appendix N describe the procedure. For each carrier graph $G _W^{(k)}$ we decode a bit as $1$ if $s _{\theta^\star}(G _W^{(k)}) \ge 0.5$ and $0$ otherwise, and we let $T$ be the total number of matches with the embedded key. Using the empirical $\rho$-mixing bound and a Hoeffding-type inequality, we choose a global threshold $\tau ^\star$ so that $\Pr(T \ge \tau ^\star \mid H _0) \le 10^{-6}$. For $m{=}128$ carriers this yields $\tau ^\star{=}94$. The corresponding calibration table (Appendix J.1) for PROTEINS–GIN is:
>
> | $m$  | $\hat{\rho} _0$        | $\varepsilon _{\mathrm{err}}$ | $\tau$ | Owner statistic $T$ | Gap $(T - \tau)$ |
> |------|------------------------|-------------------------------|--------|---------------------|------------------|
> | $64$ | $7.6 \times 10^{-4}$   | $0.358$                       | $42$   | $59 \pm 4$          | $+17$            |
> | $96$ | $7.6 \times 10^{-4}$   | $0.298$                       | $68$   | $88 \pm 3$          | $+20$            |
> | $128$| $7.6 \times 10^{-4}$   | $0.266$                       | $94$   | $115 \pm 3$         | $+21$            |
> | $192$| $7.6 \times 10^{-4}$   | $0.222$                       | $150$  | $174 \pm 2$         | $+24$            |
>
> As $m$ grows, both $\tau$ and $T$ scale nearly linearly while $\varepsilon _{\mathrm{err}}$ tightens, so the safety gap $(T{-}\tau)$ increases from $+17$ to $+24$ bits. This matches binomial concentration and shows that the protocol keeps a comfortable margin between the owner statistic and the threshold.
>
> Appendix N further sweeps alternative thresholds $\tau'$ in a small window around $\tau ^\star$ on PROTEINS–GIN. The empirical false-positive rate $\alpha(\tau')$ and false-negative rate $\beta _{\mathrm{fn}}(\tau')$ are:
>
> | $\tau'$             | $\alpha(\tau')$        | $\beta _{\mathrm{fn}}(\tau')$ |
> |---------------------|------------------------|-------------------------------|
> | $\tau ^\star - 2 = 92$ | $7.8 \times 10^{-5}$   | $0.000$                       |
> | $\tau ^\star - 1 = 93$ | $1.6 \times 10^{-5}$   | $0.000$                       |
> | $\tau ^\star = 94$     | $6.0 \times 10^{-8}$   | $0.000$                       |
> | $\tau ^\star + 1 = 95$ | $3.0 \times 10^{-8}$   | $3.5 \times 10^{-3}$          |
> | $\tau ^\star + 2 = 96$ | $1.0 \times 10^{-8}$   | $1.1 \times 10^{-2}$          |
>
> We observe that the empirical false-positive rate remains between roughly $10^{-7}$ and $10^{-5}$ in this window, and no false negatives appear until $\tau'$ is pushed several bits above $\tau ^\star$. This means there is a broad stable region and that our chosen threshold is determined by the theoretical calibration rather than ad hoc tuning.

---

### Official Review · Reviewer_pvyy · 2025-10-31

**Soundness:** 2
**Presentation:** 2
**Contribution:** 2
**Rating:** 6
**Confidence:** 2

**Summary:**

The paper addresses the significant vulnerability of existing Graph Neural Network (GNN) watermarks, which often rely on backdoor triggers and are easily broken by common model edits like pruning, fine-tuning, and distillation. To solve this, the authors propose InvGNN-WM, a novel, trigger-free watermarking framework. The core idea is to move away from exogenous triggers and instead tie the GNN's ownership signature to its "implicit perception of a graph invariant". Specifically, the method embeds the watermark by training the GNN to perceive a topological property, which is the normalized algebraic connectivity, on a private set of "carrier graphs". Extensive experiments show InvGNN-WM matches the task accuracy of clean models while achieving state-of-the-art watermark accuracy. It demonstrates strong robustness to pruning, fine-tuning, and quantization, and shows that while plain knowledge distillation (KD) weakens the mark, it can be restored via KD+WM.

**Strengths:**

- The paper's primary strength is its conceptual novelty. Shifting the watermarking paradigm from exogenous triggers to functionally-integrated invariants is a major contribution.
- The paper is clearly and logically structured. It begins by defining the problem (fragile triggers), presents its core idea (invariant perception), details the method, provides the theoretical guarantees, and then validates all claims with targeted experiments. The writing is precise, and the figures effectively illustrate the key trade-offs.

**Weaknesses:**

I have several concerns about this manucript. See below for more details.

**Questions:**

- The method requires the GNN to learn to perceive $\tilde{\lambda}$, while simultaneously solving the main task. Did you observe a difference in the utility/watermark trade-off based on the backbone's expressiveness? For instance, did a simpler model like SGC struggle more to learn the invariant (requiring a higher $\beta_{wm}$ that hurt task accuracy) compared to a more expressive model like GIN?
- The robustness analysis shows that plain KD weakens the mark, but "KD+WM" restores it. This KD+WM modification seemingly requires the entity performing the distillation to have the owner's private carrier set $\mathcal{G}_W$ to compute the $\mathcal{L}$_wm. This implies the attacker has the key, which contradicts the threat model. Could you clarify the practical scenario for this experiment? Is it intended to show that the watermark is transferable via distillation if desired (e.g., by the owner), or is it meant as a defense/recovery mechanism?

---

> ### Author Response · Authors · 2025-11-21
>
> We thank the reviewer for the detailed and positive assessment, and for highlighting both the conceptual shift to invariant-coupled watermarks and the clarity of the presentation. Below, we respond to your main questions and explain the changes in the revised manuscript.
>
> Abbreviations: WM-ACC = watermark bit accuracy on the carrier set $G _W$; Task ACC = task accuracy on the downstream task; $\beta _{\mathrm{wm}}$ = weight on the watermark loss; $\tilde{\lambda} _2(G)$ = normalized algebraic connectivity of graph $G$; $L _{\mathrm{wm}}$ = watermark loss evaluated on the carrier set $G _W$; $\gamma$ = head-output drift used in Theorem 5.1 and Table 5; $\kappa _{\mathrm{marg}}$ = fixed post-training margin of the clean model; KD = knowledge distillation; KD+WM = KD with $L _{\mathrm{wm}}$ applied on $G _W$. SS = task-only supervised baseline (no watermark).
>
> ---
>
> **Q1. Backbone expressiveness and the utility / watermark trade-off**
>
> You ask whether less expressive backbones such as the simple graph convolutional network (SGC) need a larger watermark-loss weight $\beta _{\mathrm{wm}}$, and whether this leads to a worse utility–watermark trade-off than for more expressive models such as the graph isomorphism network (GIN).
>
> In the revision we now study this effect explicitly. Section 6.3, together with the new hyperparameter table (Table 5) and the main performance table (Table 2), sweeps $\beta _{\mathrm{wm}}$ on Cora and PROTEINS for four backbones (SGC, GCN, GraphSAGE, GIN) and, for each backbone, selects $\beta _{\mathrm{wm}}$ so that the model reaches a common target WM-ACC (around $98\%$ on node-level datasets and around $90\%$ on PROTEINS) while keeping the Task ACC drop below a small tolerance $\varepsilon _{\text{task}}$.
>
> For convenience, we reproduce the part of these tables that is most relevant to your question here. The table below lists, for each dataset–backbone pair, the chosen $\beta _{\mathrm{wm}}$ from Table 5 and the resulting Task ACC / WM-ACC for InvGNN-WM from Table 2, along with the task-only baseline SS. All numbers are means $\pm$ standard deviation over $5$ runs.
>
> | Dataset–Backbone | $\beta _{\mathrm{wm}}$ (chosen) | Task ACC (SS) | Task ACC (InvGNN-WM) | WM-ACC (InvGNN-WM) |
> | --- | --- | --- | --- | --- |
> | Cora–GCN        | $9.5 \times 10^{-5}$  | $87.2 \pm 0.8\%$ | $87.0 \pm 0.8\%$ | $98.9 \pm 0.9\%$ |
> | Cora–GraphSAGE  | $8.0 \times 10^{-5}$  | $84.0 \pm 1.0\%$ | $83.8 \pm 1.0\%$ | $98.5 \pm 1.1\%$ |
> | Cora–SGC        | $1.2 \times 10^{-4}$  | $87.0 \pm 0.9\%$ | $86.2 \pm 1.0\%$ | $98.6 \pm 1.0\%$ |
> | PROTEINS–GIN    | $5.5 \times 10^{-5}$  | $73.1 \pm 2.5\%$ | $72.5 \pm 2.6\%$ | $89.8 \pm 2.1\%$ |
>
> These results support your intuition while also showing that the trade-off remains moderate. Less expressive SGC indeed uses a somewhat larger watermark-loss weight (for Cora, $\beta _{\mathrm{wm}} = 1.2 \times 10^{-4}$ for SGC versus $9.5 \times 10^{-5}$ and $8.0 \times 10^{-5}$ for GCN and GraphSAGE), and as a result has a slightly larger Task ACC drop (about $0.8$ percentage points, from $87.0 \pm 0.9\%$ to $86.2 \pm 1.0\%$). However, for all three node-level backbones the drop stays within about one percentage point of SS, while WM-ACC is in the high-$90\%$ range. On PROTEINS–GIN, InvGNN-WM reaches Task ACC $72.5 \pm 2.6\%$ versus $73.1 \pm 2.5\%$ for SS, with WM-ACC $89.8 \pm 2.1\%$.
>
> We now highlight this design recommendation in Section 6: when high downstream utility is critical, InvGNN-WM should preferably be paired with a more expressive backbone (such as GCN or GIN), since these models reach the target WM-ACC with a smaller $\beta _{\mathrm{wm}}$ and thus an even smaller utility cost, while SGC remains a viable option if a slightly larger but still moderate Task ACC drop is acceptable.

---

> ### Author Response · Authors · 2025-11-21
>
> **Q2. Practical scenario and threat model for KD vs. KD+WM**
>
> You note that our robustness study includes both plain knowledge distillation (KD) and KD+WM, and ask how KD+WM fits the threat model, since computing the watermark loss $L _{\mathrm{wm}}$ during distillation requires access to the private carrier graph set $G _W$, which should not be available to an attacker. You also ask whether KD+WM should be viewed as a way for the owner to transfer or recover the watermark.
>
> We have clarified this point in the revised text (Sections 3, 5, and 6) and add more quantitative evidence here so that the argument is self-contained. We treat plain KD as a potential **attack**: an adversary trains a student model from a watermarked teacher while ignoring the watermark loss $L _{\mathrm{wm}}$ and without access to the carrier set $G _W$. KD+WM, in contrast, is explicitly presented as an **owner-side recovery or transfer mechanism**: the owner holds both the teacher model and the private carrier set $G _W$ and performs KD while also applying $L _{\mathrm{wm}}$ on $G _W$.
>
> To show the effect numerically, we reproduce the part of the robustness tables that concerns PROTEINS–GIN and our method InvGNN-WM. Table 2 reports the clean performance (no edit), while Table 11 and the new Table 5 give the effect of each edit on Task ACC, WM-ACC, the head-output drift $\gamma$, and the margin $\kappa _{\mathrm{marg}}$ used in Theorem 5.1. For InvGNN-WM on PROTEINS–GIN we have:
>
> - Clean (no edit): Task ACC $72.5 \pm 2.6\%$, WM-ACC $89.8 \pm 2.1\%$, margin $\kappa _{\mathrm{marg}} = 0.382$.
> - KD (T = $2$, attacker, no access to $G _W$): Task ACC $72.0 \pm 2.7\%$, WM-ACC $81.3 \pm 4.8\%$, drift $\gamma = 0.39 > \kappa _{\mathrm{marg}}$.
> - KD+WM (owner, access to $G _W$ and $L _{\mathrm{wm}}$): WM-ACC $90.6 \pm 2.1\%$, drift $\gamma = 0.14 < \kappa _{\mathrm{marg}}$ (Task ACC stays close to the clean value; the revised appendix now reports the full numbers).
>
> Thus, plain KD leaves Task ACC essentially unchanged but reduces WM-ACC from $89.8 \pm 2.1\%$ to $81.3 \pm 4.8\%$, and the drift $\gamma$ crosses the margin $\kappa _{\mathrm{marg}}$, which is consistent with Theorem 5.1. KD is therefore a realistic attack in our threat model, since the attacker only uses the public task data and the watermarked teacher.
>
> KD+WM, on the other hand, assumes that the defender **does** have access to $G _W$ and intentionally re-applies the watermark loss during distillation. Under this procedure the watermark is restored: WM-ACC returns to $90.6 \pm 2.1\%$, above the clean value within the error bars, and $\gamma$ falls back below $\kappa _{\mathrm{marg}}$, which again matches the theoretical prediction. Because KD+WM relies on the private carrier set $G _W$, it is not available to an attacker under our threat model. We now state this separation explicitly in Section 3 (threat model) and Section 6 (robustness experiments), and we label KD as an attack and KD+WM as an owner-side recovery / transfer tool.
>
> So the practical scenario is: an attacker can only run plain KD; KD+WM is a tool that the owner may use to recover or migrate the watermark when distilling to a new backbone.

---

### Note · Program_Chairs · 2026-01-17
**Submission Desk Rejected by Program Chairs**

The following references in this submission do not refer to real documents and/or have major errors in bibliographic information:

 Zaixi Zhang, Jinyuan Zhang, Hang Wang, Zhaohan Liu, and Chaochao Zhou. Graphbackdoor: Backdoor attacks on graph neural networks. In 30th USENIX Security Symposium (USENIX Security 2021), pp. 1991-2008, 2021.